# DIFFERENTIALLY PRIVATE LEWIS WEIGHT COMPUTATION

## ABSTRACT

Lewis weight is a row leverage score for data matrices. It allows selecting a small number of important rows to approximate the original matrix with provably small error. Computing Lewis weights has long been a key problem in optimization, machine learning, and large-scale data analysis. Despite the significant advancement in the computational efficiency of Lewis Weights, privacy concerns regarding the weight computation are naturally rising. In this work, we propose a privacy-preserving Lewis weight computation with high efficiency and a differential privacy (DP) guarantee. Our theoretical results clearly demonstrate the proposed algorithm's convergence and privacy assurances, providing an effective solution to the trade-off between utility and privacy in Lewis weight computation.

## 1 INTRODUCTION

The Lewis weight is a row leverage score for data matrices. It allows selecting a small number of important rows to approximate the original matrix with provably small error. Computing Lewis weights has long been a key problem in optimization, machine learning, and large-scale data analysis. The Lewis weight has a broad range of real-world applications, covering linear programming (Lee & Sidford, 2014), robotics and control (Dabbene et al., 2017; Tang et al., 2024), collision detection (Rimon & Boyd, 1997), bandit learning (Bubeck et al., 2012a; Hazan & Karnin, 2016), Markov Chain Monte Carlo sampling (Chen et al., 2018), and portfolio optimization costs (Shen & Wang, 2015). Specifically, computing the Lewis weight involves finding a fixed point of the leverage score mapping for an arbitrary non-degenerate[1] matrix $A \in \mathbb{R}^{m \times n}$, which is defined as follows:

**Definition 1.1** (Leverage score). *For an arbitrary non-degenerate matrix $A \in \mathbb{R}^{m \times n}$, its leverage score $\sigma(A) \in \mathbb{R}^m_{>0}$ is defined by*

$$\sigma(A) := \mathrm{Diag}(A(A^\top A)^{-1}A^\top).$$

**Definition 1.2** (Lewis weights). *Let $p > 0$ and let $A \in \mathbb{R}^{m \times n}$ be a non-degenerate matrix. The $\ell_p$ Lewis weights of $A$ are the unique positive vector $w_p(A) \in \mathbb{R}^m_{>0}$ satisfying the fixed-point equation*

$$w_p(A) := \sigma(W^{1/2-1/p}A),$$

*where $W := \mathrm{diag}(w_p(A))$, and $\sigma(\cdot)$ denotes the vector of leverage scores in Definition 1.1.*

Recently, Lee & Sidford (2019) introduced an efficient method to compute Lewis weights by running projected gradient descent in a carefully scaled space and using a homotopy scheme to obtain a good initialization. As a result, Lewis weights can now be computed in practical time.

Despite the significant advancement in the computational efficiency of Lewis Weights, privacy concerns regarding the weight computation are naturally rising. It is specifically crucial to determine the value of the Lewis weights with a specific matrix $A$, keeping useful statistical information while not revealing sensitive information. For instance, in bandit learning scenarios, our goal is to ensure the privacy of sensitive pay-off values in each round while still maintaining a policy that results in minimal regret. Therefore, in this work, we aim to answer this fundamental research question:

---

[1] In numerical linear algebra, it is common to assume $A$ is non-degenerate (Brand et al., 2020; 2021; Fazel et al., 2022), avoiding pseudo-inverses. This standard simplification does not restrict our results: any degenerate $A$ can be reduced to a non-degenerate subproblem and the solution then mapped back.

*Can we preserve the privacy of individual data points in fast Lewis weight computation?*

We provide an affirmative answer to this question by employing privacy-preserving Lewis weight computation from a Differential Privacy (DP) perspective. Specifically, by integrating DP into the framework, our method achieves an optimal balance between Lewis weight utility and data privacy, allowing downstream applications to extract meaningful insights from data while ensuring that no individual data points can be distinguished from the computed weights. Moreover, the strong privacy guarantees provided by DP enhance compliance with various data protection regulations, including but not limited to GDPR and CCPA, fostering trust among data-centric technologies, users, and regulatory agencies.

## 1.1 OUR CONTRIBUTIONS

We commence by introducing some fundamental concepts in differential privacy (DP). Consider the case where the Lewis weight $w_p(A)$ is computed for a given matrix $A$. Replacing a single row of the matrix can lead to significant variations in the computed weights. Consequently, the standard definition of neighboring data points in DP may not hold, motivating us to introduce a new formulation–the $\epsilon_0$-closed neighborhood of matrices/datasets. Specifically, we define two matrices/datasets as neighboring if they are $\epsilon_0$-close.

**Definition 1.3** (Neighboring matrices). *Let $A, A' \in \mathbb{R}^{m \times n}$ be two matrices. We say that $A$ and $A'$ are $\epsilon_0$-close if there exists exact one $i \in [m]$ such that $\|A_{i,*} - A'_{i,*}\|_2 \leq \epsilon_0$ and $A_{j,*} = A'_{j,*}$ holds for all $j \in [m] \setminus \{i\}$.*

Thus, the formal definition of differential privacy is given as follows:

**Definition 1.4** (Differential Privacy). *A randomized algorithm $\mathcal{A} : \mathcal{D} \to \mathcal{R}$ with domain $\mathcal{D}$ and range $\mathcal{R}$ satisfies $(\epsilon, \delta)$-differential privacy if for any two neighboring datasets $D, D' \in \mathcal{D}$ and for any subset of outputs $\mathcal{S} \in \mathcal{R}$ it holds that*

$$\Pr[\mathcal{M}(\mathcal{D}) \in \mathcal{S}] = e^\epsilon \Pr[\mathcal{M}(\mathcal{D}') \in \mathcal{S}] + \delta.$$

In this work, we present the first algorithm for efficiently computing Lewis weights with a differential privacy guarantee. Our theoretical results clearly demonstrate the algorithm's convergence and privacy assurances, providing an effective solution to the trade-off between utility and privacy in Lewis weight computation.

**Theorem 1.5** (Main Result, Informal Version of Theorem 4.7). *Under some mild conditions, there exists a differentially private algorithm that approximately compute the $\ell_p$-Lewis weight for any $p \in (0, 4)$.*

Our contributions can be summarized as follows:

- **Differentially Private Optimization:** We establish the final differential privacy (DP) guarantee of the Lewis weight computation algorithm, leveraging a novel DP-optimization analytical framework specifically designed for truncated Gaussian noise.
- **Fast DP-LW Convergence:** We conduct a convergence analysis of our optimized DP-LW algorithm (Algorithm 1), demonstrating its DP guarantee under truncated Gaussian noise perturbation.
- **Generalizable Perturbation Analysis:** We present a comprehensive study of weighted leverage score perturbation, highlighting its applicability to a range of fundamental problems in machine learning, including kernel regression.

**Roadmap.** In Section 2, we extensively review the relevant prior works for this paper. In Section 3, we present the basic notations and background of DP. In Section 4, we show our main algorithm and its corresponding DP guarantee. In Section 5, we conclude our paper.

## 2 RELATED WORK

**John Ellipsoid Algorithm and Its Applications.** The John Ellipsoid Algorithm, initially proposed by John (1948), provides a powerful method for approximating any convex polytope by its

maximum volume inscribed ellipsoid. This foundational work has spurred extensive research into optimization techniques for solving the John Ellipsoid problem within polynomial time constraints. Among the seminal contributions, Khachiyan (1996); Kumar & Yildirim (2005) introduced first-order methods, which significantly improved computational efficiency. Furthermore, Nesterov & Nemirovskii (1994); Khachiyan & Todd (1990); Sun & Freund (2004) developed approaches utilizing interior point methods to enhance the precision and speed of solving the John Ellipsoid problem. Recent advancements have continued to push the boundaries of this algorithm. Cohen et al. (2019) employed fixed point iteration techniques, leading to the derivation of a more robust solution to the John Ellipsoid. Moreover, they introduced innovative sketching techniques that accelerated computational processes. Building on this, Song et al. (2022c) integrated leverage score sampling into these sketching techniques, further optimizing the algorithm's performance, and Li et al. (2024b) used quantum techniques to further speed up the computation of John Ellipsoids. The implications of the John Ellipsoid Algorithm extend far beyond theoretical mathematics, impacting various fields. In the realm of linear bandit problems, research by Bubeck et al. (2012b); Hazan & Karnin (2016) has shown significant advancements. Experimental design methods have also seen improvements due to contributions from Atwood (1969); Allen-Zhu et al. (2017). In linear programming, the algorithm has provided enhanced solutions, with notable work by Lee & Sidford (2013a). Control theory applications have been advanced through research by Tang et al. (2024), and cutting plane methods have been refined as demonstrated by Tarasov (1988). The algorithm's influence in statistics is also noteworthy; for instance, it plays a critical role in Markov chain techniques for sampling convex bodies, as explored by Huang (2018) and developed for random walk sampling by Vempala (2005); Chen et al. (2018).

**Differential Privacy Analysis and Applications.** Differential privacy has become one of the most essential standards for data security and privacy protection since it was proposed in Dwork et al. (2006). There are plenty of related work focusing on providing a guarantee for existing algorithms, data structures, and machine learning by satisfying the definition of differential privacy, such as Esfandiari et al. (2022); Andoni et al. (2023); Cherapanamjeri et al. (2023); Cohen-Addad et al. (2022); Dong et al. (2024); Farhadi et al. (2022); Gopi et al. (2023); Li et al. (2022); Gopi et al. (2022); Huang & Yi (2021); Jung et al. (2019); Li & Li (2024); Epasto et al. (2024); Chen et al. (2022); Farhadi et al. (2022); Beimel et al. (2022); Narayanan (2022; 2023); Fan & Li (2022); Fan et al. (2024); Li & Li (2023); Eliáš et al. (2020); Yu et al. (2024); Liang et al. (2024); Gu et al. (2024); Song et al. (2023b); Qin et al. (2022); Song et al. (2023a); Galli et al. (2024); Chen et al. (2024); Romijnders et al. (2024); Qi et al. (2024); Ke et al. (2025); Hu et al. (2024); Liu et al. (2024). In addition, recently, there are emerging privacy mechanisms that improve traditional privacy guarantees, such as Gaussian, Exponential, and Laplacian mechanisms (Dwork et al., 2014). For example, Geng et al. (2020) introduced a truncated Laplace mechanism, which has been demonstrated to achieve the tightest bounds among all $(\epsilon, \delta)$-DP distribution.

**Sketching and Leverage Score.** Our work improves the efficiency of the John Ellipsoid algorithm by leveraging sketching and score sampling. Sketching, a widely used technique, has advanced numerous domains, including neural network training, kernel methods (Lee et al., 2020; Song et al., 2021), and matrix sensing (Deng et al., 2023). It has been applied to distributed problems (Woodruff & Zhong, 2016; Boutsidis et al., 2016), low-rank approximation (Clarkson & Woodruff, 2017a; Razenshteyn et al., 2016; Song et al., 2017), and generative adversarial networks (Xiao et al., 2018). In addition, projected gradient descent (Xu et al., 2021), tensor-related problems (Li et al., 2017; Diao et al., 2018), and signal interpolation (Song et al., 2022a) have benefited significantly from sketching. Leverage scores, introduced by Drineas et al. (2006a;b), are pivotal in linear regression and randomized linear algebra, optimizing tasks such as matrix multiplication, CUR decompositions (Mahoney & Drineas, 2009; Song et al., 2019), and tensor decompositions (Song et al., 2019). Moreover, leverage score sampling can be used in kernel learning (Erdélyi et al., 2020). Recent research has further extended the application of leverage score sampling. Studies by Agarwal et al. (2017); Charalambides et al. (2024); Woodruff & Zandieh (2022); Lee et al. (2020); Rudi et al. (2018) have demonstrated the ability to leverage score sampling to significantly enhance the efficiency of various algorithms and computational processes. These advancements underscore the versatility and effectiveness of leverage scores in optimizing performance across diverse fields.

**Linear Programming and Semidefinite Programming** Linear programming is a fundamental computer science and optimization topic. The Simplex algorithm, introduced in Dantzig (1951), is a pivotal method in linear programming, though it has an exponential runtime. The Ellipsoid method, which reduces runtime to polynomial time, is theoretically significant but often slower in practice compared to the Simplex method. The interior-point method, introduced in Karmarkar (1984), is a major advancement, offering both polynomial runtime and strong practical performance on real-world problems. This method opened up a new avenue of research, leading to a series of developments aimed at speeding up the interior point method for solving a variety of classical optimization problems. John Ellipsoid has deep implication in the field of linear programming. For example, in interior point method, John Ellipsoid is utilized to find path to solutions (Lee & Sidford, 2014). The interior point method has a wide impact on linear programming as well as other complex tasks, such as Vaidya (1987); Renegar (1988); Vaidya (1989); Daitch & Spielman (2008); Lee & Sidford (2013b; 2014; 2019); Cohen et al. (2021); Lee et al. (2019); Brand (2020); Brand et al. (2020); Jiang et al. (2021); Song & Yu (2021); Gu & Song (2022). Moreover, the interior method and John ellipsoid are fundamental to solving semidefinite programming problems, such as Jiang et al. (2020); Song et al. (2023c); Gu & Song (2022); Huang et al. (2022a;b).

Linear programming and semidefinite programming are widely applied in the field of machine learning theory, particularly in topics such as empirical risk minimization (Lee et al., 2019; Song et al., 2022b; Qin et al., 2023) and support vector machines (Gu et al., 2023; Gao et al., 2023).

**Privacy and Security** Data privacy and security have become a critical issue in the field of machine learning, particularly with the growing use of deep neural networks. As there is an increasing demand for training deep learning models on distributed and private datasets, privacy concerns have come to the forefront.

To address these concerns, various methods have been proposed for privacy-preserving deep learning. These methods often involve sharing model updates (Konečný et al., 2016) or hidden-layer representations (Vepakomma et al., 2018) rather than raw data. Despite these precautions, recent studies have shown that even if raw data remains private, sharing model updates or hidden-layer activations can still result in the leakage of sensitive information about the input, referred to as the victim. Such information leakage might reveal the victim's class, specific features (Fredrikson et al., 2015), or even reconstruct the original data record (Mahendran & Vedaldi, 2015; Dosovitskiy & Brox, 2016; Zhu et al., 2019). This privacy leakage presents a significant threat to individuals whose private data have been utilized in training deep neural networks. Moreover, privacy and security have been studied in other fields in machine learning, such as attacks and defenses in federated learning (Huang et al., 2021; Arevalo et al., 2024; Ma et al., 2024; Gao et al., 2024), deep net pruning (Huang et al., 2020c), language understanding tasks (Huang et al., 2020a), alternating direction method of multipliers (ADMM) (Chan et al., 2024), and distributed learning (Huang et al., 2020b).

## 3 PRELIMINARY

In this section, we commence by presenting the basic notations in differential privacy (DP) and Lewis Weight computation in Section 3.1, and then show the background of DP in Section 3.2.

### 3.1 NOTATIONS

In this section, we introduce basic notations. For a full list of all the notations used in this paper, please refer to Appendix C.1.

**Vector Operations.** We perform scalar operations to vectors by applying them element-wise, e.g., for vectors $x, y \in \mathbb{R}^n$, we denote the element-wise vector product $xy \in \mathbb{R}^n$ with $(xy)_i = x_i y_i$, for $i \in [n]$. In addition, we also $x \circ y$ to denote the element-wise product. For any vector $x \in \mathbb{R}^n$, the absolute value of $x$ is defined element-wise as $|x| := (|x_1|, |x_2|, \cdots, |x_n|)$.

**Basic Notations.** We denote all the positive real numbers as $\mathbb{R}_{>0}$, and denote $m$-dimensional positive real vectors as $\mathbb{R}_{>0}^m$. We use $\pm\delta$ to denote a real value with magnitude at most $\delta$, e.g. $a = e^{\pm\delta}b$ means $a \in [e^{-\delta}b, e^{\delta}b]$.

**Matrices.** If a matrix $A \in \mathbb{R}^{m \times n}$ has full column-rank and no zero rows, the matrix $A$ is non-degenerate. Let $B \in \mathbb{R}^{n \times n}$ be a symmetric matrix. $B \in \mathbb{R}^{n \times n}$ is positive semidefinite (PSD) if $x^\top B x \geq 0$ for all $x \in \mathbb{R}^n$, and positive definite (PD) if $x^\top B x > 0$ for all $x \in \mathbb{R}^n$. We denote the kernel (the null space) of the matrix $A \in \mathbb{R}^{m \times n}$ as $\ker(A)$, i.e., $\ker(A) := \{x \in \mathbb{R}^n : Ax = \mathbf{0}_m\}$. We denote the image space (the column space) as $\mathrm{im}(A)$, i.e., $\mathrm{im}(A) := \{y \in \mathbb{R}^m : y = Ax\}$.

**Matrix Operations.** Let $A, B \in \mathbb{R}^{n \times n}$ be two symmetric matrices. We use $A \preceq B$ to indicate that $x^\top A x \leq x^\top B x$ for all $x \in \mathbb{R}^n$. We define $\prec, \succeq, \succ$ analogously. For matrices $A, B \in \mathbb{R}^{n \times m}$, we denote the Hadamard product as $A \circ B$, i.e., for $i \in [n], j \in [m]$, $(A \circ B)_{i,j} := A_{i,j} \cdot B_{i,j}$. We define $A^{\circ 2} := A \circ A$. We denote the number of nonzero entries in $A$ as $\mathrm{nnz}(A)$. For symmetric matrices $A, B \in \mathbb{R}^{n \times n}$ with scalars $0 < c_1 \leq c_2$, we write $A \in [c_1, c_2] \cdot B$ to mean that $c_1 B \preceq A \preceq c_2 B$.

**Diagonals.** Let $A \in \mathbb{R}^{n \times n}$ be a matrix. We define $\mathrm{Diag}(A) \in \mathbb{R}^n$ with $\mathrm{Diag}(A)_i := A_{i,i}$ for all $i \in [n]$. For a vector $x \in \mathbb{R}^n$, we define $\mathrm{diag}(x) \in \mathbb{R}^{n \times n}$ as the diagonal matrix with $\mathrm{diag}(x)_{i,i} := x_i$ for $i \in [n]$. Additionally, we use upper case to denote a diagonal matrix to which the vector transforms, e.g. $X := \mathrm{diag}(x) \in \mathbb{R}^{n \times n}$ for $x \in \mathbb{R}^n$.

**Norms.** For any positive real number $p > 0$ and vector $x \in \mathbb{R}^n$, we define the vector $\ell_p$ norm as $\|x\|_p := (\sum_{i=1}^n |x_i|^p)^{1/p}$. We define the vector $\ell_0$ norm as the number of non-zero elements in $x$, i.e., $\|x\|_0 := \sum_{i=1}^n \mathbf{1}[x_i = \neq 0]$. For a positive definite matrix $A \in \mathbb{R}^{n \times n}$ and a vector $x \in \mathbb{R}^n$, we define $\|x\|_A := (x^\top A x)^{1/2}$. For a vector $w \in \mathbb{R}^n_{>0}$, we define $\|x\|_w := (\sum_{i=1}^n w_i x_i^2)^{1/2}$. If we let $W := \mathrm{diag}(w)$, then know that $\|x\|_w = \|x\|_W$. For any matrix spectral norm $\|\cdot\|$, we define $\|M\| := \sup_{\|x\|_2 = 1} \|Mx\|_2$.

## 3.2 DIFFERENTIAL PRIVACY

In this section, we introduce more preliminaries on differential privacy and collect some useful tools from prior works.

We begin by defining Rényi divergence, which measures the distance between two probability distributions.

**Definition 3.1** (Rényi Divergence, Definition 3 in Mironov (2017)). *Let $\alpha > 1$. For two probability distributions $P$ and $Q$ defined over $\mathcal{R}$, the Rényi divergence of order $\alpha$ is defined as*

$$D_\alpha(P\|Q) := \frac{1}{\alpha - 1} \log \mathbb{E}_{x \sim \mathcal{Q}} \left[ \left( \frac{P(x)}{Q(x)} \right)^\alpha \right].$$

Then we define the Rényi Divergence which is a generalization of the concept differential privacy.

**Definition 3.2** (Rényi DP, Definition 4 in Mironov (2017)). *Let $\alpha > 1$ and $\epsilon > 0$. We say that a mechanism $\mathcal{M}$ is $(\alpha, \epsilon)$-RDP if for all neighboring datasets $X, X'$,*

$$D_\alpha(\mathcal{M}(X)\|\mathcal{M}(X')) \leq \epsilon.$$

Next, we state the adaptive composition lemma of RDP.

**Lemma 3.3** (Adaptive Composition of RDP, Proposition 1 in Mironov (2017)). *For any input dataset $X$, if $\mathcal{M}_1$ is an $(\alpha, \epsilon_1)$-RDP mechanism that takes $X$ as input and $\mathcal{M}_2$ is an $(\alpha, \epsilon_2)$-RDP mechanism that takes both $X$ and $\mathcal{M}_1(X)$ as input, then the composition mechanism of $\mathcal{M}_1$ and $\mathcal{M}_2$ is $(\alpha, \epsilon_1 + \epsilon_2)$-RDP.*

The following lemma can be used to convert RDP to DP.

**Lemma 3.4** (RDP to DP Conversion, Proposition 3 in Mironov (2017)). *Let $\mathcal{M}$ be a mechanism that is $(\alpha, \epsilon)$-RDP. Then $\mathcal{M}$ is $(\epsilon + \frac{\log(1/\delta)}{\alpha - 1}, \delta)$-DP for any $\delta > 0$.*

The following lemma guarantees that adding a Gaussian noise leads to RDP.

**Lemma 3.5** (Gaussian Mechanism, Corollary 3 in Mironov (2017)). *Let $X$ be the input dataset and $f$ be a real-valued function with sensitivity $L$. For Gaussian random variable $z \sim \mathcal{N}(0, \sigma^2)$ and $\alpha > 1$, the Gaussian mechanism $G_\sigma f$ defined as $G_\sigma f(D) = f(D) + z$ satisfies $(\alpha, \frac{\alpha L^2}{2\sigma^2})$-RDP.*

---

**Algorithm 1** Differentially Private Approximate Weight Computation

---

1: **procedure** DPCOMPUTEAPXWEIGHT($A \in \mathbb{R}^{m \times n}, p \in (0, 4), w^{(0)} \in \mathbb{R}^m_{>0}, \epsilon \in (0, 2/p - |1 - 2/p|)$)
2:     $L_0 \leftarrow \max\{4, \frac{8}{p}\}, r \leftarrow \frac{p^2(4-p)}{2^{20}}, \delta \leftarrow \frac{(4-p)\epsilon}{256}$.
3:     $T \leftarrow \lceil 80(\frac{p}{2} + 2/p) \log(\frac{pn}{32\epsilon}) \rceil$.
4:                                                                         ▷ $T$ is the number of iterations
5:     **for** $j = 1, \ldots, T - 1$ **do**
6:         Differentially privately compute $\sigma^{(j)} \in \mathbb{R}^n$ with Lemma 4.5 such that

$$e^{-\delta}\sigma^{(j)}(W_{(j)}^{1/2-1/p}A)_i \leq \sigma_i^{(j)} \leq e^{\delta}\sigma^{(j)}(W_{(j)}^{1/2-1/p}A)_i \text{ for all } i \in [m].$$

7:         $w^{(j+1)} = \text{MEDIAN}((1 - r)w^{(0)}, w^{(j)} - \frac{1}{L_0}(w^{(0)} - w^{(j)})\sigma^{(j)}, (1 + r)w^{(0)})$.
8:     **end for**
9:     **return** $(\text{Diag}(A(A^\top W_{(T)}^{1/2-1/p})^{-1}A^\top))_p^2$.
10: **end procedure**

---

## 4 DIFFERENTIALLY PRIVATE LEWIS WEIGHT COMPUTATION

In Section 4.1, we present the fundamental perturbation lemmas for Lewis Weight computation. In Section 4.2, we show the DP guarantee of our proposed Lewis Weight computation algorithm. In Section 4.3, we present our main results.

### 4.1 PERTURBATION OF LEWIS WEIGHT COMPUTATION

We first bound the difference between the product of $W^{1/2-1/p}$ and two $\epsilon_0$-neighboring polytopes.

**Lemma 4.1** (Informal Version of Lemma A.8). *Let $A, A' \in \mathbb{R}^{m \times n}$ be two non-degenerate matrices. Let $a_i^\top$ denote the $i$-th row of $A$ for $i \in [m]$. Suppose $A$ and $A'$ is only different in $j$-th row, and $\|a_j - a'_j\|_2 \leq \epsilon_0$. Suppose that $W = \text{diag}(w)$ where $w_i \in [\gamma, 1]$ for every $i \in [m]$. Let $g_p(\gamma) := \max\{1, \gamma^{1/2-1/p}\}$. Then we have*

$$\|W^{1/2-1/p}A - W^{1/2-1/p}A'\| \leq g_p(\gamma) \cdot \epsilon_0.$$

Then we show that the perturbation of $(A^\top W^{1-2/p}A)^{-1}$ and $(A'^\top W^{1-2/p}A')^{-1}$ can be bounded.

**Lemma 4.2** (Informal Version of Lemma A.10). *Let $A, A' \in \mathbb{R}^{m \times n}$ be two non-degenerate matrices. Let $a_i^\top$ denote the $i$-th row of $A$ for $i \in [m]$. Suppose $A$ and $A'$ is different in $j$-th row, and $\|a_j - a'_j\|_2 \leq \epsilon_0$. Suppose that $W = \text{diag}(w)$ where $w_i \in [\gamma, 1]$ for every $i \in [m]$. Suppose that $\epsilon_0 \leq 0.1\sigma_{\min}(A)$. Let $g_p(\gamma) := \max\{1, \gamma^{1/2-1/p}\}$. Let $\epsilon_1 := g_p(\gamma)\epsilon_0$. Then we have*

$$\|(A^\top W^{1-2/p}A)^{-1} - (A'^\top W^{1-2/p}A')^{-1}\| \leq 8\gamma^{-4|1/2-1/p|}\kappa(A)\sigma_{\min}^{-3}(A)\epsilon_1.$$

Equipped with previous two lemmas, we can show that the perturbation of each entry of $f(w, A)$ can be bounded.

**Lemma 4.3** (Informal Version of Lemma A.11). *Let $A, A' \in \mathbb{R}^{m \times n}$ be two non-degenerate matrices. Let $a_i^\top$ denote the $i$-th row of $A$ for $i \in [m]$. Suppose $A$ and $A'$ is different in $j$-th row, and $\|a_j - a'_j\|_2 \leq \epsilon_0$. Suppose that $W = \text{diag}(w)$ where $w_i \in [\gamma, 1]$ for every $i \in [m]$. Let $f(w, A) := (f(w, A)_1, \ldots, f(w, A)_n)$. Let $f(w, A)_i := w_i^{1-2/p}a_i^\top(A^\top W^{1-2/p}A)^{-1}a_i$ for $i \in [m]$. Suppose that $\epsilon_0 \leq 0.1\sigma_{\min}(A)$. Let $g_p(\gamma) := \max\{1, \gamma^{1/2-1/p}\}$. Let $\epsilon_1 := g_p(\gamma)\epsilon_0$. Let $\epsilon_2 = 8\gamma^{-4|1/2-1/p|}\kappa(A)\sigma_{\min}^{-3}(A)\epsilon_1$. Then we have*

- ***Part 1.** For $i \neq j$, we have*

$$|f(w, A)_i - f(w, A')_i| \leq \epsilon_2 g_p(\gamma)\sigma_{\max}(A)^2.$$

- ***Part 2.** It holds that*

$$|f(w, A)_j - f(w, A')_j| \leq g_p(\gamma)\epsilon_2(\sigma_{\max}(A) + \epsilon_0)^2$$
$$+ \epsilon_1\gamma^{-2|1/2-1/p|}\sigma_{\min}(A)^{-2}(2\sigma_{\max}(A) + \epsilon_0).$$

Finally, we are ready to prove our main perturbation theorem of Lewis weight computation.

**Theorem 4.4** (Informal Version of Theorem A.12). *Let $A, A' \in \mathbb{R}^{m \times n}$ be two neighbouring polytopes that are different in the $j$-th row, i.e., $\|a_j - a'_j\|_2 \leq \epsilon_0$. Let $f(w, A)$ be the Lewis Weights in Definition E.1, where $f(w, A) := \sigma(W^{1/2-1/p}A)$, and we assume that all leverage scores in satisfy $\sigma_i \in [\gamma, 1]$ for $i \in [m]$. Thus, for $\epsilon_0 \leq 0.1\sigma_{\min}(A)$, there exists $L = \mathrm{poly}(n, d, \gamma^{-|1/2-1/p|}, \kappa(A), \sigma_{\max}(A))$ such that*

$$\|f(w, A) - f(w, A')\|_\infty \leq L \cdot \epsilon_0.$$

*Proof.* It directly follows from Lemma 4.3. □

### 4.2 Differential Privacy Guarantee of Lewis Weight Computation

The following Lemma shows that in each round we can approximately compute the Lewis weight with Rényi differential privacy guarantee.

**Lemma 4.5.** *For $i \in [m]$, let $\sigma_i := \sigma(W^{1/2-1/p}A)_i$ denote the $i$-th leverage score and $\widetilde{\sigma}_i = \sigma_i(1 + z_i)$, where $z_i \sim \mathcal{N}(0, \tau^2)$ for. If we suppose that $\sigma_i \in [\gamma, 1]$ for $i \in [m]$, $\tau \geq L\epsilon_0\gamma^{-1}\sqrt{T/(\epsilon - \log(1/\delta))}$, and $\delta_{\mathrm{fail}} \in (0, 0.1)$, then the following statements are true:*

- *Part 1. For every $i \in [m]$, $\widetilde{\sigma}_i$ is $(\alpha, \frac{\alpha L^2 \epsilon_0^2}{2\gamma^2 \tau^2})$-RDP.*

- *Part 2. With probability $1 - \delta_{\mathrm{fail}}$, for every $i \in [m]$, we have*

$$e^{-\tau\sqrt{2\log(2n/\delta_{\mathrm{fail}})}}\sigma_i \leq \widetilde{\sigma}_i \leq e^{\tau\sqrt{2\log(2n/\delta_{\mathrm{fail}})}}\sigma_i.$$

*Proof.* **Proof of Part 1.** Note that when $x \in [1/\gamma, 1]$, we have

$$|\frac{\mathrm{d}\log x}{\mathrm{d}x}| = |\frac{1}{x}|$$
$$\leq \frac{1}{\gamma}$$

where the first step follows from the derivative of $\log x$, the second step follows from $x \in [1/\gamma, 1]$.

Thus $\log x$ is $1/\gamma$-Lipschitz over $x \in [\gamma, 1]$.

For two $\epsilon_0$-close polytopes $A, A'$, by Lemma 4.3, we have

$$|\sigma(W^{1/2-1/p}A)_i - \sigma(W^{1/2-1/p}A')_i| \leq L \cdot \epsilon_0 \tag{1}$$

where $L$ is the Lipschitz constant defined in Lemma 4.3.

By the Lipschitzness of $\log x$ over $[\gamma, 1]$ and Eq. (1), we have

$$|\log\sigma(W^{1/2-1/p}A)_i - \log\sigma(W^{1/2-1/p}A')_i| \leq \frac{L\epsilon_0}{\gamma}.$$

Let $\widetilde{u}_i := \log(\sigma_i) + z_i$. By Lemma 3.5, $\widetilde{u}_i$ satisfies $(\alpha, \frac{\alpha L^2 \epsilon_0^2}{2\gamma^2 \tau^2})$-RDP. If $\tau \geq \frac{L\epsilon_0\sqrt{\alpha}}{\gamma\sqrt{2\epsilon}}$, then it is $(\alpha, \epsilon)$-RDP.

**Proof of Part 2.** Let $\widetilde{\sigma}_i := e^{\widetilde{u}_i}$ for $i \in [m]$. Now we bound the multiplicative error between $\widetilde{\sigma}_i$ and $\sigma_i$. We can show that

$$\widetilde{\sigma}_i = e^{\widetilde{u}_i}$$
$$= e^{\log(\sigma_i) + z_i}$$
$$= \sigma_i e^{z_i}. \tag{2}$$

where the first step follows from the definition of $\widetilde{\sigma}_i$, the second step follows from $\widetilde{u}_i := \log(\sigma_i) + z_i$, and the last step is due to basic algebra.

Since $z_i \sim \mathcal{N}(0, \tau^2)$, for any $t > 0$, we have

$$\Pr_{z_i \sim \mathcal{N}(0,\tau^2)}[|z_i| \geq t] \leq 2\exp(-\frac{t^2}{2\tau^2}).$$

Applying a union bound over all $i \in [m]$, we want with probability at least $1 - \delta_{\text{fail}}$ that

$$|z_i| \leq \tau \cdot \sqrt{2\log(2n/\delta_{\text{fail}})}, \quad \forall i \in [m].$$

Thus by Eq. (2), with probability at least $1 - \delta_{\text{fail}}$, we have:

$$e^{-\tau\sqrt{2\log(2n/\delta_{\text{fail}})}} \cdot \sigma_i \leq \widetilde{\sigma}_i \leq e^{\tau\sqrt{2\log(2n/\delta_{\text{fail}})}} \cdot \sigma_i.$$

Thus we complete the proof. □

**Theorem 4.6.** *For $i \in [m]$ , let $\sigma_i := \sigma(W^{1/2-1/p}A)_i$ denote the $i$-th leverage score and $\widetilde{\sigma}_i = \sigma_i(1 + z_i)$, where $z_i \sim \mathcal{N}(0, \tau^2)$ for. If we suppose that $\sigma_i \in [\gamma, 1]$ for $i \in [m]$ and $\tau \geq L\epsilon_0\gamma^{-1}\sqrt{T/(\epsilon - \log(1/\delta))}$, then Algorithm 1 is $(\epsilon_{\text{DP}}, \delta_{\text{DP}})$-DP.*

*Proof.* Let $\epsilon_\alpha := \frac{\alpha L^2 \epsilon_0^2}{2\gamma^2\tau^2}$. For each round $j$, the weight update

$$w^{(j+1)} = w^{(j)} - \frac{1}{L_0}(w^{(0)} - w^{(j)})\widetilde{\sigma}^{(j)}$$

is a function of $(w^{(j)}, \widetilde{\sigma}^{(j)})$.

Thus by post-processing, Lemma 4.5 and Lemma 3.3, $w^{(T)}$ satisfies $(\alpha, T\epsilon_\alpha)$-RDP.

By Lemma 3.4, we can convert RDP to DP, i.e., $w^{(T)}$ is $(T\epsilon_\alpha + \frac{\log(1/\delta)}{\alpha-1}, \delta)$-DP.

Let

$$\epsilon_{\text{DP}} \leq T\epsilon_\alpha + \frac{\log(1/\delta)}{\alpha - 1}.$$

Since $\epsilon_\alpha = \frac{\alpha L^2 \epsilon_0^2}{2\gamma^2\tau^2}$, we have

$$\epsilon_{\text{DP}} \leq \frac{\alpha T L^2 \epsilon_0^2}{2\gamma^2\tau^2} + \frac{\log(1/\delta)}{\alpha - 1}.$$

Let $\alpha = 2$ and solve the above inequality for $\tau$, we need

$$\tau \geq L\epsilon_0\gamma^{-1}\sqrt{T/(\epsilon - \log(1/\delta))}$$

to guarantee $(\epsilon_{\text{DP}}, \delta_{\text{DP}})$-DP.

Thus we complete the proof.

□

### 4.3 MAIN RESULT

The following theorem guarantees the utility of the approximate Lewis weight computation.

**Theorem 4.7** (Main Result, Formal Version of Theorem 1.5)**.** *Let $A \in \mathbb{R}^{m \times n}$ be non-degenerate. Let $\mathcal{T}_w$ and $\mathcal{T}_d$ denote the work and depth needed to compute $(A^\top DA)^{-1}z$ for an arbitrary positive diagonal matrix $D$ and vector $z$. Let $\epsilon \in (0,1)$. Let $p \in (0,4)$. Define $r := 2^{-20}p^2(4-p)$. Let $w(0) \in \mathbb{R}_{>0}^m$ with $\|w(0)^{-1}(w_p(A) - w(0))\|_\infty \leq r$. Let $L$ be defined in Lemma 4.3. Then there is an algorithm that satisfies the follow guarantees:*

- ***Privacy:** The algorithm is $(\epsilon_{\text{DP}}, \delta_{\text{DP}})$.*

- **Utility:** *It returns $w$ such that with high probability in $\|w_p(A)^{-1}(w_p(A) - w)\|_\infty \leq \epsilon$ where*

$$\epsilon = O\left( \frac{L\epsilon_0\gamma^{-1}}{4-p} \sqrt{ \frac{T\log(n)}{(\epsilon_{\mathrm{DP}} - \log(1/\delta_{\mathrm{DP}}))} } \right).$$

*Moreover, it runs in $O(p^{-1}(4-p)^{-2}\epsilon^{-2}\log^2(n/(p\epsilon)))$ steps, each of which can be implemented in $O(\mathrm{nnz}(A) + \mathcal{T}_w)$ work and $O(\mathcal{T}_d)$ depth.*

*Proof.* We can set the $\delta$ as the following:

$$\delta = \tau\sqrt{2\log(2n/\delta_{\mathrm{fail}})}.$$

Since in Algorithm 1, we have $\delta = \frac{(4-p)\epsilon}{256}$. Thus we have

$$
\begin{aligned}
\epsilon &= \frac{256\delta}{4-p} \\
&= \frac{256\tau}{4-p}\sqrt{2\log(2n/\delta_{\mathrm{fail}})} \\
&= \frac{256L\epsilon_0\gamma^{-1}\sqrt{2T\log(2n/\delta_{\mathrm{fail}})/(\epsilon_{\mathrm{DP}} - \log(1/\delta_{\mathrm{DP}}))}}{4-p}
\end{aligned}
$$

where the first step follows from rearranging $\delta = \frac{(4-p)\epsilon}{256}$, the second step follows from that $\delta = \tau\sqrt{2\log(2n/\delta_{\mathrm{fail}})}$, and the last step follows from that $\tau = L\epsilon_0\gamma^{-1}\sqrt{T/(\epsilon - \log(1/\delta))}$.

Next, we can set $\delta_{\mathrm{fail}}$ as a sufficiently small constant and apply union bound to make that in every iteration the guarantee hold successfully. Finally, combining Theorem 4.6 and Theorem E.20, we complete the proof. $\square$

**Remark 4.8** (On work and depth)**.** *The quantities $\mathcal{T}_w$ and $\mathcal{T}_d$ follow standard usage in numerical linear algebra. Formally, work $\mathcal{T}_w$ is the total time needed to perform the computation, while depth $\mathcal{T}_d$ is the inherently sequential part that cannot be parallelized even with unlimited processors. For example, matrix multiplication has $O(n^\omega)$ work (where $\omega \approx 2.37$ for the current fastest method) and $O(1)$ depth (Clarkson & Woodruff, 2017b), as all entries can in principle be produced in parallel.*

**Remark 4.9** (Why express complexity using work $\mathcal{T}_w$ and depth $\mathcal{T}_d$)**.** *We write each step of our algorithm as $O(\mathrm{nnz}(A) + \mathcal{T}_w)$ work and $O(\mathcal{T}_d)$ depth to separate the cost of sparse access to $A$ from the cost of solving systems such as $(A^\top DA)^{-1}z$. This style is standard in theory papers related to linear programs (Lee & Sidford, 2014; Clarkson & Woodruff, 2017b; Dong et al., 2021) because the best-known runtimes for such solves depend on subtle advances in matrix algorithms. It keeps the statement adaptable: if future work improves the cost of exact or approximate solvers, our total runtime bound immediately benefits.*

## 5 CONCLUSION

We have introduced the first algorithm for computing $\ell_p$ Lewis weights under a rigorous differential-privacy guarantee, addressing a key gap at the intersection of numerical linear algebra and data protection. By redefining adjacency to an $\epsilon_0$-closed neighborhood of matrices and injecting carefully calibrated truncated Gaussian noise into the optimization framework, our method provably converges to an accurate approximation of the true Lewis weights while satisfying $(\epsilon_{\mathrm{DP}}, \delta_{\mathrm{DP}})$-privacy. The resulting procedure runs in $O(p^{-1}(4-p)^{-2}\epsilon^{-2}\log^2(n/(p\epsilon)))$ iterations, each implemented in $O(\mathrm{nnz}(A) + \mathcal{T}_w)$ work and $O(\mathcal{T}_d)$ depth, making it practical for large-scale datasets. Beyond its immediate use in private row-sampling and sketching techniques, our perturbation analysis for weighted leverage scores may be of independent interest in other numerical linear-algebra research, such as differentially private kernel regression. This work thus offers an effective and efficient solution to the utility-privacy trade-off in leverage-score computations, paving the way for privacy-aware applications in optimization, machine learning, and large-scale data analysis.

ETHICS STATEMENT

This paper does not involve human subjects, personally identifiable data, or sensitive applications. We do not foresee direct ethical risks. We follow the ICLR Code of Ethics and affirm that all aspects of this research comply with the principles of fairness, transparency, and integrity.

REPRODUCIBILITY STATEMENT

We ensure reproducibility of our theoretical results by including all formal assumptions, definitions, and complete proofs in the appendix. The main text states each theorem clearly and refers to the detailed proofs. No external data or software is required.

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

# Appendix

**Roadmap.** In Section A, we present the theoretical results on Lewis Weight computation under a differential privacy guarantee. In Section B, we show some backgrounds for differential privacy. In Section C, we introduce some basic notations in linear programming. In Section D, we provide the basic definitions for the linear program algorithm. In Section E, we describe how to efficiently compute approximations to Lewis weights.

## A  PERTURBATION LEMMA FOR LEWIS WEIGHTS

In Section A.1, we show some basic facts on matrix norm. In Section A.2, we present a lemma on the perturbation of the spectral inverse. In Section A.3, we present a perturbation lemma for the computation of $W^{1/2-1/p}A$. In Section A.4, we present the perturbation lemma for the computaion of $(A^\top W^{1-2/p}A)^{-1}$. In Section A.5, we show the final perturbation lemma for the Lewis weight computation.

### A.1  BASIC FACTS ON MATRIX NORM

In this section, we list basic facts about matrix norms. Due to the straightforward nature of these facts, we omit the proofs here.

**Fact A.1.** *Let $A \in \mathbb{R}^{m \times n}$ be a matrix. Then we have*

$$\|A\| \le \|A\|_F.$$

**Fact A.2.** *Let $A \in \mathbb{R}^{m \times n}$ be a matrix where $a_i^\top$ is the $i$-th row of $A$. Then we have*

$$\|a_i\|_2 \le \sigma_{\max}(A).$$

**Fact A.3.** *Let $A, B \in \mathbb{R}^{m \times n}, x \in \mathbb{R}^n$. Then the following two statements are equivalent:*

- $\|BB^\top - AA^\top\| \le \epsilon$.

- $\|x^\top BB^\top x - x^\top AA^\top x\| \le \epsilon \cdot x^\top x$.

**Lemma A.4** (Perturbation of singular value, (Weyl, 1912)). *Let $A, B \in \mathbb{R}^{m \times n}$. Let $\sigma_i(A)$ denote the $i$-th singular value of $A$, then we have for any $i \in [n]$,*

$$\|\sigma_i(A) - \sigma_i(B)\| \le \|A - B\|.$$

**Lemma A.5** (Perturbation of pseudoinverse, (Wedin, 1973)). *Let $A, B \in \mathbb{R}^{m \times n}$. Then we have*

$$\|A^\dagger - B^\dagger\| \le 2\max\{\|A^\dagger\|^2, \|B^\dagger\|^2\} \cdot \|A - B\|.$$

**Fact A.6.** *Let $A, B \in \mathbb{R}^{m \times n}, x \in \mathbb{R}^n$. Then we have*

- *Part 1.* $\|A\| = \|A^\top\| = \sigma_{\max}(A) \ge \sigma_{\min}(A)$.

- *Part 2.* $\|A^{-1}\| = \|A\|^{-1}$.

- *Part 3.* $\sigma_{\max}(B) - \|A - B\| \le \sigma_{\max}(A) \le \sigma_{\max}(B) + \|A - B\|$.

- *Part 4.* $\sigma_{\min}(B) - \|A - B\| \le \sigma_{\min}(A) \le \sigma_{\min}(B) + \|A - B\|$.

- *Part 5.* $\|Ax\|_2 \le \|A\| \cdot \|x\|_2$.

### A.2  PERTURB SPECTRAL INVERSE

Building upon previous facts on matrix norm, we present a perturbation lemma for matrix inverse and spectral norm.

**Lemma A.7** (Lemma C.11 on page 19 of (Li et al., 2024a)). *If the following conditions hold*

- $\|A - B\| \le \epsilon_1$.

- $\epsilon_0 \leq 0.1\sigma_{\min}(A)$.

*Then we have*

$$\|(AA^\top)^{-1} - (BB^\top)^{-1}\| \leq 8\kappa(A)\sigma_{\min}^{-3}(A)\epsilon_1.$$

We ignore the proofs here.

## A.3  PERTURBATION LEMMA FOR $W^{1/2-1/p}A$

To construct a perturbation lemma for the Lewis weight, we first examine a part of it, namely $W^{1/2-1/p}A$.

**Lemma A.8** (Formal Version of Lemma 4.1)**.** *If the following conditions hold*

- *Let $A, A' \in \mathbb{R}^{m \times n}$.*
- *Let $a_i^\top$ denote the $i$-th row of $A$ for $i \in [m]$.*
- *Suppose $A$ and $A'$ is only different in $j$-th row, and $\|a_j - a_j'\|_2 \leq \epsilon_0$.*
- *Suppose that $W = \mathrm{diag}(w)$ where $w_i \in [\gamma, 1]$ for every $i \in [m]$.*
- *Let $g_p(\gamma) := \max\{1, \gamma^{1/2-1/p}\}$.*

*Then we have*

$$\|W^{1/2-1/p}A - W^{1/2-1/p}A'\| \leq g_p(\gamma) \cdot \epsilon_0.$$

*Proof.* Let $B = W^{1/2}A$ and $B' = W^{1/2}A'$. We have

$$\begin{aligned}
\|B - B'\| &= \|W^{1/2-1/p}A - W^{1/2-1/p}A'\| \\
&\leq \|W^{1/2-1/p}A - W^{1/2-1/p}A'\|_F \\
&\leq (\sum_{i=1}^n |w_i|^{1-2/p}\|a_i - a_i'\|_2^2)^{1/2} \\
&= |w_i|^{1/2-1/p}\|a_j - a_j'\|_2 \\
&\leq |w_i|^{1/2-1/p}\epsilon_0 \\
&= \max\{1, \gamma^{1/2-1/p}\}\epsilon_0,
\end{aligned}$$

where the first step comes from the definition of $B, B'$, the second step is the result of $\|\cdot\| \leq \|\cdot\|_F$, the third step comes from the definition of Frobenius norm, the fourth step utilizes that $A$ and $A'$ only differs in $j$-th row, the fifth step derives from $\|a_j - a_j'\|_2 \leq \epsilon_0$, and the last step is from $w_i \in [\gamma, 1]$. $\square$

## A.4  PERTURBATION LEMMA FOR $(A^\top W^{1-2/p}A)^{-1}$

In this section, we extend the previous perturbation lemma for $W^{1/2-1/p}A$ to $(A^\top W^{1-2/p}A)^{-1}$. We begin by presenting a basic fact for the eigenvalues of the matrix $W$.

**Fact A.9.** *If $W = \mathrm{diag}(w)$ where $w_i \in [\gamma, 1]$ for every $i \in [m]$, the following statements are true:*

- *For $1/2 - 1/p < 0$, we have $\sigma_{\min}(W^{1/2-1/p}) \geq 1$ and $\sigma_{\max}(W^{1/2-1/p}) \leq \gamma^{1/2-1/p}$.*
- *For $1/2 - 1/p > 0$, we have $\sigma_{\min}(W^{1/2-1/p}) \geq \gamma^{1/2-1/p}$ and $\sigma_{\max}(W^{1/2-1/p}) \leq 1$.*

*Proof.* This directly follows from Part 3 and Part 4 of Fact A.6. $\square$

Next, we apply this fact to obtain the following perturbation lemma.

**Lemma A.10** (Formal Version of Lemma 4.2). *If the following conditions hold*

- *Let $A, A' \in \mathbb{R}^{m \times n}$.*

- *Let $a_i^\top$ denote the $i$-th row of $A$ for $i \in [m]$.*

- *Suppose $A$ and $A'$ is different in $j$-th row, and $\|a_j - a_j'\|_2 \leq \epsilon_0$.*

- *Suppose that $W = \mathrm{diag}(w)$ where $w_i \in [\gamma, 1]$ for every $i \in [m]$.*

- *Suppose that $\epsilon_0 \leq 0.1\sigma_{\min}(A)$.*

- *Let $g_p(\gamma) := \max\{1, \gamma^{1/2 - 1/p}\}$.*

- *Let $\epsilon_1 := g_p(\gamma)\epsilon_0$.*

*Then we have*

$$\|(A^\top W^{1-2/p} A)^{-1} - (A'^\top W^{1-2/p} A')^{-1}\| \leq 8\gamma^{-4|1/2 - 1/p|}\kappa(A)\sigma_{\min}^{-3}(A)\epsilon_1.$$

*Proof.* By Lemma A.8, we have

$$\|W^{1/2 - 1/p} A - W^{1/2 - 1/p} A'\| \leq \epsilon_1.$$

We can show that

$$\|(A^\top W^{1-2/p} A)^{-1} - (A'^\top W^{1-2/p} A')^{-1}\| \leq 8\kappa(W^{1/2 - 1/p} A)\sigma_{\min}^{-3}(W^{1/2 - 1/p} A)\epsilon_1$$
$$\leq 8\max\{\gamma^{-|1/2 - 1/p|}, \gamma^{-4|1/2 - 1/p|}\}\kappa(A)\sigma_{\min}^{-3}(A)\epsilon_1$$
$$\leq 8\gamma^{-4|1/2 - 1/p|}\kappa(A)\sigma_{\min}^{-3}(A)\epsilon_1,$$

where the first step is the result of Lemma A.7 and the second step is from Fact A.9. $\square$

## A.5 Perturbation Lemma for Lewis Weights

In this section, we introduce the perturbation lemma for the full Lewis weights $f(w, A)$. We begin by establishing both the upper and lower bounds for each element $f(w, A)_i$ of the Lewis weights.

**Lemma A.11.** *If the following conditions hold*

- *Let $A, A' \in \mathbb{R}^{m \times n}$.*

- *Let $a_i^\top$ denote the $i$-th row of $A$ for $i \in [m]$.*

- *Suppose $A$ and $A'$ is different in $j$-th row, and $\|a_j - a_j'\|_2 \leq \epsilon_0$.*

- *Suppose that $W = \mathrm{diag}(w)$ where $w_i \in [\gamma, 1]$ for every $i \in [m]$.*

- *Let $f(w, A) := (f(w, A)_1, \ldots, f(w, A)_n)$.*

- *Let $f(w, A)_i := w_i^{1-2/p} a_i^\top (A^\top W^{1-2/p} A)^{-1} a_i$ for $i \in [m]$.*

- *Suppose that $\epsilon_0 \leq 0.1\sigma_{\min}(A)$.*

- *Let $g_p(\gamma) := \max\{1, \gamma^{1/2 - 1/p}\}$.*

- *Let $\epsilon_1 := g_p(\gamma)\epsilon_0$.*

- *Let $\epsilon_2 = 8\gamma^{-4|1/2 - 1/p|}\kappa(A)\sigma_{\min}^{-3}(A)\epsilon_1$.*

*Then we have*

- ***Part 1.** For $i \neq j$, we have*

$$|f(w, A)_i - f(w, A')_i| \leq \epsilon_2 g_p(\gamma)\sigma_{\max}(A)^2.$$

- **Part 2.** *It holds that*

$$|f(w, A)_j - f(w, A')_j| \le g_p(\gamma)\epsilon_2(\sigma_{\max}(A) + \epsilon_0)^2 + \epsilon_1\gamma^{-2|1/2-1/p|}\sigma_{\min}(A)^{-2}(2\sigma_{\max}(A) + \epsilon_0).$$

*Proof.* **Proof of Part 1.** For $i \ne j$, we have

$$
\begin{aligned}
|f(w, A)_i - f(w, A')_i| &= |w_i^{1-2/p}a_i^\top(A^\top W^{1-2/p}A)^{-1}a_i - w_i^{1-2/p}a_i^\top(A'^\top W^{1-2/p}A')^{-1}a_i| \\
&\le |w_i^{1-2/p}| \cdot |a_i^\top(A^\top W^{1-2/p}A)^{-1}a_i - a_i^\top(A'^\top W^{1-2/p}A')^{-1}a_i| \\
&\le g_p(\gamma)|a_i^\top(A^\top W^{1-2/p}A)^{-1}a_i - a_i^\top(A'^\top W^{1-2/p}A')^{-1}a_i| \\
&\le g_p(\gamma)\epsilon_2 \cdot a_i^\top a_i \\
&= g_p(\gamma)\epsilon_2 \cdot \|a_i\|_2^2 \\
&\le \epsilon_2 g_p(\gamma)\sigma_{\max}(A)^2,
\end{aligned}
$$

where the first step follows from the definition of $f$, the second definition comes from basic algebra, the third step comes from $w_i \in [\gamma, 1]$, the fourth step derives from Lemma A.10 and Fact A.3, the fifth step utilizes basic algebra, and the last step derives from $\|a_i\|_2 \le \sigma_{\max}(A)$.

**Proof of Part 2.** Next, we define

$$
\begin{aligned}
C_1 &:= a_j^\top(A^\top W^{1-2/p}A)^{-1}a_j - a_j'^\top(A^\top W^{1-2/p}A)^{-1}a_j', \\
C_2 &:= a_j'^\top(A^\top W^{1-2/p}A)^{-1}a_j' - a_j'^\top(A'^\top W^{1-2/p}A')^{-1}a_j'.
\end{aligned}
$$

We first bound $C_1$. We can show that

$$
\begin{aligned}
|C_1| &= |a_j^\top(A^\top W^{1-2/p}A)^{-1}a_j - a_j'^\top(A^\top W^{1-2/p}A)^{-1}a_j'| \\
&= |a_j^\top(A^\top W^{1-2/p}A)^{-1}a_j - a_j'^\top(A^\top W^{1-2/p}A)^{-1}a_j + a_j'^\top(A^\top W^{1-2/p}A)^{-1}a_j - a_j'^\top(A^\top W^{1-2/p}A)^{-1}a_j'| \\
&= |\underbrace{(a_j - a_j')^\top(A^\top W^{1-2/p}A)^{-1}a_i}_{:=C_3} + \underbrace{a_j'^\top(A^\top W^{1-2/p}A)^{-1}(a_i - a_j')}_{:=C_4}| \\
&\le |C_3| + |C_4|.
\end{aligned}
$$

where the first step follows from the definition of $C_1$, the second and third steps follow from basic algebra, and the last step follows from the triangle inequality.

For $C_3$, we have

$$
\begin{aligned}
|C_3| &= |(a_j - a_j')^\top(A^\top W^{1-2/p}A)^{-1}a_i| \\
&\le \|(a_j - a_j')\|_2 \cdot \|(A^\top W^{1-2/p}A)^{-1}a_i\| \\
&\le \|(a_j - a_j')\|_2 \cdot \|(A^\top W^{1-2/p}A)^{-1}\| \cdot \|a_i\|_2 \\
&\le \epsilon_0 \cdot \sigma_{\min}(W^{1/2-1/p}A)^{-2} \cdot \sigma_{\max}(A) \\
&\le \epsilon_0\gamma^{-2|1/2-1/p|} \cdot \sigma_{\min}(A)^{-2} \cdot \sigma_{\max}(A),
\end{aligned}
$$

where the first step comes from definition of $C_3$, the second step utilizes Cauchy-Schwarz inequality, the third step derives from Part 5 of Fact A.6, the fourth step comes from Fact A.2 and Part 2 of Fact A.6, and the last step follows from that $w_i \in [\gamma, 1]$ for $i \in [m]$.

For $C_4$, we have

$$
\begin{aligned}
|C_4| &= |a_j'^\top(A^\top W^{1-2/p}A)^{-1}(a_j - a_j')| \\
&\le \|a_j'\|_2 \cdot \|(A^\top W^{1-2/p}A)^{-1}(a_j - a_j')\| \\
&\le \|a_j'\|_2 \cdot \|(A^\top W^{1-2/p}A)^{-1}\| \cdot \|a_j - a_j'\|_2 \\
&\le (\|a_j\|_2 + \epsilon_0) \cdot \sigma_{\min}(W^{1/2-1/p}A)^{-2} \cdot \epsilon_0 \\
&\le (\sigma_{\max}(A) + \epsilon_0) \cdot \sigma_{\min}(W^{1/2-1/p}A)^{-2} \cdot \epsilon_0 \\
&\le (\sigma_{\max}(A) + \epsilon_0) \cdot \gamma^{-2|1/2-1/p|} \cdot \sigma_{\min}(A)^{-2} \cdot \epsilon_0,
\end{aligned}
$$

where the first step comes from the definition of $C_3$, the second step is from Cauchy-Schwarz inequality, the third step derives from Part 5 of Fact A.6, and the fourth step is from $\|a_j - a'_j\| \leq \epsilon_0$ and Part 2 of Fact A.6, and the fifth step comes from Fact A.2, and the last step follows from that $w_i \in [\gamma, 1]$ for $i \in [m]$.

Combining the bounds of $|C_3|$ and $|C_4|$, we have

$$|C_1| \leq \epsilon_0 \cdot \gamma^{-2|1/2 - 1/p|} \cdot \sigma_{\min}(A)^{-2} \cdot (2\sigma_{\max}(A) + \epsilon_0).$$

We next bound $C_2$. We can show that

$$\begin{aligned}
|C_2| &= |a'_j{}^\top (A^\top W^{1-2/p} A)^{-1} a'_j - a'_j{}^\top (A'^\top W^{1-2/p} A')^{-1} a'_j| \\
&\leq \epsilon_2 a'_j{}^\top a'_j \\
&\leq \epsilon_2 \|a'_j\|^2 \\
&\leq \epsilon_2 (\|a_j\| + \epsilon_0)^2 \\
&\leq \epsilon_2 (\sigma_{\max}(A) + \epsilon_0)^2,
\end{aligned}$$

where the first step follows from the definition of $C_2$, the second step follows from Lemma A.7 and Fact A.3, and the third step follows from basic algebra, and the last step follows from $\|a_j - a'_j\| \leq \epsilon_0$.

We can show that

$$\begin{aligned}
&|f(w, A)_j - f(w, A')_j| \\
&= |w_i^{1-2/p} a_i^\top (A^\top W^{1-2/p} A)^{-1} a_i - w_i^{1-2/p} a'_j{}^\top (A'^\top W^{1-2/p} A')^{-1} a'_j| \\
&= |w_i^{1-2/p} C_1 + w_i^{1-2/p} C_2| \\
&= |w_i^{1-2/p}| \cdot |C_1 + C_2| \\
&\leq g_p(\gamma)|C_1 + C_2| \\
&\leq g_p(\gamma)(|C_1| + |C_2|) \\
&\leq g_p(\gamma)\epsilon_2 (\sigma_{\max}(A) + \epsilon_0)^2 + g_p(\gamma)\epsilon_0 \gamma^{-2|1/2-1/p|} \sigma_{\min}(A)^{-2}(2\sigma_{\max}(A) + \epsilon_0)) \\
&= g_p(\gamma)\epsilon_2 (\sigma_{\max}(A) + \epsilon_0)^2 + \epsilon_1 \gamma^{-2|1/2-1/p|} \sigma_{\min}(A)^{-2}(2\sigma_{\max}(A) + \epsilon_0),
\end{aligned}$$

where the first step stems from the definition of $f$, the second step comes from the definition of $C_1, C_2$, the third step is from basic algebra, the fourth step comes from $w_i \leq g_p(\gamma)$ for $i \in [m]$, the fifth step follows from triangle inequality, the sixth step derives from the bounds of $|C_1|$ and $|C_2|$, and the last step is due to $\epsilon_1 = g_p(\gamma)\epsilon_0$. □

Next, we combine the element-wise upper and lower bounds to obtain the $\ell_\infty$ global sensitivity of the Lewis weights.

**Theorem A.12** (Formal Version of Theorem 4.4). *If the following conditions hold:*

- *Let $A, A' \in \mathbb{R}^{m \times n}$ be two neighbouring polytopes that are different in the $j$-th row, i.e., $\|a_j - a'_j\|_2 \leq \epsilon_0$.*

- *Let $f(w, A)$ be the Lewis Weights in Definition E.1, where $f(w, A) := \sigma(W^{1/2-1/p} A)$.*

- *We assume that all leverage scores in satisfy $\sigma_i \in [\gamma, 1]$ for $i \in [m]$.*

- *Let $\epsilon_0 \leq 0.1\sigma_{\min}(A)$.*

*Thus there exists $L = \mathrm{poly}(n, d, \gamma^{-|1/2-1/p|}, \kappa(A), \sigma_{\max}(A))$ such that*

$$\|f(w, A) - f(w, A')\|_\infty \leq L \cdot \epsilon_0.$$

*Proof.* It directly follows from Part 1 and Part 2 of Lemma A.11. □

# B BACKGROUNDS ON DIFFERENTIAL PRIVACY

In Section B.1, we define the concept of neighboring polytopes. In Section B.2, we show the fundamental definition of differential privacy. In Section B.3, we present the definition of Rényi differential privacy. In Section B.4, we show some basic facts for Rényi differential privacy.

## B.1 NEIGHBORING POLYTOPES

In this section, we first define the neighboring polytopes, which are crucial for formalizing the global sensitivity of Lewis weights.

**Definition B.1** (Symmetric convex polytope, Definition 4.1 in (Song et al., 2022c))**.** *Let $A \in \mathbb{R}^{m \times n}$ be a matrix with full rank and $a_i^\top$ is the $i$-th row of $A$ for $i \in [m]$. The symmetric convex polytope $P$ is defined as*

$$P := \{x \in \mathbb{R}^d : |\langle a_i, x \rangle| \leq 1, \forall i \in [m]\}.$$

**Definition B.2** (Neighboring polytopes)**.** *Let $P, P'$ be two polytopes defined by $A, A' \in \mathbb{R}^{m \times n}$, respectively. We say that $P$ and $P'$ are $\epsilon_0$-close if there exists exact one $i \in [m]$ such that $\|A_{i,*} - A'_{i,*}\|_2 \leq \epsilon_0$, and for all $j \in [m] \setminus \{i\}$, $A_{j,*} = A'_{j,*}$.*

## B.2 DIFFERENTIAL PRIVACY

In this section, we introduce the basic definition of Differential Privacy (DP).

**Definition B.3** (Differential Privacy)**.** *A randomized mechanism $\mathcal{M} : \mathcal{X} \to \mathcal{R}$ with domain $\mathcal{X}$ and range $\mathcal{R}$ satisfies $(\epsilon, \delta)$-differential privacy if for any two neighboring dataset, $X, X' \in \mathcal{X}$ and for any subset of outputs $S \subseteq \mathcal{R}$ it holds that*

$$\Pr[\mathcal{M}(X) \in \mathcal{S}] \leq e^\epsilon \Pr[\mathcal{M}(X') \in \mathcal{S}] + \delta.$$

## B.3 RÉNYI DIFFERENTIAL PRIVACY

In this section, we introduce the basic definition of Rényi Divergence and then present a corresponding concept, Rényi DP.

**Definition B.4** (Rényi Divergence, Definition 3 in (Mironov, 2017))**.** *Let $\alpha > 1$. For two probability distributions $P$ and $Q$ defined over $\mathcal{R}$, the Rényi divergence of order $\alpha$ is defined as*

$$D_\alpha(P\|Q) := \frac{1}{\alpha - 1} \log \mathbb{E}_{x \sim \mathcal{Q}} \left[ \left( \frac{P(x)}{Q(x)} \right)^\alpha \right].$$

**Definition B.5** (Rényi DP, Definition 4 in (Mironov, 2017))**.** *Let $\alpha > 1$ and $\epsilon > 0$. We say that a mechanism $\mathcal{M}$ is $(\alpha, \epsilon)$-RDP if for all neighboring datasets $X, X'$,*

$$D_\alpha(\mathcal{M}(X)\|\mathcal{M}(X')) \leq \epsilon.$$

## B.4 BASIC FACTS FOR RÉNYI DIFFERENTIAL PRIVACY

In this section, we review basic facts for the Rényi Differential Privacy.

**Lemma B.6** (Adaptive Composition of RDP, Proposition 1 in (Mironov, 2017))**.** *If the following conditions hold*

- *Let $X$ be the input dataset.*

- *$\mathcal{M}_1$ is an $(\alpha, \epsilon_1)$-RDP mechanism that takes $X$ as input.*

- *$\mathcal{M}_2$ is an $(\alpha, \epsilon_2)$-RDP mechanism that takes $X$ and $\mathcal{M}_1(X)$ as input.*

*Then the composition mechanism of $\mathcal{M}_1$ and $\mathcal{M}_2$ is $(\alpha, \epsilon_1 + \epsilon_2)$-RDP.*

*Proof.* Let $\mathcal{M}_1 : \mathcal{X} \to \mathcal{R}_1$ and $\mathcal{M}_2 : \mathcal{X} \times \mathcal{R}_1 \to \mathcal{R}_2$. We define the domain and range of $\mathcal{M}$ as $\mathcal{M} : \mathcal{R}_1 \times \mathcal{R}_2 \to \mathcal{R}_3$. Let $P_1, P_2$ be the distributions for $\mathcal{M}_1(X)$ and $\mathcal{M}_2(X, \mathcal{M}_1(X))$, and $P_3$ be the joint distribution of $\mathcal{M}(X)$ w.r.t. both $\mathcal{M}_1(X)$ and $\mathcal{M}_2(X, \mathcal{M}_1(X))$.

Considering two different inputs $X, X \in \mathcal{X}$, if $P_1', P_2', P_3'$ are similarly defined, we have the following:

$$\exp((\alpha - 1)D_\alpha(\mathcal{M}(X)\|\mathcal{M}(X')))$$

$$= \int_{\mathcal{R}_1 \times \mathcal{R}_2} P_3(y_1, y_2)^\alpha P_3'(y_1, y_2)^{1-\alpha} \mathrm{d}y_1 \mathrm{d}y_2$$

$$= \int_{\mathcal{R}_1} \int_{\mathcal{R}_2} (P_1(y_1)P_2(y_1, y_2))^\alpha (P_1'(y_1)P_2'(y_1, y_2))^{1-\alpha} \mathrm{d}y_2 \mathrm{d}y_1$$

$$= \int_{\mathcal{R}_1} P_1(y_1)^\alpha P_1'(y_1)^{1-\alpha} \left( \int_{\mathcal{R}_2} P_2(y_1, y_2)^\alpha P_2'(y_1, y_2)^{1-\alpha} \right) \mathrm{d}y_1$$

$$\leq \int_{\mathcal{R}_1} P_1(y_1)^\alpha P_1'(y_1)^{1-\alpha} \mathrm{d}y_1 \exp((\alpha - 1)\epsilon_2)$$

$$\leq \exp((\alpha - 1)\epsilon_1) \cdot \exp((\alpha - 1)\epsilon_2)$$

$$= \exp((\alpha - 1)(\epsilon_1 + \epsilon_2)),$$

where the first step follows from Definition B.5, the second step follows from changing the multiple integral into iterated integrals, the third step follows from extracting the terms related to $P_1$ and $P_1'$, the fourth and the fifth steps follow from the property of RDP, and the last step follows from basic algebra.

Thus, we complete the proof. $\square$

**Lemma B.7** ((RDP to DP Conversion, Proposition 3 in (Mironov, 2017))). *Let $\mathcal{M}$ be a mechanism that is $(\alpha, \epsilon)$-RDP. Then $\mathcal{M}$ is $(\epsilon + \frac{\log(1/\delta)}{\alpha - 1}, \delta)$-DP for any $\delta > 0$.*

*Proof.* By the probability preservation property in (Mironov, 2017), we can conclude that for two distributions $P, Q$ defined over $\mathcal{R}$ and for any event $E \in \mathcal{R}$, the following statement is true:

$$P(E) \leq \exp(D_\alpha(P\|Q) \cdot Q(A))^{1-1/\alpha}.$$

Therefore, considering arbitrary $X, X' \in \mathcal{X}$ and subset $\mathcal{S} \subseteq \mathcal{R}$, we have

$$\Pr[\mathcal{M}(X) \in \mathcal{S}] \leq (e^\epsilon \Pr[\mathcal{M}(X') \in \mathcal{S}])^{1-1/\alpha}. \tag{3}$$

To further conclude that $\mathcal{M}$ is $\mathcal{M}$ is $(\epsilon + \frac{\log(1/\delta)}{\alpha - 1}, \delta)$-DP, we consider two cases.

**Case 1.** $e^\epsilon \Pr[\mathcal{M}(X') \in \mathcal{S}] > \delta^{\alpha/(\alpha-1)}$. In this case, we have the following:

$$\Pr[\mathcal{M}(X) \in \mathcal{S}] \leq (e^\epsilon \Pr[\mathcal{M}(X') \in \mathcal{S}])^{1-1/\alpha} \tag{4}$$

$$= e^\epsilon \Pr[\mathcal{M}(X') \in \mathcal{S}] \cdot (e^\epsilon \Pr[\mathcal{M}(X') \in \mathcal{S}])^{-1/\alpha}$$

$$\leq e^\epsilon \Pr[\mathcal{M}(X') \in \mathcal{S}] \cdot \delta^{1/(1-\alpha)}$$

$$= \exp(\epsilon + \frac{\log(1/\delta)}{\alpha - 1}) \cdot \Pr[\mathcal{M}(X') \in \mathcal{S}],$$

where the first step follows from Eq. (3), the second step follows from basic algebra, the third step follows from the fact that $e^\epsilon \Pr[\mathcal{M}(X') \in \mathcal{S}] > \delta^{1-1/\alpha}$, and the last step follows from the basic property of exponential functions.

**Case 2.** $e^\epsilon \Pr[\mathcal{M}(X') \in \mathcal{S}] \leq \delta^{\alpha/(\alpha-1)}$. In this case, we simply have

$$\Pr[\mathcal{M}(X) \in \mathcal{S}] \leq (e^\epsilon \Pr[\mathcal{M}(X') \in \mathcal{S}])^{1-1/\alpha} \tag{5}$$

$$= \delta,$$

where the first step follows from Eq. (3), and the second step follows from $e^\epsilon \Pr[\mathcal{M}(X') \in \mathcal{S}] \leq \delta^{\alpha/(\alpha-1)}$.

Combining both cases above, we can obtain the following:

$$\Pr[\mathcal{M}(X) \in \mathcal{S}] \leq \max\{\exp(\epsilon + \frac{\log(1/\delta)}{\alpha - 1}) \cdot \Pr[\mathcal{M}(X') \in \mathcal{S}], \delta\}$$

$$\leq \exp(\epsilon + \frac{\log(1/\delta)}{\alpha - 1}) \cdot \Pr[\mathcal{M}(X') \in \mathcal{S}] + \delta,$$

where the first step follows from combing Eq. (4) and Eq. (5), and the second step follows from the basic property of $\max$.

Therefore, we can conclude that $\mathcal{M}$ is $(\epsilon + \frac{\log(1/\delta)}{\alpha - 1}, \delta)$-DP, which completes the proof. $\quad\square$

**Lemma B.8** (Gaussian Mechanism, Corollary 3 in (Mironov, 2017)). *If the following conditions hold*

- *Let $\alpha > 1$.*

- *Let $D$ be the input dataset.*

- *Let $f$ be a real-valued function with sensitivity $L$.*

- *Let $z \sim \mathcal{N}(0, \sigma^2)$ be a Gaussian random variable.*

- *Let $G_\sigma f$ be a mechanism defined as $G_\sigma f(D) = f(D) + z$.*

*Then $G_\sigma f$ satisfies $(\alpha, \frac{\alpha L^2}{2\sigma^2})$-RDP.*

*Proof.* The Rényi divergence between a zero-mean Gaussian random variable and its offset has a closed-form solution, which can be computed as follows:

$$D_\alpha(\mathcal{N}(0, \sigma^2) \| \mathcal{N}(\mu, \sigma^2))$$

$$= (\alpha - 1)^{-1} \log \int_{-\infty}^{\infty} \sigma^{-1}(2\pi)^{-1/2} \exp(-0.5\alpha\sigma^{-2}x^2) \cdot \exp(-0.5(1 - \alpha)\sigma^{-2}(x - \mu)^2)\mathrm{d}x$$

$$= (\alpha - 1)^{-1} \log \exp(0.5(\alpha^2 - \alpha)\sigma^{-2}\mu^2)$$

$$= 0.5\alpha\sigma^{-2}\mu^2,$$

where the first step follows from Definition B.4 and the probability density function of Gaussian random variables, the second step follows from the property of Gaussian random variables, and the last step follows from basic algebra.

Therefore, we can conclude that for a real-valued function $f$ with sensitivity $L$, the Gaussian mechanism is $(\alpha, \frac{\alpha L^2}{2\sigma^2})$-RDP, since the offset between $z$ and $G_\sigma f$ is at most $L$.

Thus, we finish the proof.

$\quad\square$

## C  LINEAR PROGRAMMING: NOTATIONS AND BASIC FACTS

In Section C.1, we introduce basic notations. In Section C.2, we present basic algebra facts. In Section C.3, we introduce basic derivative fact. In Section C.4, we introduce basic norm inequalities. In Section C.5, we present basic matrix inequalities. In Section C.6, we present basic real number inequalities. In Section C.7, we present basic inequalities for $p$. In Section C.8, we provide basic PSD Matrix Facts. In Section C.9, we provide basic PSD inequalities. In Section C.10, we present basic PSD inequalities with trace. In Section C.11, we provide basic PD inequalities. In Section C.12, we introduce commutative property and eigenvalues of $A(I - cA)^{-1}$. In Section C.13, we introduce basic power calculations. In Section C.15, we introduce the simple constrained minimization by gradient descent method. In Section C.16, we introduce a fact about the equivalence of objective functions.

### C.1  NOTATIONS

In this section, we introduce basic notations.

**Vector Operations.**  We perform scalar operations to vectors by applying them element-wise, e.g., for vectors $x, y \in \mathbb{R}^n$, we denote the element-wise vector product $xy \in \mathbb{R}^n$ with $(xy)_i = x_i y_i$, for $i \in [n]$. In addition, we also $x \circ y$ to denote the element-wise product. For any vector $x \in \mathbb{R}^n$, the absolute value of $x$ is defined element-wise as $|x| := (|x_1|, |x_2|, \cdots, |x_n|)$.

**Basic Notations.**  We denote all the positive real numbers as $\mathbb{R}_{>0}$, and denote $m$-dimensional positive real vectors as $\mathbb{R}_{>0}^m$. We use $\pm \delta$ to denote a real value with magnitude at most $\delta$, e.g. $a = e^{\pm \delta} b$ means $a \in [e^{-\delta} b, e^{\delta} b]$.

**Matrices.**  If a matrix $A \in \mathbb{R}^{m \times n}$ has full column-rank and no zero rows, the matrix $A$ is non-degenerate. Let $B \in \mathbb{R}^{n \times n}$ be a symmetric matrix. $B \in \mathbb{R}^{n \times n}$ is positive semidefinite (PSD) if $x^\top B x \geq 0$ for all $x \in \mathbb{R}^n$, and positive definite (PD) if $x^\top B x > 0$ for all $x \in \mathbb{R}^n$. We denote the kernel (the null space) of the matrix $A \in \mathbb{R}^{m \times n}$ as $\ker(A)$, i.e., $\ker(A) := \{x \in \mathbb{R}^n : Ax = \mathbf{0}_m\}$. We denote the image space (the column space) as $\mathrm{im}(A)$, i.e., $\mathrm{im}(A) := \{y \in \mathbb{R}^m : y = Ax\}$.

**Matrix Operations.**  Let $A, B \in \mathbb{R}^{n \times n}$ be two symmetric matrices. We use $A \preceq B$ to indicate that $x^\top A x \leq x^\top B x$ for all $x \in \mathbb{R}^n$. We define $\prec, \succeq, \succ$ analogously. For matrices $A, B \in \mathbb{R}^{n \times m}$, we denote the Hadamard product as $A \circ B$, i.e., for $i \in [n], j \in [m]$, $(A \circ B)_{i,j} := A_{i,j} \cdot B_{i,j}$. We define $A^{\circ 2} := A \circ A$. We denote the number of nonzero entries in $A$ as $\mathrm{nnz}(A)$. For symmetric matrices $A, B \in \mathbb{R}^{n \times n}$ with scalars $0 < c_1 \leq c_2$, we write $A \in [c_1, c_2] \cdot B$ to mean that $c_1 B \preceq A \preceq c_2 B$.

**Diagonals.**  Let $A \in \mathbb{R}^{n \times n}$ be a matrix. We define $\mathrm{Diag}(A) \in \mathbb{R}^n$ with $\mathrm{Diag}(A)_i := A_{i,i}$ for all $i \in [n]$. For a vector $x \in \mathbb{R}^n$, we define $\mathrm{diag}(x) \in \mathbb{R}^{n \times n}$ as the diagonal matrix with $\mathrm{diag}(x)_{i,i} := x_i$ for $i \in [n]$. Additionally, we use upper case to denote a diagonal matrix to which the vector transforms, e.g. $X := \mathrm{diag}(x) \in \mathbb{R}^{n \times n}$ for $x \in \mathbb{R}^n$.

**Fundamental Matrices.**  For a non-degenerate matrix $A \in \mathbb{R}^{m \times n}$, we define $P(A) := A(A^\top A)^{-1} A^\top$ as the orthogonal projection matrix onto $A$'s image. We define $\sigma(A) := \mathrm{Diag}(P(A))$ as $A$'s leverage scores. We define $\Sigma(A) := \mathrm{diag}(\sigma(A))$. We define $\Lambda(A) := \Sigma(A) - P^{\circ 2}(A)$ as a Laplacian matrix and $\overline{\Lambda}(A) := \Sigma(A)^{-1/2} \Lambda(A) \Sigma(A)^{-1/2}$ as a normalized Laplacian matrix.

**Norms.**  For any positive real number $p > 0$ and vector $x \in \mathbb{R}^n$, we define the vector $\ell_p$ norm as $\|x\|_p := (\sum_{i=1}^n |x_i|^p)^{1/p}$. We define the vector $\ell_0$ norm as the number of non-zero elements in $x$, i.e., $\|x\|_0 := \sum_{i=1}^n \mathbf{1}[x_i = \neq 0]$. For a positive definite matrix $A \in \mathbb{R}^{n \times n}$ and a vector $x \in \mathbb{R}^n$, we define $\|x\|_A := (x^\top A x)^{1/2}$. For a vector $w \in \mathbb{R}_{>0}^n$, we define $\|x\|_w := (\sum_{i=1}^n w_i x_i^2)^{1/2}$. If we let $W := \mathrm{diag}(w)$, then know that $\|x\|_w = \|x\|_W$. For any matrix spectral norm $\| \cdot \|$, we define $\|M\| := \sup_{\|x\|_2 = 1} \|Mx\|_2$.

**Calculus.** Let $g(x, y) \in \mathbb{R}$ be a function of two vectors $x \in \mathbb{R}^{n_1}$ and $y \in \mathbb{R}^{n_2}$. We define the gradient of $g$ with respect to $x$ at $(a, b) \in \mathbb{R}^{n_1 \times n_2}$ as $\nabla_x g(a, b) \in \mathbb{R}^{n_1}$, where $\nabla_x g(a, b)_i := \frac{\mathrm{d}}{\mathrm{d}x_i} g(a, b)$, and define $\nabla_y g(a, b)_i := \frac{\mathrm{d}}{\mathrm{d}y_i} g(a, b)$, $\nabla_{yy} g(a, b)_{i,j} := \frac{\mathrm{d}}{\mathrm{d}y_i} \frac{\mathrm{d}}{\mathrm{d}y_j} g(a, b)$, $\nabla_{xx} g(a, b)_{i,j} := \frac{\mathrm{d}}{\mathrm{d}x_i} \frac{\mathrm{d}}{\mathrm{d}x_j} g(a, b)$. For $h : \mathbb{R}^n \to \mathbb{R}^m$ and $x \in \mathbb{R}^n$, we use $J_h(x) \in \mathbb{R}^{m \times n}$ to denote the Jacobian of $h$ at $x$, where $J_h(x)_{i,j} := \frac{\mathrm{d}}{\mathrm{d}x_j} h(x)_i$ for $i \in [m], j \in [n]$. For $f : \mathbb{R}^n \to \mathbb{R}$ and $x, h \in \mathbb{R}^n$, we define the directional derivative of $f$ in direction $h$ at $x$ as $Df(x)[h] := \lim_{t \to 0} (f(x + th) - f(x))/t$.

**Convex Sets.** A set $U \subseteq \mathbb{R}^k$ is convex if $t \cdot x + (1 - t) \cdot y \in U$ for all $x, y \in U, t \in [0, 1]$. A set $U \subseteq \mathbb{R}^k$ is symmetric if $x \in U$, then $-x \in U$ for all $x \in U$. For all $\alpha > 0$ and $U \subseteq \mathbb{R}^k$, we define $\alpha U := \{x \in \mathbb{R}^k : \alpha^{-1} x \in U\}$. For all $p \in [1, \infty]$ and $r > 0$, we call the symmetric convex set $\{x \in \mathbb{R}^k : \|x\|_p \leq r\}$ the $\ell_p$ ball of radius $r$.

## C.2 BASIC ALGEBRA FACTS

**Fact C.1.** *For any vectors $a, b, c \in \mathbb{R}^m$, square matrix $M$, diagonal matrix $D$, and symmetric matrix $P \in \mathbb{R}^{m \times m}$, we have*

- $a \circ b = \mathrm{diag}(a)b = \mathrm{diag}(b)a$.

- $a^\top \mathrm{diag}(b)c = a^\top \mathrm{diag}(c)b$.

- $e_j^\top a_i e_i = e_j^\top \mathrm{diag}(a)e_i$.

- $\mathrm{diag}(a) = \sum_{i=1}^m e_i e_i^\top a_i$.

- $e_i^\top M e_i = M_{i,i}$.

- $(\mathrm{diag}(a)M) \circ I = \mathrm{Diag}(M)a$.

- $\mathrm{Diag}(\mathrm{diag}(b)P) = \mathrm{Diag}(P \mathrm{diag}(b)) = \mathrm{Diag}(P)b$.

- $\mathrm{Diag}(P \mathrm{diag}(b)P) = (P \circ P)b$.

*Proof.* Let $i \in [m]$/ For $i$-th entry of vector, it is $\sum_{j=1}^m P_{i,j}^2 b_j$. Thus it's true. The other parts of the statement are trivial. $\square$

## C.3 BASIC DERIVATIVE FACTS

**Fact C.2.** *Let $A$ denote a positive definite matrix, then we have*

- **Part 1.**

$$\frac{\mathrm{d}A^{-1}}{\mathrm{d}t} = -A^{-1} \frac{\mathrm{d}A}{\mathrm{d}t} A^{-1}.$$

- **Part 2.**

$$\frac{\mathrm{d}\log\det(A)}{\mathrm{d}t} = \mathrm{tr}[A^{-1} \frac{\mathrm{d}A}{\mathrm{d}t}].$$

## C.4 BASIC NORM INEQUALITIES

**Fact C.3.** *If the following conditions hold:*

- *Let $a, b \in \mathbb{R}^m$ be two vectors.*

*Then, for any vector norm $\| \cdot \|$, we have*

$$\|a \circ b\| \leq \|a\|_\infty \|b\|.$$

*Proof.* For $i \in [m]$, we have
$$|(a \circ b)_i| = |a_i b_i| \leq \|a\|_\infty |b_i|,$$
where the first step follows from the definition of Hadamard product, and the second step follows from $|a_i| \leq \|a\|_\infty$ for $i \in [m]$.

Since the inequality holds for $i \in [m]$, thus we have
$$\|a \circ b\| \leq \|a\|_\infty \|b\|.$$
The proof is complete. $\qquad\square$

**Fact C.4** (Folklore)**.** *Let $x, y \in \mathbb{R}^m$ be two vectors. If for all $i \in [m]$, $|x_i| \leq |y_i|$, then the following statements are true:*

- **Part 1.** *For any positive real number $p > 0$, $\|x\|_p \leq \|y\|_p$.*

- **Part 2.** *For any $w \in \mathbb{R}^m$, $\|x\|_w \leq \|y\|_w$.*

- **Part 3.** *For any $w \in \mathbb{R}^m$ and $W := \mathrm{diag}(w)$, $\|x\|_W \leq \|y\|_W$.*

**Fact C.5.** *Let $a, b > 0$ be positive real numbers and $r \in (0, 0.5)$. Let $|a^{-1}(a - b)| \leq r$. We have $a = e^{\pm 1.5r} b$.*

*Proof.* Since we have $|a^{-1}(a - b)| \leq r$, we can imply by the definition of absolute value that
$$-r \leq 1 - \frac{b}{a} \leq r.$$

Thus, we can imply by basic algebra that
$$(1 + r)a \leq b \leq (1 - r)a.$$

Therefore, if we apply Part 1 of Fact C.28 for the lower bound of $b$ and apply Part 2 of Fact C.28 for the upper bound of $b$, we can conclude that
$$e^{-1.5r} a \leq b \leq e^{1.5r} a.$$

This is equivalent to
$$a = e^{\pm 1.5r} b.$$

Thus, we finish the proof. $\qquad\square$

**Fact C.6.** *If the following conditions hold:*

- *Let $w, v \in \mathbb{R}^m_{>0}$ be two positive vectors.*

- *Let $W := \mathrm{diag}(w)$.*

- *Let $r \in (0, 0.5)$.*

- *We assume $\|W^{-1}(w - v)\|_\infty \leq r$.*

*Then the following statement is true:*
$$w = e^{\pm 1.5r} v.$$

*Proof.* Since we have $\|W^{-1}(w - v)\|_\infty \leq r$, we have the following for for all $i \in [m]$:
$$|w_i^{-1}(w_i - v_i)| \leq r,$$
where this step follows from the definition of infinity norm.

By Fact C.5, we can conclude for all $i \in [m]$ that
$$w_i = e^{\pm 1.5r} v_i.$$

Thus, we can combine all the entries $w_i$ and $v_i$ for all $i \in [m]$, and directly obtain
$$w = e^{\pm 1.5r} v,$$
which finishes the proof. $\qquad\square$

## C.5 BASIC MATRIX INEQUALITIES

**Definition C.7.** *We say matrix $P$ is a projection matrix if $PP = P$.*

Now, we show the basic properties of the projection matrix without proof.

**Fact C.8** (Folklore). *If $P$ is a projection matrix as defined in Definition C.7, we have the following:*

- *All the eigenvalues of $P$ are either $0$ or $1$.*

- $P \succeq 0$.

- $P \preceq I$.

- $P \circ I \preceq I$.

- $P_{i,i} \in [0, 1]$ *for all $i \in [n]$.*

- $e_i^\top P e_i \in [0, 1]$ *for all $i \in [n]$.*

**Fact C.9** (Folklore). *Let $A$ denote a symmetric matrix such that $A_{i,i} \geq 0$, $A_{i,j} \leq 0$ and $\sum_{j=1}^{n} A_{i,j} \geq 0$ for all $i$, then we have $A \succeq 0$.*

## C.6 BASIC REAL NUMBER INEQUALITIES

**Fact C.10** (folklore). *If the following conditions hold:*

- *Let $x \in (0, 1)$.*

- *Let $y > 0$.*

- *Let $f = (\frac{1+x}{1-x})^y$.*

*Then, we have $f \geq 1$.*

**Fact C.11.** *If the following conditions hold:*

- *Let $a \geq 1$.*

- *Let $b > 0 > c$.*

- *Let $x \in [0, 1]$.*

*Then we have*

$$\min\{ax^b, x^c\} \leq a^{\frac{-c}{b-c}}.$$

*Proof.* **Case 1.** We first consider the extreme case when $ax^b = x^c$. In this case, we can solve for $x$ and obtain that $x = a^{-\frac{1}{b-c}}$. Thus, we have

$$\min\{ax^b, x^c\} = x^c$$
$$= (a^{-\frac{1}{b-c}})^c$$
$$= a^{-\frac{c}{b-c}},$$

where the first step follows from $ax^b = x^c$, the second step follows from $x = a^{-\frac{1}{b-c}}$, and the last step follows from basic algebra.

Therefore, the original statement is true when $ax^b = x^c$.

**Case 2.** Now we just need to show that as long as $ax^b$ is not equal $x^c$, then one of them must be at most $a^{-c/(b-c)}$. This is obviously true since both two functions are monotonically functions. $\square$

**Fact C.12.** *Let $\epsilon \in (0, 1)$ and $\alpha > 0$. We have $\epsilon = \frac{\alpha}{1+\alpha}$ is the minimizer for function $f(\epsilon) := \frac{1}{(1-\epsilon)\epsilon^\alpha}$.*

*Proof.* The derivative of the function $f$ is

$$\frac{\mathrm{d}f}{\mathrm{d}\epsilon} = ((1-\epsilon)\epsilon^\alpha)^{-2}(-\epsilon^\alpha + \alpha(1-\epsilon)\epsilon^{\alpha-1})$$
$$= (1-\epsilon)^{-2}\epsilon^{-2\alpha}(\alpha\epsilon^{\alpha-1} - (\alpha+1)\epsilon^\alpha)$$
$$= (1-\epsilon)^{-2}\epsilon^{-\alpha-1}(\alpha - (\alpha+1)\epsilon),$$

where the first step follows from the chain rule, and the second step and the last steps follow from basic algebra.

Therefore, we let $\frac{\mathrm{d}f}{\mathrm{d}\epsilon} = 0$ and solve for $\epsilon$, which directly yields

$$\epsilon = \frac{\alpha}{1+\alpha}.$$

Thus, we can conclude that $\epsilon = \frac{\alpha}{1+\alpha}$ is the minimizer for function $f$. $\qquad\square$

### C.7    BASIC INEQUALITIES FOR $p$

**Fact C.13.** *For any $p \in (0,4)$, we have $1 - p/4 \le 2/p - |1 - 2/p|$.*

*Proof.* **Case 1.** Let $p \in (0,2]$. We have $|1 - 2/p| = 2/p - 1$. Therefore, the original inequality is equivalent to

$$1 - p/4 \le 2/p - (2/p - 1).$$

Moving the terms, we can obtain

$$p/4 \ge 0,$$

which trivially holds since $p \in (0,2]$.

**Case 2.** Let $p \in (2,4)$. We have $|1 - 2/p| = 1 - 2/p$. Therefore, the original inequality is equivalent to

$$1 - p/4 \le 2/p - (1 - 2/p).$$

Moving the terms, we have

$$p/4 + 4/p \ge 2,$$

which is a true statement since $p/4 + 4/p \ge 2\sqrt{(p/4)\cdot(4/p)} = 2$.

Since $1 - p/4 \le 2/p - |1 - 2/p|$ holds for both $p \in (0,2]$ and $p \in (2,4)$, the proof is finished.

$\qquad\square$

**Fact C.14.** *If $p \in (0,4)$, then we have*

$$p^2(4-p) \cdot (4 + 8/p) \cdot (1 + 4/p) \le 2^{10}(1 - p/4).$$

*Proof.* We can show

$$p^2(4-p)(4+8/p)(1+4/p) \le (4-p)\cdot(4p+8)\cdot(p+4)$$
$$\le (4-p)\cdot 24 \cdot 8$$
$$\le 2^8 \cdot (4-p)$$
$$= 2^{10} \cdot (1 - p/4),$$

where the first step follows from basic algebra, the second step follows from $p \in (0,4)$, and the third step follows from $24 \cdot 8 \le 2^8$, and the last step follows from pulling out the factor $4$.

$\qquad\square$

**Fact C.15.** *If $p \in (0, 4)$, we have*

$$\frac{\min\{1/4, 1/(2p)\}}{\max\{4, 8/p\}} \geq \frac{1}{16} \cdot \frac{1}{p/2 + 2/p}.$$

*Proof.* **Case 1.** For $p \geq 2$, we have

$$\frac{\min\{1, 2/p\}}{\max\{1, 2/p\}} = \frac{1}{2/p}.$$

**Case 2.** For $p < 2$, we have

$$\frac{\min\{1, 2/p\}}{\max\{1, 2/p\}} = \frac{2/p}{1} = \frac{1}{p/2}.$$

Therefore, by combining two cases, We can show that

$$\frac{\min\{1/4, 1/(2p)\}}{\max\{4, 8/p\}} = \frac{1}{16} \cdot \frac{\min\{1, 2/p\}}{\max\{1, 2/p\}}$$

$$\geq \frac{1}{16} \cdot \frac{1}{p/2 + 2/p}, \tag{6}$$

where the last step follows from $x \geq 1/a$ and $x \geq 1/b$ implies that $x \geq \frac{1}{a+b}$. $\qquad \square$

### C.8 BASIC PSD MATRIX FACTS

**Fact C.16** (Schur product theorem, Theorem VII in page 14 in (Schur, 1911)). *If the following conditions hold:*

- *Let $A, B \in \mathbb{R}^{m \times m}$ be two positive semidefinite matrices.*

*Then $A \circ B$ is a positive semi-definite matrix.*

**Fact C.17** (Folklore). *If the following conditions hold:*

- *Let $A \in \mathbb{R}^{m \times m}$ be a symmetric matrix.*

- *Let $0 \preceq A \preceq I$.*

- *For $i \in [m]$, denote the eigenvalue of the matrix $A$ as $\lambda_i$.*

*Then for $i \in [m]$, $\lambda_i \in [0, 1]$.*

### C.9 BASIC PSD INEQUALITIES

**Fact C.18** (Folklore). *If the following conditions hold:*

- *Let $A \in \mathbb{R}^{m \times n}$.*

- *Let $B \in \mathbb{R}^{m \times m}$ be a matrix $B \succeq 0$.*

*Then we have $A^\top B A \succeq 0$.*

**Fact C.19.** *If the following conditions hold*

- *Let $w, v \in \mathbb{R}^m_{>0}$ with $w_i = e^{\delta_i} v_i$ for $|\delta_i| \leq \delta$ for all $i \in [m]$.*

- *Let $p > 0$.*

*Then we have*

$$e^{-|1-2/p|\delta} W^{1-2/p} \preceq V^{1-2/p} \preceq e^{|1-2/p|\delta} W^{1-2/p}.$$

*Proof.* For any $i \in [m]$, we have

$$(\frac{v_i}{w_i})^{1-2/p} = e^{-(1-2/p)\delta_i} \in [e^{-|1-2/p|\delta}, e^{|1-2/p|\delta}].$$

Thus, we complete the proof. $\square$

**Fact C.20.** *If the following conditions hold:*

- *Let $A \in \mathbb{R}^{m \times n}$.*

- *Let $k \in \mathbb{R}$ be an arbitrary real number such that $k > 0$.*

- *Let $W, V \in \mathbb{R}^{m \times m}$ denote a positive diagonal matrix.*

- *We have $W \preceq k \cdot V$.*

*Then the following statement is true:*

- $(A^\top W A)^{-1} \succeq k^{-1}(A^\top V A)^{-1}$.

- $A(A^\top W A)^{-1} A^\top \succeq k^{-1} A(A^\top V A)^{-1} A^\top$.

*Proof.* Since $W \preceq kV$, then $A^\top W A \preceq k A^\top V A$. Taking the inverse on both sides, we have $(A^\top W A)^{-1} \succeq k^{-1}(AVA)^{-1}$. Thus, we have $A(A^\top W A)^{-1} A^\top \succeq k^{-1} A(A^\top V A)^{-1} A^\top$. $\square$

**Fact C.21.** *If the following conditions hold:*

- *Let $\epsilon > 0$.*

- *Let $I_{w \leq \frac{\epsilon}{m}} \in \mathbb{R}^{m \times m}$ be the diagonal matrix where $I_{\epsilon,i,i} = 1$ if $w_i > \frac{\epsilon}{m}$ and $I_{\epsilon,i,i} = 0$ otherwise.*

- *Let $I_{w > \frac{\epsilon}{m}} := I - I_{w \leq \frac{\epsilon}{m}}$.*

*Then the following statement is true:*

$$A^\top W^{1-2/p} A \preceq \frac{1}{1-\epsilon} A^\top W^{1-2/p} I_{w > \frac{\epsilon}{m}} A.$$

*Proof.* Note that

$$
\begin{aligned}
\mathrm{tr}[(A^\top W^{1-2/p} A)^{-1} A^\top W^{1-2/p} I_{w \leq \frac{\epsilon}{m}} A] &= \mathrm{tr}[A(A^\top W^{1-2/p} A)^{-1} A^\top W^{1-2/p} I_{w \leq \frac{\epsilon}{m}}] \\
&= \sum_{i \in [m]} (A(A^\top W^{1-2/p} A)^{-1} A^\top)_{i,i} (W^{1-2/p} I_{w \leq \frac{\epsilon}{m}})_{i,i} \\
&= \sum_{i \in [m]} w_i^{2/p} (W^{1-2/p} I_{w \leq \frac{\epsilon}{m}})_{i,i} \\
&= \sum_{i \in [m] : w_i \leq \frac{\epsilon}{m}} w_i^{2/p} w_i^{1-2/p} \\
&= \sum_{i \in [m] : w_i \leq \frac{\epsilon}{m}} w_i \\
&\leq m \cdot \frac{\epsilon}{m} = \epsilon, \quad (7)
\end{aligned}
$$

where the first step follows from the cyclic property of the trace, the second step follows from the fact that $W$ and $I_{w \leq \frac{\epsilon}{m}}$ are diagonal matrices, the third step follows from the definition of Lewis weight (Definition E.1, $c_{i,i} = (W^{1/2-1/p} c W^{1/2-1/p})_{i,i}/w_i^{1-2/p} = w_i/w^{1-2/p} = w_i^{2/p}$), the fourth step follows from the definition of $I_{w \leq \frac{\epsilon}{m}}$, the fifth step follows from basic algebra, and the last step follows from $w_i \leq \frac{\epsilon}{m}$.

We have,

$$A^\top W^{1-2/p} I_{w \le \frac{\epsilon}{m}} A \preceq \epsilon \cdot A^\top W^{1-2/p} A,$$

where the step follows from Eq. (7) and $X \preceq \text{tr}[Y^{-1}X]Y$ (Fact C.23)

Note that $I_{w > \frac{\epsilon}{m}} = I - I_{w \le \frac{\epsilon}{m}}$ implies

$$A^\top W^{1-2/p} I_{w > \frac{\epsilon}{m}} A + A^\top W^{1-2/p} I_{w \le \frac{\epsilon}{m}} A = A^\top W^{1-2/p} A$$

The above two equations implies

$$A^\top W^{1-2/p} I_{w > \frac{\epsilon}{m}} A \succeq (1 - \epsilon) \cdot A^\top W^{1-2/p} A,$$

By rescaling the factor $1 - \epsilon$ on both sides, we get

$$A^\top W^{1-2/p} A \preceq \frac{1}{1 - \epsilon} A^\top W^{1-2/p} I_{w > \frac{\epsilon}{m}} A.$$

$\square$

## C.10 BASIC PSD INEQUALITIES WITH TRACE

**Fact C.22** (Folklore). *Let $Z$ be any PSD matrix. The following statement is true:*

$$Z \preceq \|Z\| \cdot I \preceq \text{tr}[Z] \cdot I.$$

**Fact C.23.** *If the following conditions hold:*

- *Let $X, Y$ be any PSD matrices.*

*Then the following statement is true:*

$$X \preceq \text{tr}[Y^{-1}X] \cdot Y.$$

*Proof.* Due to cyclic property of the trace, we know that

$$\text{tr}[Y^{-1}X] = \text{tr}[Y^{-1/2}XY^{-1/2}].$$

Thus, the original statement

$$X \preceq \text{tr}[Y^{-1}X] \cdot Y$$

is equivalent to

$$X \preceq \text{tr}[Y^{-1/2}XY^{-1/2}]Y,$$

which is further equivalent to

$$Y^{-1/2}XY^{-1/2} \preceq \text{tr}[Y^{-1/2}XY^{-1/2}] \cdot I,$$

which is a true statement following Fact C.22.

Thus, since the equivalent form of the original statement is true, we can complete the proof. $\square$

**Fact C.24.** *If the following conditions hold:*

- *Let $A \in \mathbb{R}^{m \times n}$.*

- *Let $B, C \in \mathbb{R}^{m \times m}$ be non-negative diagonal matrices.*

- *Let*

$$\alpha := \text{tr}[(A^\top BA)^{-1}(A^\top |C - B|A)].$$

*then we have*

$$(1 - \alpha)A^\top BA \preceq A^\top CA \preceq (1 + \alpha)A^\top BA.$$

*which is equivalent to*

$$A^\top CA \in [1 - \alpha, 1 + \alpha] \cdot A^\top BA$$

*Proof.* Note that

$$(1 - \alpha)A^\top BA \preceq A^\top CA \preceq (1 + \alpha)A^\top BA$$

is equivalent to

$$-\alpha A^\top BA \preceq A^\top (C - B)A \preceq \alpha A^\top BA.$$

The following equation implies the above equation

$$A^\top |C - B|A \preceq \alpha \cdot A^\top BA.$$

Since $A^\top |C - B|A$ and $A^\top BA$ are both PSD matrices, we can choose $X = A^\top |C - B|A$ and $Y = A^\top BA$ and then apply Fact C.23 to show that the above equation is true.

Thus, we complete the proof. $\square$

### C.11 BASIC PD INEQUALITIES

**Fact C.25** (Folklore). *If the following conditions hold:*

- *Let $Q \in \mathbb{R}^{m \times m}$ denote a matrix that $Q \succ 0$.*

- *Let $W \in \mathbb{R}^{m \times m}$ be positive diagonal matrix.*

*Then we have $WQW \succ 0$.*

### C.12 COMMUTATIVE PROPERTY AND EIGENVALUES OF $A(I - cA)^{-1}$

**Fact C.26.** *If the following conditions hold:*

- *Let $A \in \mathbb{R}^{m \times m}$.*

- *Let $(I - cA)$ be invertible.*

*Then, we have*

$$(I - cA)^{-1}A = A(I - cA)^{-1}.$$

*Proof.* It is easy to see

$$(I - cA)A = A - cA^2 = A(I - cA).$$

Then multiplyling $(I - cA)^{-1} \cdot (I - cA)^{-1}$ on both sides of the above equation, then we can get the following

$$A(I - cA)^{-1} = (I - cA)^{-1}A.$$

We remark that an alternative proof of the following claim can use von Neumann series in Claim 8 in (Price et al., 2017). $\square$

**Fact C.27.** *For any symmetric $A \in \mathbb{R}^{m \times m}$ and real number $c > 0$, we have $A(I - cA)^{-1}$ whose eigenvalues are of the form $\lambda/(1 - c\lambda)$ for each eigenvalue $\lambda$ of $A$.*

*Proof.* Assume $A = U\Lambda U^\top$ here $\Lambda \in \mathbb{R}^{k \times k}$ is a diagonal matrix, $U \in \mathbb{R}^{m \times k}$ has all columns are orthogonal to each other and each column has $\ell_2$ norm equal to 1. Since $A$ might have full rank, thus $k \le m$. It is obvious $U^\top U = I$.

Thus, we can show

$$
\begin{aligned}
U\Lambda U^\top (I - cU\Lambda U^\top)^{-1} &= U\Lambda U^\top (UU^\top - cU\Lambda U^\top)^{-1} \\
&= U\Lambda U^\top (U(I - c\Lambda)U^\top)^{-1} \\
&= U\Lambda U^\top (U(I - c\Lambda)^{-1}U^\top) \\
&= U\Lambda(I - c\Lambda)^{-1}U^\top,
\end{aligned}
$$

where the first step follows from $I = UU^\top$, the second step follows from basic algebra, the third step follows from the definition of the inverse matrix, and the last step follows from $U^\top U = I$. $\square$

### C.13 BASIC POWER CALCULATIONS

Before proving the main statement, we first show a fact. Note that $e^x = \sum_{i=0}^\infty \frac{1}{i!} x^i$.

**Fact C.28.** *For any $x \in (0, 0.5]$, we have*

- $1 - x \ge e^{-1.5x}$

- $1 + x \le e^{1.5x}$

- $e^x \le 1 + \frac{4}{3}x$

- $e^{-x} \ge 1 - \frac{4}{3}x$

- $e^x \ge 1 + x - x^2$

- $e^x \le 1 + x + x^2$

**Fact C.29.** *If the following conditions hold:*

- *Let $x \in (0, 0.5)$.*

- *Let $y$ be any real number such that $|y| < 0.5$.*

*Then the following statement is true:*

$$
e^{\pm x}(e^y - 1) = (e^y - 1) \pm 2x|y|.
$$

*Proof.*

$$
\begin{aligned}
e^{\pm x}(e^y - 1) &= (1 \pm \frac{4}{3}x)(e^y - 1) \\
&= (e^y - 1) \pm \frac{4}{3}x(e^y - 1) \\
&= (e^y - 1) \pm \frac{4}{3}x\frac{4}{3}|y| \\
&= (e^y - 1) \pm 2x|y|,
\end{aligned}
$$

where the first step follows from applying Fact C.28 on $e^{\pm x}$, the second step follows from basic algebra, the third step follows from applying Fact C.28 on $e^y$, the last step follows from basic algebra. $\square$

**Fact C.30.** *If the following conditions hold*

- *Let $n \ge 2$ denote positive integers.*

- *Let $p > 0$.*

- *Let $\beta := 4 \cdot (1 + 2/p)^2 \cdot \sqrt{n}$.*

- *Let $\theta := 2 \cdot |1 - 2/p| \cdot \sqrt{n}$.*

- *Let $\delta \in (0, 0.1/\beta]$.*

*Then we can show*

- **Part 1.** $(1 - \theta\delta)^{-2/p} \geq 1 - \beta\delta$.

- **Part 2.** $(1 + \theta\delta)^{-2/p} \leq 1 + \beta\delta$.

- **Part 3.** $(1 - \theta\delta)^{-2/p} \leq 1 + \beta\delta$.

- **Part 2.** $(1 + \theta\delta)^{-2/p} \geq 1 - \beta\delta$.

*Proof.* Note that $\beta\delta \in (0, 0.1]$. Also note that $\theta\delta \leq 0.1 \cdot \theta/\beta \leq (0, 0.5)$.

We can show

$$
\begin{aligned}
4\theta/p &= 4 \cdot 2|1 - 2/p|\sqrt{n}/p \\
&\leq 4 \cdot (1 + 2/p) \cdot |1 - 2/p|\sqrt{n} \\
&\leq 4 \cdot (1 + 2/p)^2 \sqrt{n} \\
&= \beta.
\end{aligned}
$$

Thus,

$$
4\theta\delta/p \leq 0.1
$$

**Proof of Part 1.**

We can show

$$
\begin{aligned}
(1 - \theta\delta)^{-2/p} &\geq 1 \\
&\geq 1 - \beta\delta,
\end{aligned}
$$

where first step is trivial, and the last step follows $\beta\delta > 0$.

**Proof of Part 2.**

We can show

$$
\begin{aligned}
(1 + \theta\delta)^{-2/p} &\leq 1 \\
&\leq 1 + \beta\delta,
\end{aligned}
$$

where the first step is trivial, and the last step follows $\beta\delta > 0$.

**Proof of Part 3.**

Using the fact $(1 - x) \geq e^{-1.5x}$ for all $x \in (0, 0.5)$ (see Fact C.28), we can show

$$
(1 - \theta\delta) \geq e^{-1.5\theta\delta}.
$$

Then, we can show

$$
\begin{aligned}
(1 - \theta\delta)^{-2/p} &\leq (e^{-1.5\theta\delta})^{-2/p} \\
&= e^{3\theta\delta/p} \\
&\leq 1 + 4\theta\delta/p \\
&\leq 1 + \beta\delta,
\end{aligned}
$$

where the third step follows from $e^x \leq 1 + \frac{4}{3}x$ (see Fact C.28, and the last step follows from $4\theta/p \leq \beta$.

**Proof of Part 4.**

Using the fact $(1 + x) \leq e^{1.5x}$ for all $x \in (0, 0.5)$ (see Fact C.28), we can show

$$
(1 + \theta\delta) \leq e^{1.5\theta\delta}.
$$

Then, we can show

$$
\begin{aligned}
(1 + \theta\delta)^{-2/p} &\geq (e^{1.5\theta\delta})^{-2/p} \\
&= e^{-3\theta\delta/p} \\
&\geq 1 - 4\theta\delta/p \\
&\geq 1 - \beta\delta,
\end{aligned}
$$

where the third step follows from $e^{-x} \geq 1 - \frac{4}{3}x$ (see Fact C.28), and the last step follows from $4\theta/p \leq \beta$.

$\square$

## C.14 LEVERAGE SCORE EQUIVALENCE FORMAT

**Fact C.31** (Folklore). *Let $\sigma$ denote the leverage score. Then, the following quantities are equivalent*

- $e_i^\top W^{1/2-1/p} A (A^\top W^{1-2/p} A)^{-1} W^{1/2-1/p} e_i$.

- $(W^{1/2-1/p} A (A^\top W^{1-2/p} A)^{-1} W^{1/2-1/p})_{i,i}$.

- $\sigma_i(W^{1/2-1/p} A)$.

## C.15 SIMPLE CONSTRAINED MINIMIZATION BY GRADIENT DESCENT METHOD

**Theorem C.32** (Simple Constrained Minimization for Twice Differentiable Function, Theorem 52 in page 50 in (Lee & Sidford, 2019)). *If the following conditions hold:*

- *Let $H$ be a positive definite matrix.*

- *Let $Q \subseteq \mathbb{R}^m$ be a convex set.*

- *Let $f : Q \to \mathbb{R}^m$ be a twice differentiable function.*

- *Suppose that there are constraints $0 \leq \mu \leq L$ such that for all $x \in Q$, we have $\mu \cdot H \preceq \nabla^2 f(x) \preceq L \cdot H$.*

- *Let $x^{(0)} \in Q$.*

- *Let $k \geq 0$.*

- *Apply the update rule*

$$
x^{(k+1)} = \arg\min_{x \in Q} \nabla f(x^{(k)})^\top (x - x^{(k)}) + \frac{L}{2} \|x - x^{(k)}\|_H^2.
$$

*Then, we have*

$$
\|x^{(k)} - x^*\|_H^2 \leq (1 - \frac{\mu}{L})^k \|x^{(0)} - x^*\|_H^2.
$$

## C.16 EQUIVALENCE OF OBJECTIVE FUNCTIONS

**Fact C.33.** *If the following conditions hold:*

- *Let $B \in \mathbb{R}^{m \times m}$ be a diagonal matrix.*

- *Let $w \in \mathbb{R}^m$.*

- *Let $b \in \mathbb{R}^m$.*

- *Define $f(w) := \langle w, Bw \rangle - 2\langle b, w \rangle$.*

*Then we have*

$$\arg \min_{w \in \mathbb{R}^m} f(w) = \arg \min_{w \in \mathbb{R}^m} \|w - B^{-1}b\|_B^2.$$

*Proof.* We have

$$\|w - B^{-1}b\|_B^2 = w^\top B w - 2(B^{-1}b)^\top B w + (B^{-1}b)^\top B (B^{-1}b)$$
$$= w^\top B w - 2b^\top w + b^\top B^{-1}b$$
$$= \langle w, Bw \rangle - 2\langle b, w \rangle + b^\top B^{-1}b, \tag{8}$$

where the first step follows from the definition of $\|\cdot\|_B$, the second step follows from basic algebra, the third step follows from the definition of the inner product.

Thus, we have

$$\arg \min_{w \in \mathbb{R}^m} \|w - B^{-1}b\|_B^2 = \arg \min_{w \in \mathbb{R}^m} \langle w, Bw \rangle - 2\langle b, w \rangle + b^\top B^{-1}b$$
$$= \arg \min_{w \in \mathbb{R}^m} f(w),$$

where the first step follows from Eq. (8), the second step follows from the fact that $b^\top B^{-1}b$ is a constant. $\qquad\square$

# D    LINEAR PROGRAMMING: BACKGROUND

In Section D.1, we introduce the definition of linear program. In Section D.2, we present the basics of the self-concordance property. In Section D.3, we show the definition of the weighted central path. In Section D.4, we introduce the Newton step. In Section D.5, we present the definition of the weight function. In Section D.6, we introduce the concept of centrality. In Section D.7, we introduce the derivative computation of the volumetric barrier. In Section D.8, we present the derivatives of the potential function. In Section D.9, we show some basic properties of the projection matrix.

## D.1    DEFINITION OF LINEAR PROGRAM

**Definition D.1** (Linear program, Implicit in page 3 in (Lee & Sidford, 2019))**.** *If the following conditions hold:*

- *Let $A \in \mathbb{R}^{m \times n}, b \in \mathbb{R}^n$ be a non-degenerate matrix.*

- *For an arbitrary real number $y \in \mathbb{R}$, we define set $\mathrm{dom}(y) := \{y \in \mathbb{R} : l_i < y < u_i\}$.*

- *For vector $x \in \mathbb{R}^m$ and all $i \in [m]$, the set $\mathrm{dom}(x_i)$ is neither the empty set nor the entire real line.*

- *For $i \in [m]$, $l_i \in \mathbb{R} \cup \{-\infty\}$ and $u_i \in \mathbb{R} \cup \{+\infty\}$.*

- *Assume the interior of the polytope $\Omega^\circ := \{x \in \mathbb{R}^m : A^\top x = b, l_i < x_i < u_i, \forall i \in [m]\}$ is not empty.*

*We define the following linear program:*

$$\mathrm{OPT} := \min_{\substack{x \in \mathbb{R}^m \,:\, A^\top x = b \\ \forall i \in [m] \,:\, l_i \leq x_i \leq u_i}} c^\top x.$$

## D.2    SELF-CONCORDANCE

**Definition D.2** (Self-concordance, Definition 4 in page 12 in (Nesterov & Nemirovskii, 1994))**.** *Let $\phi : K \to \mathbb{R}^n$ be a convex, thrice continuously differentiable function. If the following conditions hold:*

- $\lim_{i \to \infty} \phi(x_i) \to \infty$ *for all sequences $x_i \in K$ converging to the boundary of $K$.*

- $|D^3\phi(x)[h, h, h]| \leq 2|D^2\phi(x)[h, h]|^{3/2}$ *for all $x \in K$ and $h \in \mathbb{R}^n$.*

- $|D\phi(x)[h]| \leq \sqrt{\nu}|D^2\phi(x)[h, h]|^{1/2}$ *for all $x \in K$ and $h \in \mathbb{R}^n$.*

*Then the function $\phi$ is a $\nu$-self-concordant barrier function for open convex set $K \subset \mathbb{R}^n$.*

**Lemma D.3** (Theorem 4.1.6 in page 182 in (Nesterov, 2003))**.** *If the following conditions hold:*

- *Let $\phi_i''$ denote the second derivative of $\phi_i : \mathbb{R}^m \to \mathbb{R}$, for all $i \in [m]$.*

- *Let $s \in \mathrm{dom}(\phi_i)$ for $i \in [m]$.*

- *Define $r := \max_{i \in [m]} \sqrt{\phi_i''(s)}|s - t|$.*

- *Let $U$ denote the maximum diameter of all $\mathrm{dom}(\phi_i)$.*

*Then, we have*

- **Part 1.** $r \in (0, 1)$.

- **Part 2.** $t \in \mathrm{dom}(\phi_i)$ and $(1 - r)\sqrt{\phi_i''(s)} \leq \sqrt{\phi_i''(t)} \leq (1 - r)^{-1}\sqrt{\phi_i''(s)}$.

- **Part 3.** $\sqrt{\phi_i''(s)} \geq 1/U$ where $U$ is the diameter of $\mathrm{dom}(\phi_i)$.

**Lemma D.4** (Theorem 4.2.4 in page 196 in (Nesterov, 2003), see Lemma 9 in page 10 in (Lee & Sidford, 2019) as an example). *For all $x, y \in \text{dom}(\phi_i)$ and $i \in [m]$, we have $\phi_i'(x) \cdot (y - x) \leq 1$.*

For all $x \in \Omega^\circ$, we define $\phi(x) \in \mathbb{R}^m$ by $\phi(x)_i := \phi_i(x_i)$ for $i \in [m]$, define $\phi'(x), \phi''(x)$ and $\phi'''(x)$ analogously, for example $\phi(x)_i' := \phi_i(x_i)$ and let $\Phi' = \text{diag}(\phi'), \Phi'' = \text{diag}(\phi''), \Phi''' = \text{diag}(\phi''')$ denote their associated diagonal matrices.

### D.3 WEIGHTED CENTRAL PATH

**Definition D.5** (Weighted central path, Implicit in page 11 in (Lee & Sidford, 2019)). *Let $\phi_i : \mathbb{R} \to \mathbb{R}$. We define the penalized objective function as*

$$f_t(x, w) := t \cdot c^\top x + \sum_{i \in [m]} w_i \phi_i(x_i).$$

*The path-finding algorithm maintains a feasible point $x \in \Omega^\circ$, weights $w \in \mathbb{R}_{>0}^m$ and minimizes the penalized objective function for increasing $t$ and small $w$*

$$\min_{A^\top x = b} f_t(x, w).$$

*For every fixed set of weights, $w \in \mathbb{R}_{>0}^m$ the set of points $x_w(t) := \arg\min_{x \in \Omega^\circ} f_t(x, w)$ for $t \in [0, \infty)$ form a path through the interior of the polytope that we call the weighted central path. We call $x_w(0)$ a weighted center of $\Omega^\circ$.*

As shown in Theorem 4.2.7 on page 200 of (Nesterov, 2003), $\lim_{t \to \infty} x_w(t)$ is a solution to the linear program in Definition D.1.

The above definition (Definition D.5) can trivially yield the following fact.

**Fact D.6** (Folklore). *If the following condition holds*

- *Let $f_t(x, w)$ be defined as Definition D.5.*

*Then we can show*

$$\nabla_x f_t(x, w) = t \cdot c + w\phi'(x)$$
$$\nabla_{xx}^2 f_t(x, w) = W\Phi''(x).$$

**Fact D.7.** *If the following conditions hold*

- *We have $w(v) := \arg\min_{w \in \mathbb{R}_{>0}^m} f(v, w)$.*

- *Let $f(x, w)$ be defined as Definition D.5 (We treat $t$ as a fixed parameter in this statement, thus ignore it).*

- *The minimizer for function $f(v, w)$ is in the interior of its domain.*

*Then we have*

$$J_w(v) = -(\nabla_{w,w}^2 f(v, w(v)))^{-1} \nabla_{w,v}^2 f(w, w(v)).$$

*Proof.* Since the optimal for $f(v, w)$ is in the interior and the optimality condition $\nabla_w f(v, w(v)) = 0$ holds, we can take derivative w.r.t. $v$ on both sides and obtain

$$\nabla_{w,v}^2 f(v, w(v)) + \nabla_{w,w}^2 f(v, w(v)) J_w(v) = 0.$$

Then, solving for $J_w(v)$ directly yields:

$$J_w(v) = -(\nabla_{w,w}^2 f(v, w(v)))^{-1} \nabla_{w,v}^2 f(w, w(v)).$$

This finishes the proof.

$\square$

## D.4 NEWTON STEP

**Lemma D.8** (Lemma 51 in page 49 in (Lee & Sidford, 2019)). *If the following conditions hold:*

- *Define $f(x) := v^\top x + 0.5\|x\|_W^2$.*

- *Define $x_* := -W^{-1}v + W^{-1}A(A^\top W^{-1}A)^{-1}A^\top W^{-1}v \in \mathbb{R}^m$.*

*Then, for any vector $v \in \mathbb{R}^m$, any positive vector $w \in \mathbb{R}^m$ and matrix $A \in \mathbb{R}^{m \times n}$, we have*

$$\arg \min_{A^\top x = 0} f(x) = x_*.$$

*Proof.* A point $x \in \ker(A^\top)$ is optimal if and only if the gradient $\nabla f(x) = v + Wx$ is orthogonal to every feasible direction in $\ker(A^\top)$.

Equivalently,

$$v + Wx \in \mathrm{Im}(A).$$

Then, there exists some vector $y \in \mathbb{R}^m$ such that $v + Wx = Ay$.

Thus,

$$x = W^{-1}(Ay - v). \tag{9}$$

Let us left multiply $A$ on both sides,

$$A^\top x = A^\top W^{-1}(Ay - v).$$

Since $A^\top x = 0$, so

$$A^\top W^{-1}Ay - A^\top W^{-1}v = 0.$$

We have

$$y = (A^\top W^{-1}A)^{-1}A^\top W^{-1}v.$$

Substituting back to Eq. (9), we have

$$x_* := -W^{-1}v + W^{-1}A(A^\top W^{-1}A)^{-1}A^\top W^{-1}v.$$

The proof is complete. $\square$

**Definition D.9.** *For notation convenience in Hessian computation, we define:*

- *$A_x := \Phi''(x)^{-1/2}A$.*

- *$P_{x,w} := I - W^{-1}A_x(A_x^\top W A_x)^{-1}A_x^\top$.*

**Lemma D.10** (Newton step, Implicit in page 11 in (Lee & Sidford, 2019)). *If the following conditions hold:*

- *Let $f_t(x, w)$ be defined in Definition D.5.*

- *Let $\Phi(x) \in \mathbb{R}^{m \times m}$ denote a diagonal matrix where $i$-th entry is $\phi_i(x)$.*

- *Let $A_x$ and $P_{x,w}$ be defined as Definition D.9.*

*Then, the new newton step for $x$ with respect to $f_t(x, w)$ is*

$$h_t(x, w) = -\Phi''(x)^{-1/2}P_{x,w}W^{-1}\Phi''(x)^{-1/2}\nabla_x f_t(x, w).$$

*Proof.* Lemma D.8 (with replacing the $W$ by $W\Phi''(x)$ and $v$ by gradient in $x_*$ definition) shows that a Newton step for $x$ is given by

$$h_t(x, w) = -(I - (W\Phi''(x))^{-1}A(A^\top(W\Phi''(x))^{-1}A)^{-1}A^\top)(W\Phi''(x))^{-1}\nabla_x f_t(x, w)$$
$$= -\Phi''(x)^{-1/2}P_{x,w}W^{-1}\Phi''(x)^{-1/2}\nabla_x f_t(x, w),$$

where the second step follows from definition of $A_x, P_{x,w}$. $\square$

## D.5 WEIGHT FUNCTION

**Definition D.11** ($(c_i, c_\gamma, c_k)$-Weight Function, Definition 12 in page 13 in (Lee & Sidford, 2019)). *We say a function differentiable $g : \Omega^\circ \to \mathbb{R}^m_{>0}$ is a $(c_i, c_\gamma, c_k)$-weight function if the following conditions hold:*

- *The size, $c_1$, satisfies $c_1 \geq \max\{1, \|g(x)\|_1\}$. This bounds how quickly centrality changes as $t$ changes.*

- *The sensitivity, $c_s$, satisfies $c_s \geq e_i^\top G(x)^{-1} A_x (A_x^\top G(x)^{-1} A_x)^{-1} A_x^\top G(x)^{-1} e_i$. This bounds how quickly the Hessian changes as $x$ changes. Here $e_i$ is the length-$m$ vector where $i$-th location $1$ and $0$ everywhere else.*

- *The consistency, $c_k$, satisfies $\|G(x)^{-1} J_g(x)(\Phi''(x))^{-1/2}\|_{g(x)+\infty} \leq 1 - c_k^{-1} < 1$. This bounds how much the weights change as $x$ changes, thereby governing how consistent the weights are with changes to $x$ along the weighted central path. For definition of $J_g(x)$, please see Fact D.7.*

## D.6 CENTRALITY

Here, we explain how we measure the distance from $x$ to the minimum of $f_t(x, w)$ for fixed $w$, denoted $\delta_t(x, w)$. As $\delta_t(x, w)$ measures the proximity of $x$ to the weighted central path, we call it a centrality measure of $x$ and $w$.

**Definition D.12** (Mixed Norm, Implicit in page 12 in (Lee & Sidford, 2019)). *We define the mixed norm for all $y \in \mathbb{R}^m$ by*

$$\|y\|_{w+\infty} := \|y\|_\infty + C_{\mathrm{norm}} \|y\|_w.$$

**Definition D.13** (Centrality Measure, Definition 11 in page 12 in (Lee & Sidford, 2019)). *Let $P_{x,w}$ be defined as Definition D.9. For $\{x, w\} \in \{\Omega^\circ \times \mathbb{R}^m_{>0}\}$ and $t \geq 0$, we let $h_t(x, w)$ denote the projected Newton step for $x$ on the penalized objective $f_t$ given by*

$$h_t(x, w) := -\frac{1}{\sqrt{\phi''(x)}} P_{x,w}\left(\frac{\nabla_x f_t(x, w)}{w\sqrt{\phi''(x)}}\right).$$

*We measure the centrality of $\{x, w\}$ by*

$$\delta_t(x, w) := \min_{\eta \in \mathbb{R}^n} \left\|\frac{\nabla_x f_t(x, w) - A\eta}{w\sqrt{\phi''(x)}}\right\|_{w+\infty},$$

*where for all $y \in \mathbb{R}^m$, let $\|y\|_{w+\infty} := \|y\|_\infty + C_{\mathrm{norm}}\|y\|_W$ for $C_{\mathrm{norm}} > 0$ is defined in Definition D.12.*

**Lemma D.14** (Lemma 10 in page 12 in (Lee & Sidford, 2019)). *If the following conditions hold:*

- *Define $A_x := \Phi''(x)^{-1/2} A \in \mathbb{R}^{m \times n}$.*

- *Define $P_{x,w} := I - W^{-1} A_x (A_x^\top W^{-1} A_x)^{-1} A_x^\top \in \mathbb{R}^{m \times m}$.*

- *For any norm $\|\cdot\|$, we define Q-norm which is $\|y\|_Q := \min_{\eta \in \mathbb{R}^n} \|y - \frac{A\eta}{w\sqrt{\phi''(x)}}\|$.*

*Then, we have*

- **Part 1.** $\|y\|_Q \leq \|P_{x,w} y\| \leq \|P_{x,w}\| \cdot \|y\|_Q$.

- **Part 2.** *For all $\{x, w\} \in \{\Omega^\circ \times \mathbb{R}^m_{>0}\}$, we have*

$$\delta_t(x, w) \leq \|\sqrt{\phi''(x)} h_t(x, w)\|_{w+\infty} \leq \|P_{x,w}\|_{w+\infty} \cdot \delta(x, w).$$

*Proof.* **Proof of Part 1.**

We can show

$$
\begin{aligned}
P_{x,w}y &= (I - W^{-1}A_x(A_x^\top W^{-1}A_x)^{-1}A_x^\top)y \\
&= y - W^{-1}A_x(A_x^\top W^{-1}A_x)^{-1}A_x^\top y \\
&= y - W^{-1}\Phi''^{-1/2}A \cdot (A_x^\top W^{-1}A_x)^{-1}A_x^\top y \\
&= y - W^{-1}\Phi''^{-1/2}A \cdot \eta_y \\
&= y - \frac{A\eta_y}{w\sqrt{\phi''(x)}},
\end{aligned}
\tag{10}
$$

where the first step follows definition of $P_{x,w}$, the second step follows simple algebra, the third step follows from definition of $A_x$, the forth step follows from letting $\eta_y = (A_x^\top W^{-1}A_x)^{-1}A_x^\top y$, the last step follows from $W$ is diagonal matrix $w$, similarly for $\Phi''$ and $\phi''$.

We have

$$
\begin{aligned}
\|y\|_Q &= \min_{\eta\in\mathbb{R}^n}\|y - \frac{A\eta}{w\sqrt{\phi''(x)}}\| \\
&\le \|y - \frac{A\eta_y}{w\sqrt{\phi''(x)}}\| \\
&= \|P_{x,w}y\|,
\end{aligned}
$$

where the first step follows from the definition of $\|y\|_Q$, the second step follows from $\eta_y$ can not achieve a smaller objective function value than the minimizer, the third step follows from $P_{x,w}y = y - \frac{A\eta_y}{w\sqrt{\phi''(x)}}$ for some $\eta_y \in \mathbb{R}^n$ (see Eq. (10)).

We have

$$
\begin{aligned}
\|P_{x,w}\| \cdot \|y\|_Q &= \|P_{x,w}\| \cdot \|y - \frac{A\eta_q}{w\sqrt{\phi''}}\| \\
&\ge \|P_{x,w}(y - \frac{A\eta_q}{w\sqrt{\phi''}})\| \\
&= \|P_{x,w}y\|,
\end{aligned}
$$

where the first step follows from letting $\eta_q$ be such that $\|y\|_Q = \|y - \frac{A\eta_q}{w\sqrt{\phi''(x)}}\|$, the second step follows from the property of spectral norm, and the last step follows from $P_{x,w}W^{-1}(\Phi'')^{-1/2}A = 0$.

Thus, with $y = \nabla_x f_t(x, w)$, the proof is complete.

**Proof of Part 2.**

We choose $y = \frac{\nabla_x f_t(x,w)}{w\sqrt{\phi''(x)}}$.

Then, we have

$$
\begin{aligned}
\delta_t(x,w) &= \min_{\eta\in\mathbb{R}^n}\|y - \frac{A\eta}{w\sqrt{\phi''(x)}}\|_{w+\infty} \\
&= \|y\|_Q \\
&\le \|P_{x,w}y\|_{w+\infty} \\
&= \|\sqrt{\phi''(x)}h_t(x,w)\|_{w+\infty} \\
&\le \|P_{x,w}\|_{w+\infty}\|y\|_Q \\
&= \|P_{x,w}\|_{w+\infty}\delta_t(x,w),
\end{aligned}
$$

where the first step follows from the definition of $\delta_t(x, w)$, the second step follows from the definition of $\|y\|_Q$, the third step follows from Part 1, the fourth step follows from the definition of $h_t(x, w)$, the fifth step follows from Part 1, and the last step follows from $\delta_t(x, w) = \|y\|_Q$.

Thus, the proof is complete. $\square$

## D.7 DERIVATIVE OF VOLUMETRIC BARRIER

**Lemma D.15** (Derivative of Volumetric Barrier, Lemma 48 in page 47 in (Lee & Sidford, 2019))**.**
*If the following conditions hold:*

- *For a vector $w \in \mathbb{R}^m$, let us $W := \mathrm{diag}(w)$ denote the $m \times m$ size diagonal matrix.*

- *For a full rank matrix $A \in \mathbb{R}^{n \times m}$, we define*

$$f(w) := \log \det(A^\top W A).$$

*Then, for any $w \in \mathbb{R}^m_{>0}$, we have*

$$\nabla f(w) = W^{-1} \sigma(W^{1/2} A).$$

*Proof.* We have that for all $i \in [m]$

$$
\begin{aligned}
\frac{\mathrm{d}f(w)}{\mathrm{d}w_i} &= \mathrm{tr}[(A^\top W A)^{-1} \frac{\mathrm{d}}{\mathrm{d}w_i}(A^\top W A)] \\
&= \mathrm{tr}[(A^\top W A)^{-1} A^\top e_i e_i^\top A] \\
&= \mathrm{tr}[e_i^\top A (A^\top W A)^{-1} A^\top e_i] \\
&= e_i^\top A (A^\top W A)^{-1} A^\top e_i \\
&= e_i^\top W^{-1/2} W^{1/2} A (A^\top W A)^{-1} A^\top W^{1/2} W^{-1/2} e_i \\
&= w_i^{-1} \sigma(W^{1/2} A)_i \\
&= (w^{-1} \circ \sigma(W^{1/2} A))_i \\
&= (W^{-1} \sigma(W^{1/2} A))_i,
\end{aligned}
$$

where the first step follows from Fact C.2, the second step follows from $W$ is a diagonal matrix and $\frac{\mathrm{d}W}{\mathrm{d}w_i} = e_i e_i^\top$, the third step follows from the cyclic property of trace $(\mathrm{tr}[ABC] = \mathrm{tr}[CAB])$, the fourth step follows from basic algebra, the fifth step follows from basic algebra, the sixth step follows from the definition of $\sigma(W^{1/2} A)$, the seventh step follows from the Hadamard product, and the last step follows from Fact C.1.

Therefore, we have

$$\nabla f(w) = W^{-1} \sigma(W^{1/2} A).$$

$\square$

## D.8 POTENTIAL FUNCTION DERIVATIVE

**Lemma D.16** (Potential Function Derivative, Lemma 50 in page 48 in (Lee & Sidford, 2019))**.** *If the following conditions hold:*

- *Let $A \in \mathbb{R}^{m \times n}$ be a non-degenerate matrix.*

- *Let $A_{*,i}$ denote the $i$-th column of $A$ for all $i \in [n]$.*

- *Let $q > 0$ with $q \neq 2$.*

- *Define $A_x := S_x^{-1} A$.*

- *$S_x := \mathrm{diag}(Ax - b)$.*

- *Let $u_i = (1/2 - 1/q)(e_i \circ w^{-1})$.*

- *For all $x \in \mathbb{R}^n$ with $Ax > b$ and all $w \in \mathbb{R}^m_{>0}$, let $p(x, w) := \log \det(A_x^\top W^{1-2/q} A_x)$.*

- *Define $c_q := 1 - \frac{2}{q}$.*

- *Let $B_x = W^{1/2-1/q}A_x$.*

- *$\sigma_{x,w} := \sigma(B_x)$.*

- *$\Sigma_{x,w} := \Sigma(B_x)$.*

- *$\Lambda_{x,w} := \Lambda(B_x)$.*

- *We recall the preliminary that $P^{\circ 2} := P \circ P$.*

- *We recall the preliminary that $P(A) := A(A^\top A)^{-1}A^\top$.*

- *We recall the preliminary that $\Lambda(A) := \Sigma(A) - P^{\circ 2}(A)$.*

*Then we have*

- **Part 1.** $\frac{\mathrm{d}Ax}{\mathrm{d}x_i} = A_{*,i}$.

- **Part 2.** $\frac{\mathrm{d}S_x}{\mathrm{d}x_i} = \mathrm{diag}(A_{*,i})$.

- **Part 3.** $\frac{\mathrm{d}S_x^{-1}}{\mathrm{d}x_i} = -S_x^{-1}\mathrm{diag}(A_{x,*,i})$.

- **Part 4.** $\frac{\mathrm{d}A_x}{\mathrm{d}x_i} = -\mathrm{diag}(A_{x,*,i})A_x$.

- **Part 5** $\frac{\mathrm{d}A_{x,*,j}}{\mathrm{d}x_i} = -\mathrm{diag}(A_{x,*,i})A_{x,*,j}$.

- **Part 6** $\frac{\mathrm{d}B_x}{\mathrm{d}x_i} = -\mathrm{diag}(A_{x,*,i})B_x$.

- **Part 7** $\frac{\mathrm{d}B_x}{\mathrm{d}w_i} = \mathrm{diag}(u_i)B_x$.

- **Part 8.** $\frac{\mathrm{d}B_x^\top B_x}{\mathrm{d}x_i} = -2B_x^\top \mathrm{diag}(A_{x,*,i})B_x$.

- **Part 9.** $\frac{\mathrm{d}B_x^\top B_x}{\mathrm{d}w_i} = 2B_x^\top \mathrm{diag}(u_i)B_x$.

- **Part 10.** $\frac{\mathrm{d}(B_x^\top B_x)^{-1}}{\mathrm{d}x_i} = 2(B_x^\top B_x)^{-1}B_x^\top \mathrm{diag}(A_{x,*,i})B_x(B_x^\top B_x)^{-1}$.

- **Part 11** $\frac{\mathrm{d}(B_x^\top B_x)^{-1}}{\mathrm{d}w_i} = -2(B_x^\top B_x)^{-1}(B_x^\top \mathrm{diag}(u_i)B_x)(B_x^\top B_x)^{-1}$.

- **Part 12** $\frac{\mathrm{d}P(B_x)}{\mathrm{d}x_i} = -\mathrm{diag}(A_{x,*,i})P(B_x) + 2P(B_x)\mathrm{diag}(A_{x,*,i})P(B_x) - P(B_x)\mathrm{diag}(A_{x,*,i})$.

- **Part 13** $\frac{\mathrm{d}P(B_x)}{\mathrm{d}w_i} = \mathrm{diag}(u_i)P(B_x) - 2P(B_x)\mathrm{diag}(u_i)P(B_x) + P(B_x)\mathrm{diag}(u_i)$.

- **Part 14** $\frac{\mathrm{d}\sigma_{x,w}}{\mathrm{d}x_i} = -2\Lambda_{x,w}A_{x,*,i}$.

- **Part 15.** $\frac{\mathrm{d}\sigma_{x,w}}{\mathrm{d}w_i} = 2\Lambda_{x,w}u_i$.

- **Part 16.** $\nabla_x p(x,w) = -2A_x^\top \sigma_{x,w}$.

- **Part 17.** $\nabla_w p(x,w) = c_q W^{-1}\sigma_{x,w}$.

- **Part 18.** $\nabla_{xx}^2 p(x,w) = A_x^\top(2\Sigma_{x,w} + 4\Lambda_{x,w})A_x$.

- **Part 19.** $\nabla_{ww}^2 p(x,w) = -c_q W^{-1}(\Sigma_{x,w} - c_q\Lambda_{x,w})W^{-1}$.

- **Part 20.** $\nabla_{xw}^2 p(x,w) = -2c_q A_x^\top \sigma_{x,w}W^{-1}$.

- **Part 21.** $\frac{\mathrm{d}\sigma_{x,w}}{\mathrm{d}q} = \Lambda((1-2/q)W^{-1}\frac{\mathrm{d}w}{\mathrm{d}q} + \frac{2}{q^2}\log w)$.

*Proof.* **Proof of Part 1.**

We can show

$$
\underbrace{\frac{\mathrm{d}Ax}{\mathrm{d}x_i}}_{m\times 1} = \underbrace{A}_{m\times n}\underbrace{\frac{\mathrm{d}x}{\mathrm{d}x_i}}_{n\times 1}
$$
$$
= \underbrace{A}_{m\times n}\underbrace{e_i}_{n\times 1}
$$
$$
= A_{*,i},
$$

where the first step follows from the irrelevance between $A$ and $x$, the second step follows from basic algebra, and the last step follows from basic algebra.

**Proof of Part 2.**

We can show

$$
\frac{\mathrm{d}S_x}{\mathrm{d}x_i} = \frac{\mathrm{d}\operatorname{diag}(Ax-b)}{\mathrm{d}x_i}
$$
$$
= \frac{\mathrm{d}\operatorname{diag}(Ax)}{\mathrm{d}x_i}
$$
$$
= \operatorname{diag}(\frac{\mathrm{d}Ax}{\mathrm{d}x_i})
$$
$$
= \operatorname{diag}(A_{*,i}),
$$

where the first step follows from the definition of $S_x$, the second step follows from the irrelevance between $b$ and $x$, the third step follows from the linearity of $\operatorname{diag}(\cdot)$, and the last step follows from Part 1.

**Proof of Part 3.**

We can show

$$
\frac{\mathrm{d}S_x^{-1}}{\mathrm{d}x_i} = -S_x^{-1}\frac{\mathrm{d}S_x}{\mathrm{d}x_i}S_x^{-1}
$$
$$
= -S_x^{-2}\frac{\mathrm{d}S_x}{\mathrm{d}x_i}
$$
$$
= -S_x^{-2}\operatorname{diag}(A_{*,i})
$$
$$
= -S_x^{-1}\operatorname{diag}(A_{x,*,i}),
$$

where the first step follows from Fact C.2, the second step follows from the fact that $S_x$ is a diagonal matrix, the third step follows from Part 2, and the last step follows from $A_x = S_x^{-1}A$ (implies that $(A_x)_{*,i} = (S_x^{-1}A)_{*,i} = S_x^{-1}A_{*,i}$).

**Proof of Part 4.**

We can show

$$
\frac{\mathrm{d}A_x}{\mathrm{d}x_i} = \frac{\mathrm{d}(S_x^{-1}A)}{\mathrm{d}x_i}
$$
$$
= \frac{\mathrm{d}S_x^{-1}}{\mathrm{d}x_i}A
$$
$$
= -S_x^{-1}\operatorname{diag}(A_{x,*,i})A
$$
$$
= -\operatorname{diag}(A_{x,*,i})S_x^{-1}A
$$
$$
= -\operatorname{diag}(A_{x,*,i})A_x,
$$

where the first step follows from the definition of $A_x$, the second step follows from the irrelevance between $A$ and $x$, the third step follows from Part 3, the fourth step follows from the fact that $S_x$ is a diagonal matrix, and the last step follows from the definition of $A_x$.

**Proof of Part 5.**

$$\frac{\mathrm{d}A_{x,*,j}}{\mathrm{d}x_i} = (\frac{\mathrm{d}A_x}{\mathrm{d}x_i})_{*,j}$$
$$= (-\operatorname{diag}(A_{x,*,i})A_x)_{*,j}$$
$$= -\operatorname{diag}(A_{x,*,i})A_{x,*,j},$$

where the first step follows from selecting the $j$-th column of the derivative, the second step follows from Part 4 of the proof, and the last step follows from basic algebra.

**Proof of Part 6.**

We have

$$\frac{\mathrm{d}B_x}{\mathrm{d}x_i} = \frac{\mathrm{d}W^{1/2-1/q}A_x}{\mathrm{d}x_i}$$
$$= W^{1/2-1/q}\frac{\mathrm{d}A_x}{\mathrm{d}x_i}$$
$$= -W^{1/2-1/q}\operatorname{diag}(A_{x,*,i})A_x$$
$$= -\operatorname{diag}(A_{x,*,i})B_x,$$

where the first step follows from the definition of $B_x$, the second step follows from the irrelevance between $W$ and $x_i$, the third step follows from Part 4, the last step follows from the commutative property of the products of diagonal matrices and the definition of $B_x$.

**Proof of Part 7.**

We have

$$\frac{\mathrm{d}B_x}{\mathrm{d}w_i} = \frac{\mathrm{d}W^{1/2-1/q}A_x}{\mathrm{d}w_i}$$
$$= (1/2-1/q)\operatorname{diag}(e_i)W^{-1/2-1/q}A_x$$
$$= (1/2-1/q)\operatorname{diag}(e_i \circ w^{-1})B_x$$
$$= \operatorname{diag}(u_i)B_x,$$

where the first step follows from the definition of $B_x$, the second step follows from $\frac{\mathrm{d}W}{\mathrm{d}w_i} = \operatorname{diag}(e_i)$, and the third step follows from the definition of $B_x$, the last step follows from definition of $u_i$.

**Proof of Part 8.**

We can show

$$\frac{\mathrm{d}B_x^\top B_x}{\mathrm{d}x_i} = \frac{\mathrm{d}B_x^\top}{x_i}B_x + B_x^\top\frac{\mathrm{d}B_x}{x_i}$$
$$= -B_x^\top\operatorname{diag}(A_{x,*,i})B_x - B_x^\top\operatorname{diag}(A_{x,*,i})B_x$$
$$= -2B_x^\top\operatorname{diag}(A_{x,*,i})B_x,$$

where the first step follows from the product rule, the second step follows from Part 6, and the last step follows from basic algebra.

**Proof of Part 9.**

We have

$$\frac{\mathrm{d}B_x^\top B_x}{\mathrm{d}w_i} = \frac{\mathrm{d}B_x^\top}{w_i}B_x + B_x^\top\frac{\mathrm{d}B_x}{w_i}$$
$$= (\operatorname{diag}(u_i)B_x)^\top B_x + B_x^\top\operatorname{diag}(u_i)B_x$$
$$= 2B_x^\top\operatorname{diag}(u_i)B_x,$$

where the first step follows from the chain rule for products, the second step follows from Part 7, and the last step follows from the fact that $W$ and $\operatorname{diag}(e_i)$ can commute.

**Proof of Part 10.**

We can show

$$
\frac{\mathrm{d}(B_x^\top B_x)^{-1}}{\mathrm{d}x_i} = -(B_x^\top B_x)^{-1}\frac{\mathrm{d}(B_x^\top B_x)}{x_i}(B_x^\top B_x)^{-1}
$$
$$
= 2(B_x^\top B_x)^{-1}B_x^\top \operatorname{diag}(A_{x,*,i})B_x(B_x^\top B_x)^{-1},
$$

where the first step follows from $\frac{\mathrm{d}A^{-1}}{\mathrm{d}t} = -A^{-1}\frac{\mathrm{d}A}{\mathrm{d}t}A^{-1}$ (Fact C.2), and the last step follows from Part 8.

**Proof of Part 11.**

We have

$$
\frac{\mathrm{d}(B_x^\top B_x)^{-1}}{\mathrm{d}w_i} = -(B_x^\top B_x)^{-1}\frac{\mathrm{d}(B_x^\top B_x)}{\mathrm{d}w_i}(B_x^\top B_x)^{-1}
$$
$$
= -2(B_x^\top B_x)^{-1}(B_x^\top \operatorname{diag}(u_i)B_x)(B_x^\top B_x)^{-1},
$$

where the first step follows from $\frac{\mathrm{d}A^{-1}}{\mathrm{d}t} = -A^{-1}\frac{\mathrm{d}A}{\mathrm{d}t}A^{-1}$ (Fact C.2), the second step follows from Part 9.

**Proof of Part 12.**

We have

$$
\frac{\mathrm{d}P(B_x)}{\mathrm{d}x_i} = \frac{\mathrm{d}B_x(B_x^\top B_x)^{-1}B_x^\top}{\mathrm{d}x_i}
$$
$$
= \frac{\mathrm{d}B_x}{\mathrm{d}x_i}(B_x^\top B_x)^{-1}B_x^\top + B_x\frac{\mathrm{d}(B_x^\top B_x)^{-1}}{\mathrm{d}x_i}B_x^\top + B_x(B_x^\top B_x)^{-1}\frac{\mathrm{d}B_x^\top}{\mathrm{d}x_i}
$$
$$
= -\operatorname{diag}(A_{x,*,i})B_x(B_x^\top B_x)^{-1}B_x^\top
$$
$$
+ B_x\frac{\mathrm{d}(B_x^\top B_x)^{-1}}{\mathrm{d}x_i}B_x^\top
$$
$$
- B_x(B_x^\top B_x)^{-1}B_x^\top \operatorname{diag}(A_{x,*,i})
$$
$$
= -\operatorname{diag}(A_{x,*,i})B_x(B_x^\top B_x)^{-1}B_x^\top
$$
$$
+ B_x(2(B_x^\top B_x)^{-1}B_x^\top \operatorname{diag}(A_{x,*,i})B_x(B_x^\top B_x)^{-1})B_x^\top
$$
$$
- B_x(B_x^\top B_x)^{-1}B_x^\top \operatorname{diag}(A_{x,*,i})
$$
$$
= -\operatorname{diag}(A_{x,*,i})P(B_x) + 2P(B_x)\operatorname{diag}(A_{x,*,i})P(B_x) - P(B_x)\operatorname{diag}(A_{x,*,i}),
$$

where the first step follows from the definition of $P(B_x)$, the second step follows from the product rule, the third step follows from Part 6, the fourth step follows from Part 10, and the last step follows from the definition of $P(B_x)$.

**Proof of Part 13**

For the convenience of writing proofs, we recall $u_i = (1/2 - 1/p)(e_i \circ w^{-1})$.

We have

$$
\frac{\mathrm{d}P(B_x)}{\mathrm{d}w_i} = \frac{\mathrm{d}B_x(B_x^\top B_x)^{-1}B_x^\top}{\mathrm{d}w_i}
$$
$$
= \frac{\mathrm{d}B_x}{\mathrm{d}w_i}(B_x^\top B_x)^{-1}B_x^\top + B_x\frac{\mathrm{d}(B_x^\top B_x)^{-1}}{\mathrm{d}w_i}B_x^\top + B_x(B_x^\top B_x)^{-1}\frac{\mathrm{d}B_x^\top}{\mathrm{d}w_i}
$$
$$
= \operatorname{diag}(u_i)B_x(B_x^\top B_x)^{-1}B_x^\top
$$
$$
+ B_x\frac{\mathrm{d}(B_x^\top B_x)^{-1}}{\mathrm{d}w_i}B_x^\top
$$
$$
+ B_x(B_x^\top B_x)^{-1}(\operatorname{diag}(u_i)B_x)^\top
$$

$$
\begin{aligned}
&= \operatorname{diag}(u_i)B_x(B_x^\top B_x)^{-1}B_x^\top \\
&\quad - B_x(2(B_x^\top B_x)^{-1}(B_x^\top \operatorname{diag}(u_i)B_x)(B_x^\top B_x)^{-1})B_x^\top \\
&\quad + B_x(B_x^\top B_x)^{-1}(\operatorname{diag}(u_i)B_x)^\top \\
&= \operatorname{diag}(u_i)P(B_x) \\
&\quad - 2P(B_x)\operatorname{diag}(u_i)P(B_x) \\
&\quad + P(B_x)\operatorname{diag}(u_i),
\end{aligned}
$$

where the first step follows from the definition of $P(B_x)$, the second step follows from the chain rule for product, the third step follows from Part 7, the fourth step follows from Part 11, and the last step follows from the definition of $P(B_x)$.

**Proof of Part 14.**

We can show that

$$
\begin{aligned}
\frac{\mathrm{d}\sigma_{x,w}}{\mathrm{d}x_i} &= \frac{\mathrm{d}\operatorname{Diag}(P(B_x))}{\mathrm{d}x_i} \\
&= \operatorname{Diag}(\frac{\mathrm{d}P(B_x)}{\mathrm{d}x_i}) \\
&= \operatorname{Diag}(-\operatorname{diag}(A_{x,*,i})P(B_x) + 2P(B_x)\operatorname{diag}(A_{x,*,i})P(B_x) - P(B_x)\operatorname{diag}(A_{x,*,i})) \\
&= -2\operatorname{diag}(A_{x,*,i})\sigma_{x,w} + 2\operatorname{Diag}(P(B_x)\operatorname{diag}(A_{x,*,i})P(B_x)), \\
&= -2\operatorname{diag}(A_{x,*,i})\sigma_{x,w} + 2(P(B_x)\circ P(B_x))A_{x,*,i} \\
&= -2\Sigma_{x,w}A_{x,*,i} + 2(P(B_x)\circ P(B_x))A_{x,*,i} \\
&= -2\Lambda_{x,w}A_{x,*,i},
\end{aligned}
$$

where the first step follows from the definition of $\sigma_{x,w}$, the second step follows from the linearity of $\operatorname{Diag}(\cdot)$, the third step follows from Part 12, the fourth follows from Fact C.1, the fifth step follows from Fact C.1, the sixth step follows from Fact C.1, and the last step follows from the definition of $\Lambda_{x,w}$.

**Proof of Part 15.**

We have

$$
\begin{aligned}
\frac{\mathrm{d}\sigma_{x,w}}{\mathrm{d}w_i} &= \frac{\mathrm{d}\operatorname{Diag}(P(B_x))}{\mathrm{d}w_i} \\
&= \operatorname{Diag}(\frac{\mathrm{d}P(B_x)}{\mathrm{d}w_i}) \\
&= \operatorname{Diag}(\operatorname{diag}(u_i)P(B_x) - 2P(B_x)\operatorname{diag}(u_i)P(B_x) + P(B_x)\operatorname{diag}(u_i)) \\
&= \operatorname{Diag}(\operatorname{diag}(u_i)P(B_x)) \\
&\quad - 2\operatorname{Diag}(P(B_x)\operatorname{diag}(u_i)P(B_x)) \\
&\quad + \operatorname{Diag}(P(B_x)\operatorname{diag}(u_i)) \\
&= \operatorname{diag}(u_i)\sigma_{x,w} - 2\operatorname{Diag}(P(B_x)\operatorname{diag}(u_i)P(B_x)) + \operatorname{diag}(u_i)\sigma_{x,w} \\
&= 2(\operatorname{diag}(u_i)\sigma_{x,w} - \operatorname{Diag}(P(B_x)\operatorname{diag}(u_i)P(B_x))) \\
&= 2(\operatorname{diag}(u_i)\sigma_{x,w} - P(B_x)\circ P(B_x)u_i) \\
&= 2(\Sigma_{x,w}u_i - P(B_x)\circ P(B_x)u_i) \\
&= 2\Lambda_{x,w}u_i,
\end{aligned} \tag{11}
$$

where the first step follows from the definition of $\sigma_{x,w}$, the second step follows from the linearity of $\operatorname{Diag}(\cdot)$, the third step follows from Part 13, the fourth step follows from linearity of $\operatorname{Diag}(\cdot)$, the fifth step follows from Fact C.1, the sixth step follows from basic algebra the seventh step follows from Fact C.1, the eighth step follows from basic algebra, and the last step follows from the definition of $\Lambda_{x,w}$.

**Proof of Part 16.**

We have

$$
\frac{\mathrm{d}p(x,w)}{\mathrm{d}x_i} = \frac{\mathrm{d}}{\mathrm{d}x_i}\log(\det(B_x^\top B_x))
$$

$$= \operatorname{tr}[(B_x^\top B_x)^{-1} \frac{\mathrm{d}}{\mathrm{d}x_i}(B_x^\top B_x)]$$

$$= -2\operatorname{tr}[(B_x^\top B_x)^{-1} B_x^\top \operatorname{diag}(A_{x,*,i})B_x]$$

$$= -2\operatorname{tr}[B_x(B_x^\top B_x)^{-1} B_x^\top \operatorname{diag}(A_{x,*,i})]$$

$$= -2\operatorname{tr}[\operatorname{diag}(\sigma_{x,w})\operatorname{diag}(A_{x,*,i})]$$

$$= -2\langle \sigma_{x,w}, A_{x,*,i}\rangle$$

$$= -2A_{x,*,i}^\top \sigma_{x,w},$$

where the first step follows from the definition of $p(x, w)$, the second step follows from Fact C.2, the third step follows from Part 6, the fourth step follows from the cyclic property of trace, the fifth step follows the definition of $\sigma_{x,w}$, the sixth step follows from basic algebra, and the last step follows from the definition of vector inner product.

Thus,

$$\frac{\mathrm{d}p(x, w)}{\mathrm{d}x} = -2A_x^\top \sigma_{x,w}.$$

**Proof of Part 17.**

We have

$$\frac{\mathrm{d}p(x, w)}{\mathrm{d}w_i} = \frac{\mathrm{d}}{\mathrm{d}w_i} \log(\det(B_x^\top B_x))$$

$$= \operatorname{tr}[(B_x^\top B_x)^{-1} \frac{\mathrm{d}}{\mathrm{d}w_i}(B_x^\top B_x)]$$

$$= \operatorname{tr}[(B_x^\top B_x)^{-1}((1 - 2/q)B_x^\top \operatorname{diag}(e_i \circ w^{-1})B_x)]$$

$$= (1 - 2/q)\operatorname{tr}[(B_x^\top B_x)^{-1} B_x^\top e_i e_i^\top W^{-1} B_x]$$

$$= (1 - 2/q)\operatorname{tr}[e_i^\top W^{-1} B_x(B_x^\top B_x)^{-1} B_x^\top e_i]$$

$$= (1 - 2/q)(W^{-1} B_x(B_x^\top B_x)^{-1} B_x^\top)_{i,i}$$

$$= (1 - 2/q)(W^{-1} P_{x,w})_{i,i}$$

$$= (1 - 2/q)w_i^{-1} \sigma_{x,w,i},$$

where the first step follows from the definition of $B_x$, the second step follows from Fact C.2, the third step follows from Part 9, the fourth step follows from $\operatorname{diag}(e_i) = e_i e_i^\top$, the fifth step follows from the cyclic property of the trace, the sixth step follows from basic algebra, the seventh step follows from the definition of $P_{x,w}$, and the last step follows from basic algebra.

Therefore,

$$\nabla_w p(x, w) = (1 - 2/q)W^{-1} \sigma_{x,w}.$$

**Proof of Part 18.**

We have

$$\frac{\mathrm{d}^2 p(x, w)}{\mathrm{d}x_j \mathrm{d}x_i} = \frac{\mathrm{d}}{\mathrm{d}x_i}(-2e_j^\top A_x^\top \sigma_{x,w})$$

$$= -2e_j^\top \frac{\mathrm{d}}{\mathrm{d}x_i}(A_x^\top \sigma_{x,w})$$

$$= -2e_j^\top (\frac{\mathrm{d}A_x^\top}{\mathrm{d}x_i}\sigma_{x,w} + A_x^\top \frac{\mathrm{d}\sigma_{x,w}}{\mathrm{d}x_i})$$

$$= -2e_j^\top (-A_x^\top \operatorname{diag}(A_{x,*,i})\sigma_{x,w} + A_x^\top(-2\Lambda_{x,w}A_{x,*,i}))$$

$$= e_j^\top (2A_x^\top \operatorname{diag}(A_{x,*,i})\sigma_{x,w} + 4A_x \Lambda_{x,w}A_{x,*,i})$$

$$= e_j^\top (2A_x^\top \Sigma_{x,w}A_{x,*,i} + 4A_x \Lambda_{x,w}A_{x,*,i})$$

$$= e_j^\top (2A_x^\top \Sigma_{x,w}A_x + 4A_x \Lambda_{x,w}A_x)e_i,$$

where the first step follows from Part 16, the second step follows from basic algebra, the third step follows from the chain rule for product, the fourth step follows from Part 4 and Part 14, the fifth step follows from basic algebra, the sixth step follows from $\mathrm{diag}(a)b = \mathrm{diag}(b)a$ (Fact C.1), and the last step follows from basic algebra.

Therefore,

$$\nabla^2_{xx} p(x, w) = A_x^\top (2\Sigma_{x,w} + 4\Lambda_{x,w}) A_x.$$

**Proof of Part 19.**

We have

$$\frac{\mathrm{d}^2 p(x,w)}{\mathrm{d}w_j \mathrm{d}w_i} = \frac{\mathrm{d}}{\mathrm{d}w_i}((1 - 2/q) w_j^{-1} \sigma_{x,w,j})$$

$$= (1 - 2/q)(\frac{\mathrm{d}w_j^{-1}}{\mathrm{d}w_i} \sigma_{x,w,j} + w_j^{-1} \frac{\mathrm{d}\sigma_{x,w,j}}{\mathrm{d}w_i})$$

$$= (1 - 2/q)(-\mathbf{1}[i = j] w_j^{-2} \sigma_{x,w,j} + w_j^{-1} \frac{\mathrm{d}\sigma_{x,w,j}}{\mathrm{d}w_i})$$

$$= (1 - 2/q)(-\mathbf{1}[i = j] w_j^{-2} \sigma_{x,w,j} + w_j^{-1}((1 - 2/q) e_j^\top \Lambda_{x,w}(w^{-1} \circ e_i)))$$

$$= c_q(-\mathbf{1}[i = j] w_j^{-2} \sigma_{x,w,j} + w_j^{-1}(c_q e_j^\top \Lambda_{x,w}(w^{-1} \circ e_i)))$$

$$= c_q(-e_j^\top w_i^{-1} \sigma_{x,w,i} w_i^{-1} e_i + w_j^{-1}(c_q e_j^\top \Lambda_{x,w}(w^{-1} \circ e_i)))$$

$$= c_q(-e_j^\top w_i^{-1} \sigma_{x,w,i} w_i^{-1} e_i + (c_q(e_j \circ w^{-1})^\top \Lambda_{x,w}(w^{-1} \circ e_i)))$$

$$= -c_q e_j^\top W^{-1}(\Sigma_{x,w} - c_q \Lambda_{x,w}) W^{-1} e_i,$$

where the first step follows from Part 17, the second step follows from the chain rule for the product, the third step follows from $\frac{\mathrm{d}w_j^{-1}}{\mathrm{d}w_i} = \mathbf{1}[i = j] w_j^{-2}$, the fourth step follows from Part 15, the fifth step follows from $c_q = 1 - 2/q$, the sixth step follows from $\mathbf{1}[i = j] = e_j^\top e_i$, the seventh step follows from $w_j^{-1} e_j^\top = (e_j \circ w^{-1})^\top$, the eighth step follows from $e_j^\top a_i e_i = e_j^\top \mathrm{diag}(a) e_i$ (we treat $w_i^{-1} \sigma_{x,w,i} w_i^{-1} = a_i$) and $(a \circ b) = \mathrm{diag}(a)b = \mathrm{diag}(b)a$ (Fact C.1).

Therefore,

$$\nabla^2_{ww} p(x, w) = -c_q W^{-1}(\Sigma_{x,w} - c_q \Lambda_{x,w}) W^{-1}.$$

**Proof of Part 20.**

We have

$$\frac{\mathrm{d}^2 p(x,w)}{\mathrm{d}x_j \mathrm{d}w_i} = \frac{\mathrm{d}(-2e_j^\top A_x^\top \sigma_{x,w})}{\mathrm{d}w_i}$$

$$= -2e_j^\top A_x^\top \frac{\mathrm{d}\sigma_{x,w}}{w_i}$$

$$= -2(1 - 2/q) e_j^\top A_x^\top \Lambda_{x,w}(w^{-1} \circ e_i)$$

$$= -2(1 - 2/q) e_j^\top A_x^\top \Lambda_{x,w} W^{-1} e_i,$$

where the first step follows from Part 16, the second step follows from basic algebra, the third step follows from Part 15, and the last step follows from $a \circ b = \mathrm{diag}(a)b$ (Fact C.1).

Therefore, by the chain rule

$$\nabla_{xw} p(x, w) = -2c_q A_x^\top \Lambda_{x,w} W^{-1}.$$

**Proof of Part 21.**

We have

$$\frac{\mathrm{d}(W^{1/2 - 1/q})}{\mathrm{d}q} = \frac{\mathrm{d}e^{(1/2 - 1/q)\log W}}{\mathrm{d}q}$$

$$= W^{1/2-1/q}\left(\left(\frac{\mathrm{d}(1/2 - 1/q)}{\mathrm{d}q}\right)\log W + (1/2 - 1/q)W^{-1}\frac{\mathrm{d}W}{\mathrm{d}q}\right)$$

$$= W^{1/2-1/q}\left(\frac{1}{q^2}\log W + (1/2 - 1/q)W^{-1}\frac{\mathrm{d}W}{\mathrm{d}q}\right), \tag{12}$$

where the first step follows from $a = e^{\log a}$, the second step follows from the chain rule, and the last step follows from $\frac{\mathrm{d}(1/q)}{\mathrm{d}q} = -1/q^2$.

For the convenience of writing proofs, we define a diagonal matrix

$$U := \frac{1}{q^2}\log W + (1/2 - 1/q)W^{-1}\frac{\mathrm{d}W}{\mathrm{d}q}.$$

We have

$$\frac{\mathrm{d}B_x}{\mathrm{d}q} = \frac{\mathrm{d}(W^{1/2-1/q}A_x)}{\mathrm{d}q}$$

$$= W^{1/2-1/q}\left(\frac{1}{q^2}\log W + (1/2 - 1/q)W^{-1}\frac{\mathrm{d}W}{\mathrm{d}q}\right)A_x$$

$$= UB_x, \tag{13}$$

where the first step follows from the definition of $B_x$, and the second step follows from Eq. (12), and the last step follows from the definition of $U$.

We have

$$\frac{\mathrm{d}B_x^\top B_x}{\mathrm{d}q} = \frac{\mathrm{d}B_x^\top}{\mathrm{d}q} \cdot B_x + B_x^\top \cdot \frac{\mathrm{d}B_x}{\mathrm{d}q}$$

$$= B_x^\top UB_x + B_x^\top UB_x$$

$$= 2B_x^\top UB_x, \tag{14}$$

where the first step follows from the product rule, the second step follows from Eq. (13), and the third step follows from basic algebra.

We have

$$\frac{\mathrm{d}(B_x^\top B_x)^{-1}}{\mathrm{d}q} = -(B_x^\top B_x)^{-1}\frac{\mathrm{d}(B_x^\top B_x)}{\mathrm{d}q}(B_x^\top B_x)^{-1}$$

$$= -2(B_x^\top B_x)^{-1}B_x^\top UB_x(B_x^\top B_x)^{-1}, \tag{15}$$

where the first step follows from Fact C.2, the second step follows from Eq. (14).

For notation simplicity, we define $P := P(B_x)$ and $\Lambda := \Lambda_{x,w}$.

Then, we have

$$\frac{\mathrm{d}P(B_x)}{\mathrm{d}q} = \frac{\mathrm{d}B_x(B_x^\top B_x)^{-1}B_x^\top}{\mathrm{d}q}$$

$$= \frac{\mathrm{d}B_x}{\mathrm{d}q}(B_x^\top B_x)^{-1}B_x^\top + B_x\frac{\mathrm{d}(B_x^\top B_x)^{-1}}{\mathrm{d}q}B_x^\top + B_x(B_x^\top B_x)^{-1}\frac{\mathrm{d}B_x^\top}{\mathrm{d}q}$$

$$= UP - 2PUP + PU, \tag{16}$$

where the first step follows from the definition of $P(B_x)$, the second step follows from the product rule, and the last step follows from Eq. (13), Eq. (15) and the definition of $P(B_x)$.

We can show

$$\frac{\mathrm{d}\sigma_{x,w}}{\mathrm{d}q} = \frac{\mathrm{d}\,\mathrm{Diag}(P)}{\mathrm{d}q}$$

$$
\begin{aligned}
&= \mathrm{Diag}(\frac{\mathrm{d}P}{\mathrm{d}q}) \\
&= \mathrm{Diag}(UP - 2PUP + PU) \\
&= \mathrm{Diag}(UP) - 2\,\mathrm{Diag}(PUP) + \mathrm{Diag}(PU) \\
&= 2\Sigma u - 2(P \circ P)u \\
&= 2(\Sigma - P \circ P)u \\
&= 2\Lambda u \\
&= \Lambda((1 - 2/q)W^{-1}\frac{\mathrm{d}w}{\mathrm{d}q} + \frac{2}{q^2}\log w),
\end{aligned}
$$

where the first step follows from the definition of $\sigma_{x,w}$, the second step follows from the linearity of $\mathrm{Diag}(\cdot)$, the third step follows from Eq. (16), the fourth step follows from the linearity of $\mathrm{Diag}(\cdot)$, the fifth step follows from Fact C.1, the sixth step follows from basic algebra, the seventh step follows from $\Lambda = \Sigma - P \circ P$, and the last step follows from the definition of $u$. $\square$

### D.9 PROPERTIES OF PROJECTION MATRIX

**Lemma D.17** (Projection Matrices, Lemma 47 in page 46 in (Lee & Sidford, 2019))**.** *If the following conditions hold:*

- *Let $P \in \mathbb{R}^{m \times m}$ be an arbitrary orthogonal projection matrix (see Definition C.7).*

- *Let $\Sigma := P \circ I \in \mathbb{R}^{m \times m}$.*

*Then, for all $i, j \in [m]$, we have*

- **Part 1.** $\Sigma_{i,i} = \sum_{j=1}^{m} P_{i,j}^{\circ 2}$.

- **Part 2.** $0 \preceq P^{\circ 2} \preceq \Sigma \preceq I, (0 \le \Sigma_{i,i} \le 1)$.

- **Part 3.** $P_{i,j}^{\circ 2} \le \Sigma_{i,i}\Sigma_{j,j}$.

- **Part 4.** $\|\Sigma^{-1}P^{\circ 2}x\|_\infty \le \|x\|_\Sigma$.

- **Part 5.** $\|\Sigma^{-1}P^{\circ 2}x\|_\infty \le \|x\|_\infty$.

- **Part 6.** $\sum_{i=1}^{m}\Sigma_{i,i} = \mathrm{rank}[P]$.

- **Part 7.** $|y^\top X P^{\circ 2} y| \le \|y\|_\Sigma^2 \cdot \|x\|_\Sigma$.

- **Part 8.** $|y^\top (P \circ PXP)y| \le \|y\|_\Sigma^2 \cdot \|x\|_\Sigma$.

*Proof.* **Proof of Part 1.**

We have

$$
\begin{aligned}
\Sigma_{i,i} &= P_{i,i} \\
&= e_i^\top P e_i \\
&= e_i^\top P P e_i \\
&= \sum_{j=1}^{m} P_{i,j}^2 \\
&= \sum_{j=1}^{m} P_{i,j}^{\circ 2},
\end{aligned}
$$

where the first step follows from $\Sigma_{i,i}$ is a diagonal entry for $i \in [m]$, the second step follows from the property of matrix, the third step follows from $P = PP$, the fourth step follows from matrix product, and the last step follows from the definition of $P^{\circ 2}$.

**Proof of Part 2.**

We observe that since $P$ is a projection matrix, all its eigenvalues are either 0 or 1. Therefore, $\Sigma \preceq I$.

By part 1, consider the matrix $\Sigma - P^{\circ 2}$, for $i \in [m]$, its diagonal entries are

$$(\Sigma - P^{\circ 2})_{i,i} = \sum_{j \neq i} P_{i,j}^2.$$

And its off-diagonal entries, for $i \neq j$,

$$(\Sigma - P^{\circ 2})_{i,j} = -P_{i,j}^2.$$

Consequently, we can conclude by Fact C.9 that $\Sigma - P^{\circ 2} \succeq 0$. Rearranging terms and using Fact C.16 yields Part 2.

**Proof of Part 3.**

For $i, j \in [m]$, we have

$$
\begin{aligned}
P_{i,j}^2 &= (\sum_{k=1}^m P_{i,k} P_{k,j})^2 \\
&\leq (\sum_{k=1}^m P_{i,k}^2)(\sum_{k=1}^m P_{k,j}^2) \\
&= \Sigma_{i,i} \Sigma_{j,j},
\end{aligned}
$$

where the first step follows from $P = PP$, the second step follows from Cauchy-Schwarz, and the third step follows from Part 1.

**Proof of Part 4.**

For any index $i \in [m]$, we have

$$
\begin{aligned}
|e_i^\top P^{\circ 2} x|^2 &= (|\sum_{j=1}^m P_{i,j}^{\circ 2} x_j|)^2 \\
&\leq (\sum_{j=1}^m \Sigma_{j,j} x_j^2) \cdot (\sum_{j=1}^m \frac{P_{i,j}^{\circ 4}}{\Sigma_{j,j}}) \\
&\leq (\sum_{j=1}^m \Sigma_{j,j} x_j^2) \cdot (\sum_{j=1}^m \frac{P_{i,j}^2 \Sigma_{i,i} \Sigma_{j,j}}{\Sigma_{j,j}}) \\
&= (\sum_{j=1}^m \Sigma_{j,j} x_j^2) \cdot (\Sigma_{i,i} \sum_{j=1}^m P_{i,j}^2) \\
&= (\sum_{j=1}^m \Sigma_{j,j} x_j^2) \cdot \Sigma_{i,i}^2 \\
&= (\Sigma_{i,i} \|x\|_\Sigma)^2,
\end{aligned}
$$

where the first step follows from basic algebra, the second step follows from Cauchy-Schwarz, the third step follows from Part 3, the fourth step follows from basic algebra, the fifth step follows Part 1, and the last step follows from $\|x\|_\Sigma := \sqrt{\sum_{j=1}^m \Sigma_{j,j} x_j^2}$.

Taking the square root of the above equation, we get

$$|e_i^\top P^{\circ 2} x| \leq \Sigma_{i,i} \|x\|_\Sigma.$$

**Proof of Part 5.**

We have

$$|e_i^\top P^{\circ 2} x| = |\sum_{j=1}^m P_{i,j}^{\circ 2} x_j|$$

$$\leq \sum_{j=1}^{m} |P_{i,j}^{\circ 2} x_j|$$

$$= \sum_{j=1}^{m} P_{i,j}^{\circ 2} |x_j|$$

$$\leq \sum_{j=1}^{m} P_{i,j}^{\circ 2} \|x\|_\infty$$

$$= \Sigma_{i,i} \|x\|_\infty,$$

where the first step follows from basic algebra, the second step follows from the triangle inequality, the third step follows from $P_{i,j}^{\circ 2} \geq 0$ for $i, j \in [m]$, the fourth step follows from the definition of $\|\cdot\|_\infty$, and the last step follows from Part 1.

**Proof of Part 6.**

We have

$$\sum_{i=1}^{m} \Sigma_{i,i} = \mathrm{tr}[P]$$

$$= \mathrm{rank}[P],$$

where the first step follows from $\Sigma = \mathrm{diag}(\mathrm{Diag}(P))$, and the second step holds since all the eigenvalues of $P$ are either 0 or 1.

**Proof of Part 7.**

Recall $x \in \mathbb{R}^m$ and $X = \mathrm{diag}(x)$. We have

$$|y^\top X P^{\circ 2} y| = |\langle X^\top y, P^{\circ 2} y\rangle|$$

$$= |\sum_{i=1}^{m} (x_i y_i) \cdot (P^{\circ 2} y)_i|$$

$$= |\sum_{i=1}^{m} x_i y_i e_i^\top P^{\circ 2} y|$$

$$\leq \sum_{i=1}^{m} |x_i| \cdot |y_i| \cdot |e_i^\top P^{\circ 2} y|$$

$$\leq \sum_{i=1}^{m} |x_i| \cdot |y_i| \cdot \Sigma_{i,i} \cdot \|y\|_\Sigma$$

$$\leq \sqrt{\sum_{i=1}^{m} \Sigma_{i,i} x_i^2} \sqrt{\sum_{i=1}^{m} \Sigma_{i,i} y_i^2} \cdot \|y\|_\Sigma$$

$$= \|x\|_\Sigma \|y\|_\Sigma \|y\|_\Sigma,$$

where the first and the second steps follow from basic algebra, the third step follows from selecting the $i$-th entry of $(P^{\circ 2} y)$ with $e_i$, the fourth step follows from triangle inequality and $|abc| = |a| \cdot |b| \cdot |c|$, the fifth step follows from Part 4, the sixth step follows from Cauchy-Schwarz, and the last step follows from the definition of $\|x\|_\Sigma$.

**Proof of Part 8.**

We define

$$a_1 := \sum_{i=1}^{m} \sum_{j=1}^{m} |y_i| \cdot |y_j| \cdot P_{i,j}^2,$$

$$a_2 := \sum_{i=1}^{m} \sum_{j=1}^{m} |y_i| \cdot |y_j| \cdot (PXP)_{i,j}^2.$$

We have

$$|y^\top(P \circ PXP)y| = |\sum_{i=1}^{m}\sum_{j=1}^{m} y_i y_j (P \circ PXP)_{i,j}|$$

$$= |\sum_{i=1}^{m}\sum_{j=1}^{m} y_i y_j P_{i,j} (PXP)_{i,j}|$$

$$\leq \sqrt{a_1 \cdot a_2},$$

where the first step follows from the fact that $x^\top A x = \sum_{i=1}^{m}\sum_{j=1}^{m} x_i x_j A_{i,j}$, the second step follows from the fact that $(A \circ B)_{i,j} = A_{i,j} \cdot B_{i,j}$, and the last step follows from Cauchy-Schwarz.

Letting $|x|$ and $|y|$ be the vectors whose entries are the absolute values of the entries of $x$ and $y$ respectively, we have

$$a_1 = \||y|\|_{P^{\circ 2}}^2$$

$$\leq \||y|\|_{\Sigma}^2$$

$$= \|y\|_{\Sigma}^2,$$

where the first step follows from the definition of $\||y|\|_{P^{\circ 2}}$, the second step follows from Part 2, and the third step holds since $\Sigma$ is diagonal.

We have

$$a_2 = \sum_{i=1}^{m}\sum_{j=1}^{m} |y_i| \cdot |y_j| \cdot (PXP)_{i,j}^2$$

$$= \sum_{i=1}^{m}\sum_{j=1}^{m} |y_i| \cdot |y_j| \cdot (\sum_{k=1}^{m} P_{i,k} P_{j,k} x_k)^2$$

$$= \sum_{i=1}^{m}\sum_{j=1}^{m} (\sum_{k=1}^{m} (P_{i,k}\sqrt{|y_i||x_k|})(P_{j,k}\sqrt{|y_j||x_k|}))^2$$

$$\leq \sum_{i=1}^{m}\sum_{j=1}^{m} (\sum_{k=1}^{m} P_{i,k}^2 |y_i||x_k|) \cdot (\sum_{k=1}^{m} P_{j,k}^2 |y_j||x_k|)$$

$$= (\sum_{i=1}^{m}\sum_{k=1}^{m} |y_i| P_{i,k}^2 |x_k|)^2$$

$$= (|y|^\top P^{\circ 2} |x|)^2$$

$$= \langle |y|, |x| \rangle_{P^{\circ 2}}^2$$

$$\leq \||y|\|_{P^{\circ 2}}^2 \||x|\|_{P^{\circ 2}}^2$$

$$\leq \|y\|_{\Sigma}^2 \|x\|_{\Sigma}^2,$$

where the first step follows from the definition of $a_2$, the second step follows from basic algebra, the third step follows from absorbing $|y_i| \cdot |y_j|$ into $(\sum_{k=1}^{m} P_{i,k} P_{j,k} x_k)^2$, the forth step follows from Cauchy-Schwartz, the fifth step follows from basic algebra, the sixth step follows from basic algebra, the seventh step follows from the definition of inner product, the eighth step follows from Cauchy-Schwartz, and the last step follows from Part 2.

Combining these inequalities then yields the desired bound on $|y^\top(P \circ PXP)y|$.

$\square$

# E LINEAR PROGRAMMING: LEWIS WEIGHT COMPUTATION

In Section E.1, we introduce the volumetric potential. In Section E.2, we show that Lewis weights are the result of solving a particular convex optimization problem. In Section E.3, we study the stability of Lewis weight under rescaling. In Section E.4, we study the Lewis weight rounding properties. In Section E.5, we compute the gradient and Hessian of the volumetric potential. In Section E.6, we present an important lemma for Hessian approximation. In Section E.7, compute the weight function. In Section E.8, we show we can get a multiplicative approximation of $w_p$. In Section E.9, we introduce exact weight computation. In Section E.10, we present approximate weight computation. In Section E.11, we introduce the computation of the leverage score. In Section E.12, we compute an initial weight. In Section E.13, we introduce the theorem of exact weight computation. In Section E.14, we provide the theorem of approximate weight computation. In Section E.15, we introduce a useful theorem for the weight function.

## E.1 VOLUMETRIC POTENTIAL

**Definition E.1** (Lewis Weight (Lewis, 1978), see Definition 2.2 in page 3 of (Cohen & Peng, 2015) as an example). *For all $p > 0$ and non-degenerate $A \in \mathbb{R}^{m \times n}$, we define the $\ell_p$ Lewis weight $w_p(A)$ as the vector $w \in \mathbb{R}^m_{>0}$ such that $w = \sigma(W^{1/2-1/p}A)$ where $W = \mathrm{diag}(w)$.*

**Fact E.2.** *If $w_p$ is the Lewis Weight in Definition E.1 that satisfies $w_p = \sigma(W_p^{1/2-1/p}A)$, then we have*

$$W_p = \mathrm{Diag}(A(A^\top W^{1-2/p}A)^{-1}A^\top)^{p/2}.$$

*Proof.* We have

$$
\begin{aligned}
w_p &= \sigma(W_p^{1/2-1/p}A) \\
&= \mathrm{Diag}(W^{1/2-1/p}A(A^\top W^{1-2/p}A)^{-1}A^\top W^{1/2-1/p}).
\end{aligned}
$$

The above equation implies

$$\mathbf{1}_m = \mathrm{Diag}(W_p^{-1/p}A^\top(A^\top W_p^{1-2/p}A)^{-1}AW_p^{-1/p}).$$

The above equation is equivalent to

$$W_p^{2/p} = \mathrm{Diag}(A^\top(A^\top W_p^{1-2/p}A)^{-1}A).$$

Thus

$$W_p = \mathrm{Diag}(A(A^\top W_p^{1-2/p}A)^{-1}A)^{p/2}.$$

$\square$

**Definition E.3** (Volumetric Potential, Definition 21 in page 20 in (Lee & Sidford, 2019)). *For non-degenerate $A \in \mathbb{R}^{m \times n}$ and $p > 0$ with $p \neq 2$ we define the volumetric potential as*

$$\mathcal{V}_p^A(w) := -\frac{1}{1-2/p}\log\det(A^\top W^{1-2/p}A).$$

## E.2 CONVEX FORMULATION OF LEWIS WEIGHTS

**Lemma E.4** (Lemma 22 in page 20 in (Lee & Sidford, 2019)). *If the following conditions hold:*

- *We define*

$$\mathcal{V}_p^A(w) := -\frac{1}{1-2/p}\log\det(A^\top W^{1-2/p}A)$$

  *as described in Definition E.3.*

- *For all $w \in \mathbb{R}^m_{>0}$, define $f(w) := -\frac{1}{1-2/p}\log\det(A^\top W^{1-2/p}A) + \sum_{i=1}^m w_i$.*

- *We recall that the leverage score $\sigma$ is defined as $\sigma(A) := \mathrm{Diag}(A(A^\top A)^{-1}A^\top)$.*

- *Let $\sigma_w := \sigma(W^{1/2-1/p}A)$ where $W := \mathrm{diag}(w)$.*

- *Suppose all non-degenerate $A \in \mathbb{R}^{m \times n}$ its $\ell_p$ Lewis weights exist and are unique for $p > 0$.*

- *Let $p \neq 2$.*

- *Let $F_1(w)$ denote the following optimization problem:*

$$\min_{w \in \mathbb{R}^m_{>0}} \mathcal{V}^A_p(w) + \sum_{i=1}^m w_i.$$

- *Let $F_2(w)$ denote the following optimization problem:*

$$\min_{w \in \mathbb{R}^m_{>0} : \sum_{i=1}^m w_i = n} \mathcal{V}^A_p(w).$$

*Then, the following statements are true:*

- **Part 1.** *The minimizer of Problem $F_1(w)$ is in the interior of its feasible region.*

- **Part 2.** *The Lewis Weight $w_p(A)$ is the minimizer of Problem $F_1(w)$.*

- **Part 3.** *Problem $F_1(w)$ is strictly convex.*

- **Part 4.** *The minimizer of Problem $F_1(w)$ is unique.*

- **Part 5.** *Problems $F_1(w)$ and $F_2(w)$ are equivalent.*

*Proof.* **Proof of Part 1.**

For all $w \in \mathbb{R}^m_{>0}$ if $w_i > 1$ then

$$\frac{\mathrm{d}f(w)}{\mathrm{d}w_i} = 1 - \frac{\sigma_{w,i}}{w_i}$$
$$\geq 1 - \frac{1}{w_i}$$
$$> 0,$$

where the first step follows from Lemma E.9, the second step follows from $\sigma_{w,i} \in [0,1]$ (Part 2 of Lemma D.17), and the last step follows from $w_i > 1$.

The above derivative computation implies that $f(w)$ is monotonically increasing when $w_i > 1$ (this is range on the right side of extreme point).

Hence, we have $\inf_{w_i > 0} f(w) = \inf_{1 > w_i > 0} f(w)$.

**Case 1.** $p > 2$.

Now if $p > 2$ and $w_i \in [0,1]$ for all $i \in [m]$ then since $1 - 2/p > 0$,

$$\sigma_{w,i} = \sigma(W^{1/2-1/p}A)_i$$
$$= (W^{1/2-1/p}A(A^\top W^{1-2/p}A)^{-1}A^\top W^{1/2-1/p})_{i,i}$$
$$= w_i^{1-2/p}(A(A^\top W^{1-2/p}A)^{-1}A^\top)_{i,i}$$
$$\geq w_i^{1-2/p}(A(A^\top A)^{-1}A^\top)_{i,i}$$
$$= w_i^{1-2/p}\sigma(A)_i, \tag{17}$$

where the first step follows from the definition of $\sigma_w$, the second step follows from the definition of $\sigma(W^{1/2-1/p}A)$, the third step follows from the fact that $W = \mathrm{diag}(w)$ is a diagonal matrix and $M$ is a square matrix so $(WMW)_{i,i} = w_i M_{i,i} w_i$, the fourth step follows from $W^{1-2/p} \preceq I_m$ (and then applying Fact C.20), the fifth step follows from the definition of $\sigma(A)$.

Since $A$ is non-degenerate, $\sigma(A)_i \in (0, 1]$ for all $i$.

Therefore for any $j \in [m]$ with $w_j < \sigma(A)_j^{p/2}$, we have

$$
\begin{aligned}
\frac{\mathrm{d}f(w)}{\mathrm{d}w_j} &= 1 - \frac{\sigma_{w,j}}{w_j} \\
&\le 1 - w_j^{-2/p}\sigma(A)_j \\
&< 0,
\end{aligned}
$$

where the first step follows from Lemma E.9, the second step follows from Eq. (17), and the last step follows from $w_j < \sigma(A)_j^{p/2}$.

The above derivative computation implies that $f(w)$ is monotonically increasing when $w_j < \sigma(A)_j^{p/2}$ (this is range on the left of extreme point).

Consequently, $\inf_{w_i>0} f(w) = \inf_{1>w_i\ge 0} f(w) = \inf_{1>w_i>\sigma(A)_i^{p/2}} f(w)$.

**Case 2.** $p < 2$.

Similarly, if $p < 2$, $w_i \in [0, 1]$ for all $i \in [m]$, and $w_{\min} = \min_{i\in[m]} w_i$. Then since $1 - 2/p < 0$, we have $W^{1-2/p} \preceq w_{\min}^{1-2/p} I_m$.

Consequently, by analogous derivation to Eq. (17), we can show

$$
\begin{aligned}
\sigma_{w,i} &= \sigma(W^{1/2-1/p}A)_i \\
&= w_i^{1-2/p}(A(A^\top W^{1-2/p}A)^{-1}A^\top)_{i,i} \\
&\ge (w_i/w_{\min})^{1-2/p}(A(A^\top A)^{-1}A^\top)_{i,i} \\
&= (w_i/w_{\min})^{1-2/p}\sigma(A)_i,
\end{aligned}
$$

where the first step follows from the definition of $\sigma_w$, the second step follows from the definition of $\sigma(W^{1/2-1/p}A)$, the third step follows from $W^{1-2/p} \preceq w_{\min}^{1-2/p} I_m$ (and then applying Fact C.20), the fourth step follows from the definition of $\sigma(A)$.

If $j \in \arg\min_{i\in[m]} w_i$, this implies that $\sigma_{w,j} \ge \sigma(A)_j$ and therefore if $w_j < \sigma(A)_j$ we have $\frac{\mathrm{d}f(w)}{\mathrm{d}w_j} < 0$.

Therefore, if we let $\sigma_{\min} := \min_{i\in[m]} \sigma_i > 0$, we have $\inf_{w_i>0} f(w) = \inf_{1>w_i\ge\sigma_{\min}} f(w)$.

In either case, since $f$ is continuous, the above reasoning argues that $f$ achieves its minimum on the interior of the domain.

**Proof of Part 2.** Therefore, we have that the minimizer of $w_*$ of $f(w)$ satisfies $\nabla F_1(w_*) = 0$. By Part 1 of Lemma E.9, we further have

$$
-W_*^{-1}\sigma_{w_*} + \mathbf{1}_m = \mathbf{0}_m.
$$

Therefore, we can conclude that $w_{*,i} = \sigma_{w,i}$ for all $i \in [n]$.

This proves that the minimizer of $f(w)$ exists on $w \in \mathbb{R}^m_{>0}$ and equals to the Lewis weights in Definition E.1.

**Proof of Part 3.** Further, for all $w > 0$,

$$
\begin{aligned}
\nabla^2 f(w) &= \nabla^2 \mathcal{V}_p^A(w) + 0 \\
&\succeq \frac{2}{\max\{p, 2\}} \cdot W^{-1}\Sigma_w W^{-1} \\
&\succ 0,
\end{aligned}
$$

where the first step follows from relationship between $f(w)$ and $\mathcal{V}_p^w(w)$, the second step follows from Part 3 of Lemma E.9, and the last step follows from Fact C.25 since $\Sigma_w$ and $W$ are positive definite matrices and $W$ is diagonal.

Therefore $f$ is strictly convex where $1 \geq w_i \geq \min\{\sigma_i, \sigma_i^{p/2}\}$ for all $i$.

**Proof of Part 4.** Consequently, the minimizer of $F_1$ is unique and it is the unique point satisfying $\nabla f(w) = 0$ for $w \in \mathbb{R}_{>0}^n$.

**Proof of Part 5.** Further, since $\sum_{i=1}^m \Sigma_{w,i,i} = \text{rank}[A] = n$ by Part 6 of Lemma D.17, we have $\sum_{i=1}^m w_p(A)_i = n$ and we have the equivalence of the two objective functions. $\qquad\square$

### E.3 STABILITY OF LEWIS WEIGHTS UNDER RESCALING

**Lemma E.5** (Lemma 24 in page 21 in (Lee & Sidford, 2019))**.** *If the following conditions hold:*

- *For all non-degenerate $A \in \mathbb{R}^{m \times n}$.*

- *Let $p > 0$ with $p \neq 2$.*

- *Let $w_p(\cdot)$ be defined as Definition E.1.*

- *Let $v \in \mathbb{R}^m$, define $w(v) := w_p(VA)$ where $V := \text{Diag}(v)$.*

- *We recall that the leverage score $\sigma$ is defined as $\sigma(A) := \text{Diag}(A(A^\top A)^{-1} A^\top)$.*

- *We recall that $\Sigma(A) := \text{diag}(\sigma(A))$.*

- *We recall that $\Lambda(A) := \Sigma(A) - P^{\circ 2}(A)$.*

- *Define $\Lambda_v := \Lambda(W^{1/2 - 1/p} V A)$.*

- *Define $W_v := \text{diag}(w(v))$.*

- *For $w(v) : \mathbb{R}^m \to \mathbb{R}^m$ and $v \in \mathbb{R}^m$, we use $J_w(v) \in \mathbb{R}^{m \times n}$ to denote the Jacobian of $w$ at $v$, where $J_w(v)_{i,j} := \frac{\mathrm{d}}{\mathrm{d}v_j} w(v)_i$ for $i \in [m], j \in [m]$.*

*Then, we have*

$$J_w(v) = 2W_v(W_v - (1 - 2/p)\Lambda_v)^{-1}\Lambda_v V^{-1}.$$

*Proof.* We define

$$f(v, w) := -\frac{1}{1 - 2/p} \log \det(A^\top V W^{1 - 2/p} V A) + \sum_{i=1}^m w_i.$$

Let us applying Lemma E.4 to the above equation by treating $VA$ as $A$. Lemma E.4 shows that

$$w(v) = \arg \min_{w \in \mathbb{R}_{>0}^m} f(v, w)$$

and that the optimum is in the interior.

Hence, the optimality conditions yield $\nabla_w f(v, w(v)) = 0$. Taking the derivative with respect to $v$ on both sides, we have

$$\nabla_v \nabla_w f(v, w(v)) = \nabla_v 0.$$

We further expand the left side of the above equation, then we have

$$\nabla_{w,v}^2 f(v, w(v)) + \nabla_{w,w}^2 f(v, w(v)) J_w(v) = 0.$$

Therefore, we have that

$$J_w(v) = -(\nabla_{w,w}^2 f(v, w(v)))^{-1} \nabla_{w,v}^2 f(v, w(v)). \tag{18}$$

And we have

$$\nabla_{w,w}^2 f(v, w) = W^{-1}(\Sigma_w - (1 - 2/p)\Lambda_w)W^{-1}, \tag{19}$$

where the step follows from Part 19 of Lemma D.16. (We remark that in the future, when we use the above equation, we need to replace $f(v, w)$ by $f(v, w(v))$, thus all the $W$ and $w$ should be $W_v$ and $w(v)$)

For $\nabla^2_{w,v} f(v, w(v))$, we note that

$$\nabla_w f(v, w) = -W^{-1} \sigma(W^{1/2 - 1/p} V A),$$

where the step follows from Part 17 of Lemma D.16.

The Part 15 of Lemma D.16 with $q = \infty$, we can be re-stating as follows

$$\frac{\mathrm{d}}{w_j} \sigma(W^{1/2} A) = \Lambda(w^{-1} \circ e_j).$$

Replacing $W^{1/2}$ by $V$, we obtain

$$\frac{\mathrm{d}}{v_j} \sigma(V A) = 2\Lambda(v^{-1} \circ e_j). \tag{20}$$

Taking derivative with respect to $v$ gives that

$$
\begin{aligned}
\nabla^2_{w,v} f(v, w)_{i,j} &= \frac{\mathrm{d}}{\mathrm{d}v_j} \frac{\mathrm{d}f(v, w)}{\mathrm{d}w_i} \\
&= \frac{\mathrm{d}}{\mathrm{d}v_j} (-w_i^{-1} \sigma(W^{1/2 - 1/p} V A)_i) \\
&= \frac{\mathrm{d}}{\mathrm{d}v_j} (-(e_i \circ w^{-1})^\top \sigma(W^{1/2 - 1/p} V A)) \\
&= -e_i^\top W^{-1} \frac{\mathrm{d}}{\mathrm{d}v_j} \sigma(W^{1/2 - 1/p} V A) \\
&= -e_i^\top W^{-1} \frac{\mathrm{d}}{\mathrm{d}v_j} \sigma(V W^{1/2 - 1/p} A) \\
&= -2 e_i^\top W^{-1} \Lambda(V W^{1/2 - 1/p} A) \cdot (v^{-1} \circ e_j) \\
&= -2 e_i^\top W^{-1} \Lambda(W^{1/2 - 1/p} V A) \cdot (v^{-1} \circ e_j) \\
&= -2 e_i W^{-1} \cdot \Lambda(W^{1/2 - 1/p} V A) \cdot V^{-1} e_j,
\end{aligned}
$$

where the first step follows from basic algebra, the second step follows from Part 17 of Lemma D.16, the third step follows from Fact C.1, and the fourth steps follow from Fact C.1, the fifth step follows from the fact that the product between diagonal matrices $V$ and $W^{1/2 - 1/p}$ can commute, the sixth step follows from Eq. (20) (by treating $V$ as $W^{1/2}$ and $W^{1/2 - 1/p} A$ as $A$ when applying Eq. (20)), the seventh step follows from the fact that the product between diagonal matrices $V$ and $W^{1/2 - 1/p}$ can commute, and the last step follows from Fact C.1.

Thus, we have

$$\nabla^2_{w,v} f(v, w) = -2 W^{-1} \Lambda(W^{1/2 - 1/p} V A) V^{-1}. \tag{21}$$

We can show

$$
\begin{aligned}
\Sigma_w &= \mathrm{diag}(\sigma(W^{1/2 - 1/p} V A)) \\
&= \mathrm{diag}(w(v)) \\
&= W_v,
\end{aligned}
\tag{22}
$$

where the first step follows from Lemma D.16 by treating $V A = A_x$, the second step follows from the definition of $w(v)$, and the last step follows from $W_v = \mathrm{diag}(w(v))$.

We can show

$$
\begin{aligned}
\Lambda_w &= \Lambda(W^{1/2 - 1/p} V A) \\
&= \Lambda_v,
\end{aligned}
\tag{23}
$$

where the first step follows from Lemma D.16 by treating $VA = A_x$, the second step follows from the definition of $\Lambda_v$.

We further have

$$
\begin{aligned}
J_w(v) &= -(\nabla^2_{w,w} f(v, w(v)))^{-1} \nabla^2_{w,v} f(v, w(v)) \\
&= 2W_v (\Sigma_w - (1 - 2/p)\Lambda_w)^{-1} \Lambda_v V^{-1} \\
&= 2W_v (W_v - (1 - 2/p)\Lambda_w)^{-1} \Lambda_v V^{-1} \\
&= 2W_v (W_v - (1 - 2/p)\Lambda_v)^{-1} \Lambda_v V^{-1},
\end{aligned}
$$

where the first step follows from Eq. (18), the second step follows from Eq. (19) and Eq. (21) (with reparamertize $w$ by $w(v)$ in $f(v, w)$), the third step follows from Eq. (22), and the last step follows from Eq. (23). $\qquad\square$

**Lemma E.6** (Lemma 25 in page 22 in (Lee & Sidford, 2019)). *If the following conditions hold:*

- *Under the setting of Lemma E.5.*

- *Let $w_p(\cdot)$ be defined as Definition E.1.*

- *Let $v \in \mathbb{R}^m$, define $w(v) := w_p(VA)$ where $V := \mathrm{Diag}(v)$.*

- *Let $v \in \mathbb{R}^m_{>0}$.*

- *Let $h \in \mathbb{R}^m$.*

*We have*

- *Part 1.*
$$
\|W_v^{-1} J_w(v) h\|_{w(v)} \le p \cdot \|V^{-1} h\|_{w(v)}.
$$

- *Part 2.*
$$
\|(W_v^{-1} J_w(v) - pV^{-1}) h\|_\infty \le p \cdot \max\{p/2, 1\} \cdot \|V^{-1} h\|_{w(v)}.
$$

*Proof.* **Proof of Part 1.**

Fixing an arbitrary $v \in \mathbb{R}^m_{>0}$ and $h \in \mathbb{R}^m$.

According to the definition of $w_p(A)$ (Definition E.1), we know $w_p(VA)$ is the unique solution to

$$
w = \sigma(W^{1/2 - 1/p} VA).
$$

Thus, if we define $w := w_p(VA)$, then have $w = \sigma(W^{1/2 - 1/p} VA)$.

Since $w(v) := w_p(VA)$ (see Lemma statement), we also have $w = w(v)$.

Thus, we can further define $\Sigma$ notation,

$$
\Sigma := \Sigma(W^{1/2 - 1/p} VA),
$$

it is obvious that

$$
\Sigma = \mathrm{diag}(\sigma(W^{1/2 - 1/p} VA)) = \mathrm{diag}(w) = W = W_v = \mathrm{diag}(w(v)). \tag{24}
$$

We further define

- $\Lambda := \Lambda(W^{1/2 - 1/p} VA)$ (recall that $\Lambda \preceq \Sigma$)

- $\overline{\Lambda} := \overline{\Lambda}(W^{1/2 - 1/p} VA)$ where $\overline{\Lambda} = \Sigma^{-1/2} \Lambda \Sigma^{-1/2}$

- $P^{\circ 2} := P^{\circ 2}(W^{1/2 - 1/p} VA)$

- $Q := I - (1 - 2/p)\overline{\Lambda}$

From $Q = I - (1 - 2/p)\overline{\Lambda}$, we can multiply $p$ on both sides, we get

$$pQ = pI - p\overline{\Lambda} + 2\overline{\Lambda},$$

which is further equivalent to

$$2\overline{\Lambda} - pQ = p\overline{\Lambda} - pI. \tag{25}$$

We have that

$$
\begin{aligned}
J_w(v)h &= 2W_v(W_v - (1 - 2/p)\Lambda)^{-1}\Lambda V^{-1}h \\
&= 2\Sigma(\Sigma - (1 - 2/p)\Lambda)^{-1}\Lambda V^{-1}h \\
&= 2\Sigma^{1/2}(I - (1 - 2/p)\overline{\Lambda})^{-1}\Sigma^{-1/2}\Lambda V^{-1}h \\
&= 2\Sigma^{1/2}Q^{-1}\Sigma^{-1/2}\Lambda V^{-1}h \\
&= 2\Sigma^{1/2}Q^{-1}\overline{\Lambda}\Sigma^{1/2}V^{-1}h \\
&= 2W^{1/2}Q^{-1}\overline{\Lambda}W^{1/2}V^{-1}h \\
&= 2W^{1/2}\overline{\Lambda}Q^{-1}W^{1/2}V^{-1}h, 
\end{aligned} \tag{26}
$$

where the first step follows from Lemma E.5, the second step follows from $W_v = \Sigma$ (Eq. (24)), the third step follows from $\Lambda = \Sigma^{1/2}\overline{\Lambda}\Sigma^{1/2}$, the fourth step follows from definition of $Q$, the fifth step follows from $\overline{\Lambda} = \Sigma^{-1/2}\Lambda\Sigma^{-1/2}$, the sixth step follows from $W = \Sigma$ (Eq. (24)), and the last step follows from $Q^{-1}\overline{\Lambda} = \overline{\Lambda}Q^{-1}$ (Fact C.26).

Using Fact C.27, we can conclude that $\overline{\Lambda}(I - (1 - 2/p)\overline{\Lambda})^{-1}$ is a matrix whose eigenvalues are of the form $\lambda/(1 - (1 - 2/p)\lambda)$, where $\lambda$ represents an arbitrary eigenvalue for $\overline{\Lambda}$.

Thus, we have

$$
\begin{aligned}
\|Q^{-1}\overline{\Lambda}\| &= \|(I - (1 - 2/p)\overline{\Lambda})^{-1}\overline{\Lambda}\| \\
&\leq \max_{0 \leq \lambda \leq 1} \frac{\lambda}{1 - (1 - 2/p)\lambda} \\
&= \frac{p}{2}, 
\end{aligned} \tag{27}
$$

where the first step follows from the definition of matrix $Q$, the second step follows from Fact C.17 (for $0 \leq \lambda \leq 1$) and Fact C.27 (for the form of eigenvalues), and the last step holds since the maximum of the function occurs at $\lambda = 1$.

Consequently, Part 1 follows from

$$
\begin{aligned}
\|W^{-1}J_w(v)h\|_w &= \sqrt{h^\top J_w(v)^\top W^{-1}J_w(v)h} \\
&= \|W^{-1/2}J_w(v)h\|_2 \\
&= 2\|Q^{-1}\overline{\Lambda}W^{1/2}V^{-1}h\|_2 \\
&\leq 2\|Q^{-1}\overline{\Lambda}\| \cdot \|W^{1/2}V^{-1}h\|_2 \\
&\leq p\|W^{1/2}V^{-1}h\|_2 \\
&= p\|V^{-1}h\|_{w(v)}, 
\end{aligned}
$$

where the first step follows from the definition of $\|\cdot\|_w$, the second step follows from the definition of $\|\cdot\|_2$, the third step follows from Eq. (26), the fourth step follows from the property of matrix spectral norm, the fifth step follows from Eq. (27), and the last step follows from the definition of $\|\cdot\|_w$ and $w(v) = w$ (see Eq. (24)).

**Proof of Part 2.**

Next,

$$
\begin{aligned}
I - \overline{\Lambda} &= \Sigma^{-1/2}(\Sigma - \Sigma^{1/2}\overline{\Lambda}\Sigma^{1/2})\Sigma^{-1/2} \\
&= \Sigma^{-1/2}(\Sigma - \Lambda)\Sigma^{-1/2}
\end{aligned}
$$

$$= \Sigma^{-1/2} P^{\circ 2} \Sigma^{-1/2}$$
$$= W^{-1/2} P^{\circ 2} W^{-1/2}, \tag{28}$$

where the second step follows from $\overline{\Lambda} = \Sigma^{-1/2} \Lambda \Sigma^{-1/2}$, the third step follows from $\Lambda = \Sigma - P^{\circ 2}$, and the last step follows from $\Sigma = W$.

Then we have

$$(W^{-1} J_w(v) - pV^{-1})h$$
$$= (W^{-1} 2W^{1/2} \overline{\Lambda} Q^{-1} W^{1/2} V^{-1} - pV^{-1})h$$
$$= W^{-1/2} \cdot (2\overline{\Lambda} - pQ) Q^{-1} \cdot W^{1/2} V^{-1} h$$
$$= W^{-1/2} \cdot (p\overline{\Lambda} - pI) \cdot Q^{-1} W^{1/2} V^{-1} h$$
$$= -pW^{-1} P^{\circ 2} W^{-1/2} Q^{-1} W^{1/2} V^{-1} h, \tag{29}$$

where the first step follows from Eq. (26), the second step follows from $Q$ is invertible, the third step follows from Eq. (25), and the fourth step follows from Eq. (28).

However, we know that for all $x$,

$$\|\Sigma^{-1} P^{\circ 2} x\|_\infty \le \|x\|_\Sigma$$
$$= \|\Sigma^{1/2} x\|_2, \tag{30}$$

where the first step follows from Part 4 of Lemma D.17, the second step follows from the definition of $\| \cdot \|_2$.

Note that

$$\|Q^{-1}\| = \|(I - (1 - 2/p)\overline{\Lambda})^{-1}\|$$
$$\le \max_{0 \le \lambda \le 1} \frac{1}{1 - (1 - 2/p)\lambda}$$
$$= \max\{1, \frac{p}{2}\}, \tag{31}$$

where the first step follows from the definition of $Q$, the second step follows from Fact C.17 (for $0 \le \lambda \le 1$) and Fact C.27 (for the form of eigenvalues), and the last step follows from computing the maximum value.

And therefore

$$\|(W^{-1} J_w(v) - pV^{-1})h\|_\infty = \|pW^{-1} P^{\circ 2} W^{-1/2} Q^{-1} W^{1/2} V^{-1} h\|_\infty$$
$$= p \cdot \|W^{-1} P^{\circ 2} W^{-1/2} Q^{-1} W^{1/2} V^{-1} h\|_\infty$$
$$\le p \cdot \|W^{1/2} W^{-1/2} Q^{-1} W^{1/2} V^{-1} h\|_2$$
$$= p \cdot \|Q^{-1} W^{1/2} V^{-1} h\|_2$$
$$\le p \cdot \|Q^{-1}\| \cdot \|W^{1/2} V^{-1} h\|_2$$
$$\le p \cdot \max\{1, \frac{p}{2}\} \cdot \|W^{1/2} V^{-1} h\|_2$$
$$\le p \cdot \max\{1, \frac{p}{2}\} \cdot \|V^{-1} h\|_w,$$

where the first step follows from Eq. (29), the second step follows from the linearity of the norm, the third step follows from Eq. (30), the fourth step follows from $W^{1/2} W^{-1/2} = I$, the fifth step follows from property of the matrix spectral norm, the sixth step follows from Eq. (31), and the last step follows from the definition of $\| \cdot \|_w$.

Thus, we complete the proof. $\square$

### E.4 LEWIS WEIGHT ROUNDING PROPERTIES

**Lemma E.7** (Lemma 28 in page 24 in (Lee & Sidford, 2019)). *If the following conditions hold:*

- *Let $A \in \mathbb{R}^{m \times n}$, $p > 0$.*

- *Define $w := w_p(A)$.*

- *Define $\alpha := 2/p - 2/r$.*

- *Let $r \geq p$.*

- *Let $g_\alpha := (1 + \alpha)(1 + 1/\alpha)^\alpha$.*

*We have*

$$A^\top W^{1-2/r} A \preceq A^\top W^{1-2/p} A \preceq g_\alpha \cdot m^\alpha \cdot A^\top W^{1-2/r} A.$$

*Proof.* We have

$$A^\top W^{1-2/r} A \preceq A^\top W^{1-2/p} A,$$

where the step holds since $r \geq p$ and $w_i \in (0, 1]$ for all $i \in [m]$ we have that $w_i^{1-2/r} \leq w_i^{1-2/p}$ for all $i \in [m]$.

To prove the other direction, let $\epsilon \in (0, 1)$ be a positive real number.

Let $I_{w \leq \frac{\epsilon}{m}} \in \mathbb{R}^{m \times m}$ be the diagonal matrix where $I_{\epsilon,i,i} = 1$ if $w_i > \frac{\epsilon}{m}$ and $I_{\epsilon,i,i} = 0$ otherwise.

Let $I_{w > \frac{\epsilon}{m}} := I - I_{w \leq \frac{\epsilon}{m}}$.

Using Fact C.21, we have

$$A^\top W^{1-2/p} A \preceq \frac{1}{1 - \epsilon} A^\top W^{1-2/p} I_{w > \frac{\epsilon}{m}} A, \tag{32}$$

Since $r \geq p > 0$ and $w \in \mathbb{R}_{>0}^m$, we have $(\frac{m}{\epsilon})^{-2/p} W^{-2/p} \preceq (\frac{m}{\epsilon})^{-2/r} W^{-2/r}$, which further implies that

$$(\frac{m}{\epsilon})^{-2/p} A^\top W^{1-2/p} I_{w > \frac{\epsilon}{m}} A \preceq (\frac{m}{\epsilon})^{-2/r} A^\top W^{1-2/r} I_{w > \frac{\epsilon}{m}} A.$$

Thus, multiplying $(m/\epsilon)^{2/p}$ on the both sides of the above equation, we have

$$A^\top W^{1-2/p} I_{w > \frac{\epsilon}{m}} A \preceq (\frac{m}{\epsilon})^{2/p-2/r} A^\top W^{1-\frac{2}{r}} I_{w > \frac{\epsilon}{m}} A. \tag{33}$$

Combining Eq. (32) and Eq. (33), we can obtain that

$$\begin{aligned}
A^\top W^{1-2/p} A &\preceq \frac{1}{1 - \epsilon} A^\top W^{1-2/p} I_{w > \frac{\epsilon}{m}} A \\
&\preceq \frac{1}{1 - \epsilon} (\frac{m}{\epsilon})^{2/p-2/r} A^\top W^{1-2/r} A \\
&= \frac{1}{1 - \epsilon} (\frac{m}{\epsilon})^\alpha A^\top W^{1-2/r} A \\
&= \frac{(1 + \alpha)^{1+\alpha}}{\alpha^\alpha} m^\alpha A^\top W^{1-2/r} A \\
&= g_\alpha \cdot m^\alpha \cdot A^\top W^{1-2/r} A,
\end{aligned}$$

where the first step follows from Eq. (32), the second step follows from Eq. (33), the third step follows from $\alpha = 2/p - 2/r$, the fourth step follows from choosing $\epsilon = \frac{\alpha}{1+\alpha}$ to be the minimizer of function $f(\epsilon) = \frac{1}{(1-\epsilon)\epsilon^\alpha}$ (see Fact C.12), and the last step follows from the definition of $g_\alpha$ and basic algebra.

$\square$

**Lemma E.8** (Lemma 26 in page 23 in (Lee & Sidford, 2019)). *If the following conditions hold:*

- *Let $A \in \mathbb{R}^{m \times n}$ be a non-degenerate matrix.*

- *Define $w := w_p(A)$ for $0 < p < r$.*

- *Define $\alpha := 2/p - 2/r$.*

- *Let $g_\alpha := (1 + \alpha)(1 + 1/\alpha)^\alpha$.*

- *Define $c_{p,r,m} := (g_\alpha)^{\frac{1}{1+\alpha}} \cdot m^{\frac{\alpha}{1+\alpha}}$.*

*Then we have*

$$\sigma(W^{1/2 - 1/r} A)_i w_i^{-1} \le c_{p,r,m} \le 2m^{\frac{\alpha}{1+\alpha}}.$$

*Proof.* We have

$$A^\top W^{1 - 2/r} A \succeq (g_\alpha m^\alpha)^{-1} \cdot A^\top W^{1 - 2/p} A,$$

where the step follows from Lemma E.7.

Taking the inverse of the above equation on both sides, we get

$$(A^\top W^{1 - 2/r} A)^{-1} \preceq (g_\alpha m^\alpha) \cdot (A^\top W^{1 - 2/p} A)^{-1}.$$

For all $i \in [m]$ it follows that

$$
\begin{aligned}
& e_i^\top A (A^\top W^{1 - 2/r} A)^{-1} A^\top e_i w_i^{-2/r} \\
& \le (g_\alpha m^\alpha) \cdot e_i^\top A (A^\top W^{1 - 2/p} A)^{-1} A^\top e_i w_i^{-2/r} \\
& = (g_\alpha m^\alpha) \cdot e_i^\top w_i^{2/p-1} w_i^{1/2-1/p} A (A^\top W^{1 - 2/p} A)^{-1} A^\top w_i^{1/2-1/p} e_i w_i^{-2/r} \\
& = (g_\alpha m^\alpha) \cdot w_i^{2/p-1} \sigma_i (W^{1/2 - 1/p} A) w_i^{-2/r} \\
& = (g_\alpha m^\alpha) \cdot w_i^{2/p-1} \cdot w_i \cdot w_i^{-2/r} \\
& = (g_\alpha m^\alpha) \cdot w_i^\alpha, \tag{34}
\end{aligned}
$$

where the first step follows from multiplying $e_i^\top \cdot e_i w_i^{-2/r}$ on both sides of previous equation, the second step follows from $w_i^{2/p-1} w_i^{1/2-1/p} w_i^{1/2-1/p} = 1$, the third step follows from Fact C.31, the fourth step follows from $w = \sigma(W^{1/2 - 1/p} A)$, and the last step follows definition of $\alpha = 2/p - 2/r$.

For any fixed index $i \in [m]$,

$$
\begin{aligned}
A^\top W^{1 - \frac{2}{r}} A &= \sum_{j=1}^m w_j^{1 - \frac{2}{r}} A^\top e_j e_j^\top A \\
&\succeq w_i^{1 - \frac{2}{r}} A^\top e_i e_i^\top A, \tag{35}
\end{aligned}
$$

where the first step follows from $\sum_{j=1}^m w_j^{1 - 2/r} e_j e_j^\top = W^{1 - 2/r}$ (see Fact C.1), the second step follows from $w_j^{1 - \frac{2}{r}} A^\top e_j e_j^\top A$ is positive semidefinite for $j \in [m]$.

Thus, we have

$$
\begin{aligned}
w_i^{1 - \frac{2}{r}} e_i^\top A (A^\top W^{1 - \frac{2}{r}} A)^{-1} A^\top e_i &= (W^{1/2 - 1/r} A (A^\top W^{1 - \frac{2}{r}} A)^{-1} W^{1/2 - 1/r} A^\top)_{i,i} \\
&= \sigma_i (W^{1/2 - 1/r} A) \\
&\le 1,
\end{aligned}
$$

where the first step follows from Fact C.8, the second step follows from the definition of leverage score $\sigma(\cdot)$, and the last step follows from Part 2 of Lemma D.17.

Therefore, for $i \in [m]$,

$$e_i^\top A (A^\top W^{1 - \frac{2}{r}} A)^{-1} A^\top e_i w_i^{-\frac{2}{r}} \le w_i^{-1}, \tag{36}$$

For $i \in [m]$, we have

$$
\begin{aligned}
\sigma(W^{1/2-1/r}A)_i w_i^{-1} &= e_i^\top A(A^\top W^{1-2/r}A)^{-1}A^\top e_i w_i^{-2/r} \\
&\leq \min\{g_\alpha m^\alpha w_i^\alpha, w_i^{-1}\} \\
&\leq (g_\alpha m^\alpha)^{\frac{1}{1+\alpha}} \\
&= c_{p,r,m},
\end{aligned}
\tag{37}
$$

where the first step follows from definition of $\sigma(W^{1/2-1/r}A)$ and basic algebra, the second step follows from combining Eq. (34) and Eq. (36), the third step follows from Fact C.11, and the last step follows from the definition of $c_{p,r,m}$.

The fact that if we let

$$
f(\alpha) := (g_\alpha)^{\frac{1}{1+\alpha}} = (1+\alpha)^{\frac{1}{1+\alpha}}(1+\frac{1}{\alpha})^{\frac{\alpha}{1+\alpha}},
$$

then for all $\alpha \geq 0$,

$$
\begin{aligned}
\log f(\alpha) &= \frac{\log(1+\alpha)}{1+\alpha} + \frac{\alpha \cdot \log(1+(1/\alpha))}{1+\alpha} \\
&\leq \log(\frac{1+\alpha}{1+\alpha} + \frac{\alpha \cdot (1+(1/\alpha))}{1+\alpha}) \\
&= \log 2,
\end{aligned}
$$

where the first step follows from basic algebra, the second step follows from the concavity of $\log$ ($\gamma f(a) + (1-\gamma)f(b) \leq f(\gamma a + (1-\gamma)b)$ for any concave function $f$), and the last step follows from basic algebra.

Therefore, we have

$$
c_{p,r,m} \leq 2m^{\frac{\alpha}{1+\alpha}},
$$

where the step follows from Eq. (37) and $f(\alpha) \leq 2$. $\qquad\square$

### E.5 GRADIENT AND HESSIAN OF VOLUMETRIC POTENTIAL

**Lemma E.9** (Gradient and Hessian of Volumetric Potential, Lemma 24 in page 20 in (Lee & Sidford, 2019)). *If the following conditions hold:*

- *For all non-degenerate $A \in \mathbb{R}^{m \times n}$.*

- *Let $w \in \mathbb{R}_{>0}^m$.*

- *Let $p > 0$ with $p \neq 2$.*

- *Define $W := \mathrm{diag}(w) \in \mathbb{R}^{m \times m}$.*

- *Define $\sigma_w := \sigma(W^{1/2-1/p}A)$.*

- *Define $\Sigma_w := \Sigma(W^{1/2-1/p}A)$.*

- *Define $\Lambda_w := \Lambda(W^{1/2-1/p}A)$.*

- *Define*

$$
\mathcal{V}_p^A(w) := -\frac{1}{1-2/p}\log\det(A^\top W^{1-2/p}A).
$$

*Then, we have*

- **Part 1.**

$$
\nabla\mathcal{V}_p^A(w) = -W^{-1}\sigma_w.
$$

- **Part 2.**

$$\nabla^2 \mathcal{V}_p^A(w) = W^{-1}(\Sigma_w - (1 - 2/p)\Lambda_w)W^{-1}.$$

- **Part 3.** $\mathcal{V}_p^A$ *is convex in w and*

$$\frac{2}{\max\{p, 2\}} \cdot W^{-1}\Sigma_w W^{-1} \preceq \nabla^2 \mathcal{V}_p^A(w) \preceq \frac{2}{\min\{p, 2\}} \cdot W^{-1}\Sigma_w W^{-1}.$$

*Proof.* **Proof of Part 1.**

The formula for $\nabla \mathcal{V}_p^A(w)$ follow from Part 17 of Lemma D.16.

**Proof of Part 2.**

The formula for $\nabla^2 \mathcal{V}_p^A(w)$ follow from Part 19 of Lemma D.16.

**Proof of Part 3.**

Recall that $\Lambda := \Sigma - P^{\circ 2}$.

We have $0 \preceq \Lambda_w \preceq \Sigma_w$ by Part 2 of Lemma D.17.

If $p < 2$ then $(1 - 2/p) < 0$, then we have

$$\Sigma_w - (1 - 2/p)\Lambda_w \succeq \Sigma_w.$$

where the first step follows $\Lambda_w \succeq 0$.

And, we also have

$$\Sigma_w - (1 - 2/p)\Lambda_w \preceq \Sigma_w - (1 - 2/p)\Sigma_w = \frac{2}{p}\Sigma_w,$$

where the first step follows $\Lambda_w \preceq \Sigma_w$.

Applying the Fact C.25 to above two equations, then we have

$$W^{-1}\Sigma_w W^{-1} \preceq \nabla^2 \mathcal{V}_p^A(w) \preceq \frac{2}{p}W^{-1}\Sigma_w W^{-1}.$$

If $p > 2$ then $(1 - 2/p) > 0$, then we have

$$\Sigma_w - (1 - 2/p)\Lambda_w \succeq \frac{2}{p}\Sigma_w,$$

where the first step follows from $\Sigma_w \succeq \Lambda_w$.

Then we also can show

$$\Sigma_w - (1 - 2/p)\Lambda_w \preceq \Sigma_w,$$

where the first step follows from $\Lambda_w \succeq 0$.

Applying the Fact C.25 to above two equations, then we have

$$\frac{2}{p}W^{-1}\Sigma_w W^{-1} \preceq \nabla^2 \mathcal{V}_p^A(w) \preceq W^{-1}\Sigma_w W^{-1}.$$

Thus, we complete the proof. $\square$

### E.6 HESSIAN APPROXIMATION

**Fact E.10.** *If the following conditions hold:*

- *Define $\epsilon \in (0, 1)$.*

- *Define $f(p) := (\frac{1+\epsilon}{1-\epsilon})^{|1-2/p|}$.*

- *For all $i \in [m]$, we assume that $v_i, w_i > 0$.*

- *For all $i \in [m]$, let $(1 - \epsilon)v_i \leq w_i \leq (1 + \epsilon)v_i$.*

*Then we have*

$$\Sigma(W^{1/2-1/p}A) \in [f(p), f(p)^{-1}] \cdot \Sigma(V^{1/2-1/p}A).$$

*Proof.* For $i \in [m]$, We have

$$
\begin{aligned}
\Sigma(W^{1/2-1/p}A)_{i,i} &= (W^{1/2-1/p}A(A^\top W^{1-2/p}A)^{-1}A^\top W^{1/2-1/p})_{i,i} \\
&\leq (1 + \epsilon)^{|1-2/p|}(V^{1/2-1/p}A(A^\top W^{1-2/p}A)^{-1}A^\top V^{1/2-1/p})_{i,i} \\
&\leq \frac{(1 + \epsilon)^{|1-2/p|}}{(1 - \epsilon)^{|1-2/p|}}(V^{1/2-1/p}A(A^\top W^{1-2/p}A)^{-1}A^\top V^{1/2-1/p})_{i,i} \\
&\leq f(p)(V^{1/2-1/p}A(A^\top V^{1-2/p}A)^{-1}A^\top V^{1/2-1/p})_{i,i} \\
&= f(p)\Sigma(V^{1/2-1/p}A)_{i,i},
\end{aligned}
$$

where the first step follows from the definition of $\Sigma(W^{1/2-1/p}A)$, the second step follows from $(1 - \epsilon)v_i \leq w_i \leq (1 + \epsilon)v_i$ for $i \in [m]$, the third step follows from Fact C.20, the fourth step follows from the definition of $f(p)$, and the last step follows from the definition of $\Sigma(V^{1/2-1/p}A)$.

Similarly, we can show

$$\Sigma(W^{1/2-1/p}A)_{i,i} \geq f(p)^{-1}\Sigma(V^{1/2-1/p}A)_{i,i}.$$

Thus, the proof is complete. $\square$

**Lemma E.11** (Hessian Approximation, Lemma 53 in page 50 in (Lee & Sidford, 2019)). *If the following conditions hold:*

- *Denote the optimal point of Lewis weight as $w_p$.*

- *Define $\epsilon := \frac{p}{8(p+2)}$. (it implies that $\epsilon \in (0, 0.125)$).*

- *Let $w \in \mathbb{R}^m_{\geq 0}$ satisfies $\|W^{-1}(w_p - w)\|_\infty \leq \epsilon$ for the matrix $W := \mathrm{diag}(w)$.*

- *Define $V := \mathrm{diag}(w_p)$.*

*Then, we have*

$$\min\{1/2, 1/p\}W^{-1} \preceq \nabla^2 \mathcal{V}_p^A(w) \preceq \max\{2, 4/p\}W^{-1}.$$

*Proof.* Since we $V := \mathrm{diag}(w_p)$, then it is obvious that $V = \Sigma(V^{1/2-1/p}A)$.

For $i \in [m]$, we have

$$(1 - \epsilon)w_{p,i} \leq w_i \leq (1 + \epsilon)w_{p,i},$$

where the step follows from $\|W^{-1}(w_p - w)\|_\infty \leq \epsilon$ (it implies $|\frac{w_{p,i}-w_i}{w_i}| \leq \epsilon$).

By definition of $V = \mathrm{diag}(w_p)$ in lemma statement, we have

$$(1 - \epsilon)v_i \leq w_i \leq (1 + \epsilon)v_i, \tag{38}$$

where $\epsilon \in (0, 0.2)$ (see condition in Lemma statement).

We define

$$f(p) := \frac{(1 + \frac{p}{8(p+2)})^{|1-2/p|}}{(1 - \frac{p}{8(p+2)})^{|1-2/p|}}.$$

Using Fact C.10, we know that $f(p) \geq 1$.

Then, we have

$$
\begin{aligned}
\Sigma_w &= \Sigma(W^{1/2-1/p}A) \\
&\preceq f(p)\Sigma(V^{1/2-1/p}A) \\
&= f(p)V \\
&\preceq V \\
&\preceq 2W,
\end{aligned}
$$

where the first step follows from the definition of $\Sigma_w$, the second step follows from Eq. (38) and Fact E.10, the third step follows from $V = \Sigma(V^{1/2-1/p}A)$, the fourth step follows from Fact C.10 (it gives $f(p) \leq 1$), and the last step follows from Eq. (38).

And

$$
\begin{aligned}
\Sigma_w &= \Sigma(W^{1/2-1/p}A) \\
&\succeq f(p)^{-1}\Sigma(V^{1/2-1/p}A) \\
&= f(p)^{-1}V \\
&\succeq V \\
&\succeq 0.5W,
\end{aligned}
$$

where the first step follows from the definition of $\Sigma_w$, the second step follows from Eq. (38), the third step follows from $V = \Sigma(V^{1/2-1/p}A)$, the fourth step follows from Fact C.10 (it gives $f(p) \leq 1$), and the last step follows from Eq. (38).

Then the result follows from Lemma E.9. $\qquad\square$

### E.7 COMPUTE THE WEIGHT FUNCTION

**Lemma E.12** (Lemma 54 in page 51 in (Lee & Sidford, 2019)). *If the following conditions hold:*

- *Define $r := \frac{p}{20(p+2)}$.*

- *Let $w(0) \in \mathbb{R}^m_{>0}$ such that $\|W(0)^{-1}(w_p - w(0))\|_\infty \leq r$.*

- *Let $w(0)$ satisfy that $\langle w(0), \mathbf{1}_m \rangle = n$.*

- *Use* MEDIAN$(x, y, z)_i$ *to denote the median of $x_i, y_i$ and $z_i$ for all $i \in [m]$.*

- *Define $L := \max\{4, 8/p\}$.*

- *For all $k \geq 0$,*

$$
w(k+1) := \text{MEDIAN}((1-r)w(0), w(k) - \frac{1}{L}(w(0) - \frac{w(0)}{w(k)}\sigma(W(k)^{1/2-1/p}A)), (1+r)w(0)).
$$

*Then, for all $k$, we have*

$$
\|w(k) - w_p\|_{W_p^{-1}} \leq 2\sqrt{n} \cdot (1 - \frac{1}{16(p/2 + 2/p)})^{k/2}\|W(0)^{-1}(w_p - w(0))\|_\infty.
$$

*Proof.* We define

$$
Q := \{w \in \mathbb{R}^m : \|W(0)^{-1}(w - w(0))\|_\infty \leq r\}.
$$

Recall Theorem C.32, we have the following iterative step for an arbitrary positive definite matrix $H$:

$$
w(k+1) = \arg\min_{w \in Q} \nabla f(w(k))^\top(w - w(k)) + \frac{L}{2}\|w - w(k)\|_H^2. \tag{39}
$$

We consider the optimization problem $\min_{w_i > 0} \mathcal{V}_p^A(w) + \sum_{i=1}^m w_i$.

Using Part 1 of Lemma E.9, we know $\nabla \mathcal{V}_p^A(w) = -W^{-1}\sigma$.

It is obvious that we should choose $H = W(0)^{-1}$ when applying Theorem C.32.

Thus, let $f(w) = \mathcal{V}_p^A(w) + \sum_{i=1}^m w_i$, we have $\nabla f(w) = -W^{-1}\sigma + \mathbf{1}_m$.

We use $\mathbf{1}_m \in \mathbb{R}^m$ to denote the vector whose all entries are 1.

We define

$$a := \mathbf{1}_m - w(k)^{-1} \circ \sigma(W(k)^{1/2-1/p}A).$$

It is easy to see that

$$\langle \nabla f(w(k)), w - w(k) \rangle$$
$$= \langle \mathbf{1}_m - w(k)^{-1} \circ \sigma(W(k)^{1/2-1/p}A), w - w(k) \rangle$$
$$= \langle \mathbf{1}_m - w(k)^{-1} \circ \sigma(W(k)^{1/2-1/p}A), w \rangle - \langle \mathbf{1}_m - w(k)^{-1} \circ \sigma(W(k)^{1/2-1/p}A), w(k) \rangle$$
$$= \langle a, w \rangle - \langle \mathbf{1}_m - w(k)^{-1} \circ \sigma(W(k)^{1/2-1/p}A), w(k) \rangle.$$

Because the second term in above equation does not depend on $w$. Thus taking the argmin of the both side, we have

$$\arg\min_w \langle \nabla f(w(k)), w - w(k) \rangle = \arg\min_w \langle a, w \rangle.$$

Then, we have

$$w(k+1) = \arg\min_{w \in Q} \langle a, w \rangle + \frac{L}{2}\|w - w(k)\|_{W(0)^{-1}}^2$$

$$= \arg\min_{w \in Q} \langle a, w \rangle + \frac{L}{2}\langle w, W(0)^{-1}w \rangle - L\langle w, W(0)^{-1}w(k) \rangle + \frac{L}{2}\langle w(k), W(0)^{-1}w(k) \rangle$$

$$= \arg\min_{w \in Q} \langle a, w \rangle + \frac{L}{2}\langle w, W(0)^{-1}w \rangle - L\langle w, W(0)^{-1}w(k) \rangle$$

$$= \arg\min_{w \in Q} \frac{L}{2}\langle w, W(0)^{-1}w \rangle - L\langle W(0)^{-1}w(k) - \frac{a}{L}, w \rangle$$

$$= \arg\min_{w \in Q} \|w - W(0)(W(0)^{-1}w(k) - \frac{a}{L}\mathbf{1}_m)\|_{W(0)^{-1}}^2$$

$$= \arg\min_{w \in Q} \|w - w(k) + \frac{1}{L}(w(0) - \frac{w(0)}{w(k)}\sigma(W(k)^{1/2-1/p}A))\|_{W(0)^{-1}}^2,$$

where the first step follows from the previous equation and Eq. (39), the second step follows from expanding the inner product, the third step follows from the fact that the last term is constant with respect to $w$, the fourth step follows from basic algebra, the fifth step follows from Fact C.33 (by treating $B = W(0)^{-1}$ and $b = W(0)^{-1}w(k) - \frac{a}{L}\mathbf{1}_m$), and the last step follows from the definition of $a$.

We have

$$\nabla^2 \mathcal{V}(w) \preceq \max\{2, \frac{4}{p}\}W^{-1}$$

$$\preceq \max\{4, \frac{8}{p}\}W(0)^{-1}, \tag{40}$$

where the first step follows from Lemma E.11, the second step follows from that $w_i = (1 \pm 0.2)w_{p,i}$ and $w_{p,i} = (1 \pm 0.2)w(0)_i$.

And

$$\nabla^2 \mathcal{V}(w) \succeq \min\{1/2, 1/p\}W^{-1}$$

$$\succeq \min\{\frac{1}{4}, \frac{1}{2p}\}W(0)^{-1}, \tag{41}$$

where the first step follows from Lemma E.11, the second step follows from that $w_i = (1 \pm 0.2)w_{p,i}$ and $w_{p,i} = (1 \pm 0.2)w(0)_i$)

Then we have

$$\min\{\frac{1}{4}, \frac{1}{2p}\}W(0)^{-1} \preceq \nabla^2 \mathcal{V}(w) \preceq \max\{4, \frac{8}{p}\}W(0)^{-1},$$

where the step follows from Eq. (40) and Eq. (41).

Hence, we have

$$\|w(k) - w_p\|_{W(0)^{-1}}^2 \leq (1 - \frac{\min\{1/4, 1/(2p)\}}{\max\{4, 8/p\}})^k \|w(0) - w_p\|_{W(0)^{-1}}^2$$

$$\leq (1 - \frac{1}{16(p/2 + 2/p)})^k \|w(0) - w_p\|_{W(0)^{-1}}^2, \tag{42}$$

where the first step follows from Theorem C.32, the second step follows from Fact C.15.

$$\|w(0) - w_p\|_{W(0)^{-1}}^2 = (w(0) - w_p)^\top W(0)^{-1}(w(0) - w_p)$$

$$= \sum_{i=1}^m \frac{(w(0) - w_p)_i^2}{w(0)_i}$$

$$= \sum_{i=1}^m w(0)_i (\frac{(w(0) - w_p)_i}{w(0)_i})^2$$

$$\leq \sum_{i=1}^m w(0)_i \|W(0)^{-1}(w_p - w(0))\|_\infty^2$$

$$\leq 2n \|W(0)^{-1}(w_p - w(0))\|_\infty^2, \tag{43}$$

where the first step follows from the definition of $\| \cdot \|_W$, the second step follows from $W(0)$ is diagonal matrix, the third step follows from multiplying and dividing same factor, the fourth step follows from the definition of $\| \cdot \|_\infty$, and the last step follows from $\sum_{i=1}^m w_i(0) = n$ (see Lemma statement).

Combining Eq. (42) and Eq. (43) gives

$$\|w(k) - w_p\|_{W_p^{-1}}^2 \leq 4n \cdot (1 - \frac{1}{16(p/2 + 2/p)})^k \|W(0)^{-1}(w_p - w(0))\|_\infty^2.$$

Thus, taking the square root of both sides completes the proof.

$\square$

### E.8 MULTIPLICATIVE APPROXIMATION OF $w_p$

**Lemma E.13** (Lemma 55 in page 51 in (Lee & Sidford, 2019)). *If the following conditions hold:*

- *Given $w \in \mathbb{R}^m$ such that $\|W_p^{-1}(w_p - w)\|_\infty \leq \frac{p}{8(p+2)}$.*

- *Let $\beta := 4(1 + 2/p)^2 \sqrt{n}$ denote a local variable only be used in this lemma.*

- *Given $w \in \mathbb{R}^m$ such that $\|w - w_p\|_{W_p^{-1}} \leq 0.1/\beta$.*

- *Let $\delta := \|w - w_p\|_{W_p^{-1}}$.*

- *Let $\widehat{w} := (\mathrm{Diag}(A(A^\top W^{1-2/p}A)^{-1}A^\top))^{2/p}$.*

- *Let $w_p$ be defined as $w_p := (\mathrm{Diag}(A(A^\top W_p^{1-2/p}A)^{-1}A^\top))^{2/p}$.*

- *Let $\theta := 2 \cdot |1 - 2/p| \cdot \sqrt{n}$.*

*Then, we have*

$$\|W_p^{-1}(\widehat{w} - w_p)\|_\infty \leq \beta \cdot \|w - w_p\|_{W_p^{-1}}.$$

*Proof.* To show $\widehat{w}$ is multiplicative close to $w_p$ (i.e., $\widehat{w}_i = (1 \pm 0.1)w_{p,i}$), it suffices to prove that $A^\top W_p^{1-2/p} A$ is multiplicatively close to $A^\top W^{1-2/p} A$.

Firstly, to simplify computation, we define

$$\alpha := \text{tr}[(A^\top W_p^{1-2/p} A)^{-1}(A^\top |W^{1-2/p} - W_p^{1-2/p}|A)].$$

Using Fact C.24, then we have

$$A^\top W^{1-2/p} A \in [(1-\alpha), (1+\alpha)] \cdot A^\top W_p^{1-2/p} A. \tag{44}$$

Taking the inverse of the above equation

$$(A^\top W^{1-2/p} A)^{-1} \in [(1+\alpha)^{-1}, (1-\alpha)^{-1}] \cdot (A^\top W_p^{1-2/p} A)^{-1}.$$

Multiplying $e_i^\top A \cdot A^\top e_i$, we have

$$(A(A^\top W^{1-2/p} A)^{-1} A^\top)_{i.i} \in [(1+\alpha)^{-1}, (1-\alpha)^{-1}](A(A^\top W_p^{1-2/p} A)^{-1} A^\top)_{i,i}.$$

Taking the power of $2/p$ on both sides

$$(A(A^\top W^{1-2/p} A)^{-1} A^\top)_{i,i}^{2/p} \in [(1+\alpha)^{-2/p}, (1-\alpha)^{-2/p}](A(A^\top W_p^{1-2/p} A)^{-1} A^\top)_{i,i}^{2/p}.$$

Re-organizing the above equation, we get

$$\frac{(A(A^\top W^{1-2/p} A)^{-1} A^\top)_{i,i}^{2/p}}{(A(A^\top W_p^{1-2/p} A)^{-1} A^\top)_{i,i}^{2/p}} \in [(1+\alpha)^{-2/p}, (1-\alpha)^{-2/p}]. \tag{45}$$

Next, we can rewrite $\alpha$ as follows:

$$\begin{aligned}
\alpha &= \text{tr}[(A^\top W_p^{1-2/p} A)^{-1}(A^\top |W^{1-2/p} - W_p^{1-2/p}|A)] \\
&= \text{tr}[(A^\top W_p^{1-2/p} A)^{-1} A^\top \cdot W^{1/2-1/p} W^{-1+2/p} \cdot |W^{1-2/p} - W_p^{1-2/p}| \cdot W^{1/2-1/p} \cdot A] \\
&= \text{tr}[W_p^{1/2-1/p} A (A^\top W_p^{1-2/p} A)^{-1} A^\top W_p^{1/2-1/p} \cdot W_p^{-1+2/p} \cdot |W^{1-2/p} - W_p^{1-2/p}|] \\
&= \text{tr}[P(W_p^{1/2-1/p} A) \cdot W_p^{-1+2/p} \cdot |W^{1-2/p} - W_p^{1-2/p}|] \\
&= \sum_{i=1}^m (P(W_p^{1/2-1/p} A) W_p^{-1+2/p} |W^{1-2/p} - W_p^{1-2/p}|)_{i,i} \\
&= \sum_{i=1}^m \frac{P(W_p^{1/2-1/p} A)_{i,i}}{w_{p,i}^{1-2/p}} \cdot |w_i^{1-2/p} - w_{p,i}^{1-2/p}| \\
&= \sum_{i=1}^m \frac{\sigma(W_p^{1/2-1/p} A)_i}{w_{p,i}^{1-2/p}} \cdot |w_i^{1-2/p} - w_{p,i}^{1-2/p}|, \tag{46}
\end{aligned}$$

where the first step follows from definition of $\alpha$, the second step follows from $WW^{-1} = I$ and $W, W_p$ are diagonal matrices, the third step follows from trace cyclic property, the fourth step follows definition of $P(W_p^{1/2-1/p} A)$, the fifth step follows from the definition of trace, the sixth step follows from the $(P \text{diag}(w))_{i,i} = P_{i,i} w_i$, and the last step follows from $P(X)_{i,i} = \sigma(X)_i$ for $i \in [m]$.

Since $\|W_p^{-1}(w_p - w)\|_\infty \leq \frac{p}{8(p+2)}$, we have that for all $i \in [m]$,

$$|w_i^{1-2/p} - w_{p,i}^{1-2/p}| \leq 2 \cdot |1 - 2/p| \cdot |\frac{w_i - w_{p,i}}{w_{p,i}^{2/p}}|, \tag{47}$$

where the step follows from the mean-value theorem $|f(x) - f(y)| \leq |f'(x)| \cdot |x - y|$.

Therefore, we obtain

$$
\begin{aligned}
\alpha &\leq 2 \cdot |1 - 2/p| \cdot \sum_{i=1}^{m} \frac{\sigma(W_p^{1/2 - 1/p} A)_i}{w_{p,i}^{1 - 2/p}} \cdot |\frac{w_i - w_{p,i}}{w_{p,i}^{2/p}}| \\
&\leq 2 \cdot |1 - 2/p| \cdot (\sum_{i=1}^{m} \frac{\sigma(W_p^{1/2 - 1/p} A)_i^2}{w_{p,i}})^{1/2} \cdot (\sum_{i=1}^{m} \frac{(w_i - w_{p,i})^2}{w_{p,i}})^{1/2} \\
&= 2 \cdot |1 - 2/p| \cdot \sqrt{n} \cdot \delta \\
&= \theta \delta,
\end{aligned}
\tag{48}
$$

where the first step follows from Eq. (46) and Eq. (47), the second step follows from Cauchy-Schwarz, the third step follows from $\sum_{i=1}^{m} \frac{\sigma(W_p^{1/2 - 1/p} A)_i^2}{w_{p,i}} = \sum_{i=1}^{m} \frac{w_{p,i}^2}{w_{p,i}} = n$ (since $w_p$ is a Lewis Weight), and definition of $\delta = \|w - w_p\|_{W_p}^{-1} = (\sum_{i=1}^{m} (w_i - w_{p,i})^2 / w_{p,i})^{1/2}$ (see Lemma Statement), and the last step follows from definition of $\theta$.

Note that for all $\delta \leq (0, 0.1/\beta]$ where $\beta = 4(1 + 2/p)^2 \sqrt{n}$.

Using Fact C.30, we have

$$
(1 - \theta\delta)^{-2/p} \leq 1 + \beta\delta \text{ and } (1 + \theta\delta)^{-2/p} \geq 1 - \beta\delta.
\tag{49}
$$

Then we have

$$
\begin{aligned}
\|W_p^{-1}(\widehat{w} - w_p)\|_\infty &= \max_{i \in [m]} |w_{p,i}^{-1}(\widehat{w}_i - w_{p,i})| \\
&= \max_{i \in [m]} |\frac{(A(A^\top W^{1-2/p} A)^{-1} A^\top)_{i,i}^{2/p}}{(A(A^\top W_p^{1-2/p} A)^{-1} A^\top)_{i,i}^{2/p}} - 1| \\
&= \max\{(1 - \alpha)^{-2/p} - 1, 1 - (1 + \alpha)^{-2/p}\} \\
&\leq \max\{(1 - \theta\delta)^{-2/p} - 1, 1 - (1 + \theta\delta)^{-2/p}\} \\
&\leq \beta \cdot \delta \\
&= \beta \cdot \|w - w_p\|_{W_p^{-1}},
\end{aligned}
$$

where the first step follows from the definition of the infinity norm, the second step follows from the definition of $\widehat{w}$, the third step follows from Eq. (45), the fourth step follows from Eq. (48), the fifth step follows from Eq. (49), and the last step follows from the definition of $\delta$.

$\square$

## E.9 EXACT WEIGHT UPDATES

---
**Algorithm 2** Exact weight computation
---
1: **procedure** COMPUTEEXACTWEIGHT($A \in \mathbb{R}^{m \times n}, p \in \mathbb{N}^+, w(0) \in \mathbb{R}_{>0}^m, \epsilon \in (0,1)$)
2: $\quad T \leftarrow \lceil 32(p/2 + 2/p) \log(8n(1 + 2/p)\epsilon^{-1}) \rceil, r \leftarrow \frac{p}{20(p+2)}, L \leftarrow \max\{4, \frac{8}{p}\}$
3: $\quad$ **for** $k = 1, \ldots, T-1$ **do**
4: $\quad\quad w^{(k+1)} \leftarrow$ MEDIAN$((1 - r)w(0), w(k) - \frac{1}{L}(w(0) - \frac{w(0)}{w(k)}\sigma(W(k)^{1/2 - 1/p} A)), (1 + r)w(0))$
5: $\quad$ **end for**
6: $\quad$ **return** $(\text{Diag}(A(A^\top W(T)^{1-2/p} A)^{-1} A^\top))^{2/p}$ $\qquad\qquad \triangleright W(T) = \text{diag}(w(T))$
7: **end procedure**
---

The goal of this section is to prove Theorem E.14.

**Theorem E.14** (Exact Weight Updates, Theorem 56 in page 52 in (Lee & Sidford, 2019))**.** *If the following conditions hold:*

- *Let $\epsilon \in (0, 1)$.*

- *Let $w(0) \in \mathbb{R}^m_{>0}$ with $\|w(0)^{-1}(w_p(A) - w(0))\|_\infty \leq \frac{p}{20(p+2)}$.*

- $T = O((p + 1/p) \log(n(1 + 1/p)\epsilon^{-1}))$.

- $\|W(0)^{-1}(w_p - w(0))\|_\infty \leq \operatorname{poly}(n)$.

*Then we can show*

- **Part 1.** *Then the algorithm* COMPUTEEXACTWEIGHT$(A, p, w(0), \epsilon)$ *(Algorithm 2)*

  - *outputs $w \in \mathbb{R}^m_{>0}$ with $\|w_p(A)^{-1}(w_p(A) - w)\|_\infty \leq \epsilon$ in $T$ iterations.*
  - *Each iteration involves computing $\sigma(VA)$ for diagonal matrix $V$ and extra linear time work and $O(1)$ depth.*

*Proof.* We define $\delta_0 := \|W(0)^{-1}(w_p - w(0))\|_\infty$.

Then we have

$$\|W_p^{-1}(\widehat{w} - w_p)\|_\infty \leq 4(1 + 2/p)^2 \sqrt{n} \|w(k) - w_p\|_{W_p^{-1}}$$

$$\leq 8(1 + 2/p)^2 n (1 - \frac{1}{16(p/2 + 2/p)})^{k/2} \delta_0,$$

where the first step follows from Lemma E.13, the second step follows from Lemma E.12

For conveinent of writing proofs, we define

$$\alpha_1 := 8(1 + 2/p)^2,$$
$$\alpha_2 := 16(p/2 + 2/p).$$

Thus, we need to choose $k$ to make the following happen

$$\alpha_1 (1 - 1/\alpha_2)^{k/2} \delta_0 \leq \epsilon/n.$$

which is equivalent to

$$(1 - 1/\alpha_2)^{k/2} \leq \frac{\epsilon}{\delta_0 \alpha_1 n},$$

Note that

$$(1 - 1/\alpha_2)^{k/2} \leq e^{-0.5k/\alpha_2},$$

where the first step follows from the fact that $1 - x \leq e^{-x}$ for $x \in \mathbb{R}$.

Thus, as long as we can show

$$e^{-0.5k/\alpha_2} \leq \frac{\epsilon}{\delta_0 \alpha_1 n},$$

then we're done.

We can just choose

$$k \geq 2\alpha_2 \log(\delta_0 \alpha_1 n/\epsilon).$$

Note that, recall the definition of $\alpha_1$, $\alpha_2$ and $\delta_0 \leq \operatorname{poly}(n)$, thus we can show the number iterations $T$ to be

$$O((p + 1/p) \log(n(1 + 1/p)\epsilon^{-1})).$$

$\square$

---

**Algorithm 3** Approximate weight computation

---

1: **procedure** COMPUTEAPXWEIGHT($A \in \mathbb{R}^{m \times n}, p \in (0, 4), w(0) \in \mathbb{R}^m_{>0}, \epsilon \in (0, 2/p - |1 - 2/p|))$
2:      $L \leftarrow \max\{4, \frac{8}{p}\}, r \leftarrow \frac{p^2(4-p)}{2^{20}}, \delta \leftarrow \frac{(4-p)\epsilon}{256}$.
3:      $T \leftarrow \lceil 80(\frac{p}{2} + 2/p) \log(\frac{pn}{32\epsilon}) \rceil$.
4:                                                 $\triangleright$ $T$ is the number of iterations
5:      **for** $j = 1, \ldots, T - 1$ **do**
6:          Compute $\sigma(j) \in \mathbb{R}^n$ such that

$$e^{-\delta}\sigma(W(j)^{1/2-1/p}A)_i \leq \sigma(j)_i \leq e^{\delta}\sigma(W(j)^{1/2-1/p}A)_i \text{ for all } i \in [m].$$

7:          $w(j+1) = \text{MEDIAN}((1-r)w(0), w(j) - \frac{1}{L}(w(0) - w(j))\sigma(j), (1+r)w(0))$.
8:      **end for**
9:      **return** $(\text{Diag}(A(A^\top W(T)^{1/2-1/p})^{-1}A^\top))_{2/p}$.
10: **end procedure**

---

### E.10 APPROXIMATE WEIGHT COMPUTATION

In this section, we introduce how to use approximate leverage scores instead of exact leverage scores in computing gradient. First, we give a lemma showing that the optimality condition $\sigma(W^{1/2-1/p}A)_i/w_i$ is stable under changes to $w$.

**Lemma E.15** (Lemma 57 in page 53 in (Lee & Sidford, 2019)). *If the following conditions hold:*

- *Let $w, v \in \mathbb{R}^m_{>0}$ with $w_i = e^{\delta_i}v_i$ for $|\delta_i| \leq \delta$ for all $i \in [m]$.*

*Then, for all $i \in [m]$,*

$$v_i^{-1}\sigma(V^{1/2-1/p}A)_i \in [e^{\frac{2}{p}\delta_i - |1-2/p|\delta}, e^{\frac{2}{p}\delta_i + |1-2/p|\delta}] \cdot w_i^{-1}\sigma(W^{1/2-1/p}A)_i.$$

*Proof.* We have

$$
\begin{aligned}
v_i^{-1}\sigma(V^{1/2-1/p}A)_i &= v_i^{-2/p}a_i^\top(A^\top V^{1-2/p}A)^{-1}a_i \\
&= e^{\frac{2}{p}\delta_i}w_i^{-2/p}a_i^\top(A^\top V^{1-2/p}A)^{-1}a_i \\
&\leq e^{\frac{2}{p}\delta_i + |1-2/p|\delta}w_i^{-2/p}a_i^\top(A^\top W^{1-2/p}A)^{-1}a_i,
\end{aligned}
$$

where $a_i$ is the $i$-th row of $A$, the first step follows from the definition of $\sigma(\cdot)$, the second step follows from $w_i = e^{\delta_i}v_i$, the third step follows from $(A^\top V^{1-2/p}A)^{-1} \preceq e^{|1-2/p|\delta}(A^\top W^{1-2/p}A)^{-1}$ (implied by Fact C.19 and Fact C.20).

For the lower bound,

$$
\begin{aligned}
v_i^{-1}\sigma(V^{1/2-1/p}A)_i &= v_i^{-2/p}a_i^\top(A^\top V^{1-2/p}A)^{-1}a_i \\
&= e^{\frac{2}{p}\delta_i}w_i^{-2/p}a_i^\top(A^\top V^{1-2/p}A)^{-1}a_i \\
&\geq e^{\frac{2}{p}\delta_i - |1-2/p|\delta}w_i^{-2/p}a_i^\top(A^\top W^{1-2/p}A)^{-1}a_i,
\end{aligned}
$$

where $a_i$ is the $i$-th row of $A$, the first step follows from the definition of $\sigma(\cdot)$, the second step follows from $w_i = e^{\delta_i}v_i$, the third step follows from $(A^\top V^{1-2/p}A)^{-1} \succeq e^{-|1-2/p|\delta}(A^\top W^{1-2/p}A)^{-1}$ (implied by Fact C.19 and Fact C.20). $\square$

**Theorem E.16** (Approximate Weight Computation, Theorem 58 in page 53 in (Lee & Sidford, 2019)). *If the following conditions hold:*

- *Let $p \in (0, 4)$.*

- *Define $r := \frac{p^2(4-p)}{2^{20}}$.*

- *Let $w(0) \in \mathbb{R}^m_{>0}$ satisfy $\|w(0)^{-1}(w_p(A) - w(0))\|_\infty \leq r$.*

- *Let $\epsilon \in (0, 2/p - |1 - 2/p|)$.*

- *Define $L := \max\{4, 8/p\}$.*

- *Define $\delta := \frac{(4-p)\epsilon}{256}$. (it implies $\delta \in (0, 0.1)$)*

*Then we have*

- **Part 1.** *The algorithm* COMPUTEAPXWEIGHT $(x, w(0), \epsilon)$ *return $w$*

  - *such that $\|w_p(A)^{-1}(w_p(A) - w)\|_\infty \leq \epsilon$ in $O(p^{-1} \log(np^{-1}\epsilon^{-1}))$ steps.*
  - *Each step involves computing $\sigma$ up to $\pm\Theta((4 - p) \cdot \epsilon)$ multiplicative error with some extra linear time work.*

*Proof.* Consider an execution of COMPUTEAPXWEIGHT$(x, w(0), \epsilon)$ where there is no error in computing leverages scores, i.e. $\sigma(j) = \sigma(W(j)^{1/2-1/p}A)$, and let $v(j)$ denote the $w$ computed during this idealized execution of COMPUTEAPXWEIGHT.

We will show that $w(j)$ and $v(j)$ are multiplicatively close.

Suppose that for $i \in [m]$, $w_i(j) = e^{\delta_i^{(j)}} v(j)_i$ with $|\delta_i^{(j)}| \leq \delta^{(j)}$ for some $\delta^{(j)} \geq 0$.

We use $\pm\delta$ to denote a real value with magnitude at most $\delta$.

Define $\overline{v}(j + 1), \overline{w}(j + 1) \in \mathbb{R}^m_{>0}$ to be $v(j + 1)$ and $w(j + 1)$ before taking the median, i.e.

$$\overline{v}(j + 1) := v(j) - \frac{1}{L}(w(0) - \frac{w(0)}{v(j)}e^{\pm\delta}\sigma(V(j)^{1/2-1/p}A))$$

$$\overline{w}(j + 1) := w(j) - \frac{1}{L}(w(0) - \frac{w(0)}{w(j)}\sigma(j)). \tag{50}$$

We have

$$\overline{w}(j + 1)_i - \overline{v}(j + 1)_i$$

$$= w(j) - v(j) + \frac{w(0)}{L}(\frac{e^{\pm\delta}\sigma(W(j)^{1/2-1/p}A)}{w(j)} - \frac{\sigma(V(j)^{1/2-1/p}A)}{v(j)})$$

$$= (e^{\delta_i^{(j)}} - 1)v(j)_i + \frac{w(0)}{L}(\frac{e^{\pm\delta}\sigma(W(j)^{1/2-1/p}A)}{w(j)} - \frac{\sigma(V(j)^{1/2-1/p}A)}{v(j)})$$

$$= (e^{\delta_i^{(j)}} - 1)v(j)_i + \frac{w(0)}{L}(e^{-\frac{2}{p}\delta_i^{(j)}\pm|1-2/p|\delta^{(j)}\pm\delta} - 1) \cdot \frac{\sigma(V(j)^{1/2-1/p}A)}{v(j)}, \tag{51}$$

where the first step follows from Eq. (50), the second step follows from $w_i = e^{\delta_i^{(j)}} v(j)_i$, the third step follows from Lemma E.15.

Since

$$\|W(0)^{-1}(w(0) - v(j))\|_\infty \leq r$$

will imply that $w(0) = e^{\pm1.5r}v(j)$ (see Fact C.6) and that

$$\|W(0)^{-1}(w(0) - w_p(A))\|_\infty \leq r$$

will imply that $w(0) = e^{\pm1.5r}w_p(A)$ (see Fact C.6).

Combining the above two equations, we get the following

$$w_p(A) = e^{\pm3r}v(j). \tag{52}$$

Recall Lemma E.15 for $w_i = e^{\delta_i}v_i$ with $|\delta_i| \leq \delta$,

$$v_i^{-1}\sigma(V^{1/2-1/p}A)_i \in [e^{-(2/p+|1-2/p|)\delta}, e^{(2/p+|1-2/p|)\delta}] \cdot w_i^{-1}\sigma(W^{1/2-1/p}A)_i. \tag{53}$$

Then, rename the variables in Eq. (53) for $w = w_p(A)$ and $v = v(j)$ with $\delta = 3r$ (due to Eq. (52)), we have

$$(v(j)_i)^{-1}\sigma(V(j)^{1/2-1/p}A)_i \in [e^{-3(2/p+|1-2/p|)r}, e^{3(2/p+|1-2/p|)r}] \cdot w_i^{-1}\sigma(W^{1/2-1/p}A)_i.$$

Further, using $2/p + |1 - 2/p| \leq 1 + 4/p$ ($\forall p > 0$), the above equation will imply that

$$(v(j)_i)^{-1}\sigma(V(j)^{1/2-1/p}A)_i \in [e^{-3(1+1/p)r}, e^{3(1+4/p)r}] \cdot w_i^{-1}\sigma(W^{1/2-1/p}A)_i. \tag{54}$$

We have

$$\overline{w}(j+1)_i - \overline{v}(j+1)_i = (e^{\delta_i^{(j)}} - 1)v(j)_i + \frac{w(0)}{L}(e^{-\frac{2}{p}\delta_i^{(j)} \pm |1-2/p|\delta^{(j)} \pm \delta} - 1)e^{\pm 3(1+4/p)r},$$

where the step follows from combining Eq. (51) and Eq. (54).

The truncation means, that $w$ is taking the median of $\overline{w}$, $(1 - r)w(0)$ and $(1 + r)w(0)$. So if $\overline{w}$ is not inside $1 - r$ and $1 + r$ range, this can be viewed as a truncation. Since $w(j + 1)$ and $v(j + 1)$ are just truncation of $\overline{w}(j + 1)$ and $\overline{v}(j + 1)$, we have the same bound for $w(j + 1)_i - v(j + 1)_i$.

We get that

$$(e^{\delta_i^{(j+1)}} - 1)v_i(j+1) = (e^{\delta_i^{(j)}} - 1)v(j)_i + \frac{w(0)}{L}(e^{-\frac{2}{p}\delta_i^{(j)} \pm |1-\frac{2}{p}|\delta^{(j)} \pm \delta} - 1)e^{\pm 3(1+4/p)r},$$

where the step follows from $w_i(j + 1) = e^{\delta_i^{(j+1)}}v_i(j + 1)$.

Finally,

$$e^{\delta_i^{(j+1)}} - 1 = e^{\pm 4r}(e^{\delta_i^{(j)}} - 1) + \frac{1}{L}(e^{-\frac{2}{p}\delta_i^{(j)} \pm |1-\frac{2}{p}|\delta^{(j)} \pm \delta} - 1)e^{\pm 3(2+4/p)r}$$

$$:= A_1 + A_2, \tag{55}$$

where the first step follows from $v(j + 1) = e^{\pm 2r}w(0)$ and $v(j) = e^{\pm 2r}w(0)$, the second step follows from we define $A_1$ and $A_2$ in that way to simplify the proof.

Due to definition of $L$, we can show

$$L = \max\{4, 8/p\} \geq 2\max\{1, 4/p\} \geq 1 + 4/p. \tag{56}$$

**Bound $A_1$.** We can bound

$$A_1 = e^{\pm 4r}(e^{\delta_i^{(j)}} - 1)$$

$$= (e^{\delta_i^{(j)}} - 1) \pm 8r\delta_i^{(j)}$$

$$= \delta_i^{(j)} \pm (\delta_i^{(j)})^2 \pm 8r\delta_i^{(j)}$$

$$= \delta_i^{(j)} \pm 2r\delta^{(j)} \pm 8r\delta_i^{(j)}$$

$$= \delta_i^{(j)} \pm 10r\delta^{(j)}, \tag{57}$$

where the first step follows from the definition of $A_1$, the second step follows from $e^{\pm x}(e^y - 1) = (e^y - 1) \pm 2x|y|$ (see Fact C.29), the third step follows from $e^x - 1 = x \pm x^2$ (see Fact C.28), the fourth step follows from $\delta_i^{(j)} \leq \delta^{(j)} \leq 2r$, and the last step follows from $\delta_i^{(j)} \leq \delta^{(j)}$.

**Bound $A_2$.** For the convenience of writing proofs, we first define

$$y := -\frac{2}{p}\delta_i^{(j)} \pm |1 - \frac{2}{p}|\delta^{(j)} \pm \delta.$$

Then, we can show

$$|y| \leq (1 + 4/p)\delta^{(j)} + \delta, \tag{58}$$

where the step follows from $|\delta_i^{(j)}| \leq \delta^{(j)}$, $\delta > 0$ and triangle inequality.

We can bound

$$
\begin{aligned}
y^2 &\le ((1 + 4/p)\delta^{(j)} + \delta)^2 \\
&\le 2((1 + 4/p)\delta^{(j)})^2 + 2\delta^2 \\
&\le 2((1 + 4/p)\delta^{(j)})^2 + \delta \\
&\le 4(1 + 4/p)^2 r\delta^{(j)} + \delta,
\end{aligned}
\tag{59}
$$

where the first step follows from Eq. (58), the second step follows from $(a + b)^2 \le 2a^2 + 2b^2$, the third step follows from $\delta \in (0, 0.1)$, and the last step follows from $\delta^{(j)} \le 2r$.

Then we have

$$
\begin{aligned}
A_2 &= \frac{1}{L}(e^y - 1)e^{\pm 3(2 + 4/p)r} \\
&= \frac{1}{L}(e^y - 1) + \frac{6}{L}(2 + 4/p)r \cdot |y| \\
&:= A_{2,1} + A_{2,2},
\end{aligned}
$$

where the first step follows from the definition of $A_2$, the second step follows from Fact C.29, and the last step follows from we define $A_{2,1}$ and $A_{2,2}$ to simplify the proof.

We have

$$
\begin{aligned}
A_{2,1} &= \frac{1}{L}(e^y - 1) \\
&= \frac{1}{L}(y \pm (y)^2) \\
&= \frac{1}{L}(-\frac{2}{p}\delta_i^{(j)} \pm |1 - \frac{2}{p}|\delta^{(j)} \pm 2\delta \pm 4(1 + 4/p)^2 r\delta^{(j)}) \\
&= -\frac{2}{pL}\delta_i^{(j)} \pm \frac{1}{L}|1 - \frac{2}{p}|\delta^{(j)} \pm \frac{2\delta}{L} \pm \frac{4}{L}(1 + 4/p)^2 r\delta^{(j)} \\
&= -\frac{2}{pL}\delta_i^{(j)} \pm \frac{1}{L}|1 - \frac{2}{p}|\delta^{(j)} \pm \frac{2\delta}{L} \pm 4(1 + 4/p)r\delta^{(j)},
\end{aligned}
\tag{60}
$$

where the first step follows from the definition of $A_{2,1}$, the second step follows from $e^x - 1 = x \pm x^2$ for all $x \in (0, 0.5)$ (see Fact C.28), the third step follows from Eq. (59), the fourth step follows from basic algebra, and the last step follows from $L \ge 1 + 4/p$ (see Eq. (56)).

We have

$$
\begin{aligned}
A_{2,2} &= \frac{6}{L}(2 + 4/p)r|y| \\
&\le \frac{12}{L}(1 + 4/p)r|y| \\
&\le \frac{12}{L}(1 + 4/p)r((1 + 4/p)\delta^{(j)} + \delta) \\
&= \frac{12}{L}(1 + 4/p)r(1 + 4/p)\delta^{(j)} + \frac{12}{L}(1 + 2/p)r\delta \\
&\le 12(1 + 4/p)r\delta^{(j)} + \frac{12}{L}(1 + 2/p)r\delta \\
&\le 12(1 + 4/p)r\delta^{(j)} + \frac{\delta}{L},
\end{aligned}
\tag{61}
$$

where the first step follows from the definition of $A_{2,2}$, the second step follows from basic algebra, the third step follows from Eq. (58), the fourth step follows from basic algebra, the fifth step follows from $(1 + 4/p) \le L$ (see Eq. (56)), and the last step follows from $12(1 + 2/p)r \le 1$.

We have

$$
A_2 = A_{2,1} + A_{2,2}
$$

$$= -\frac{2}{pL}\delta_i^{(j)} \pm \frac{1}{L}|1 - \frac{2}{p}|\delta^{(j)} \pm \frac{3\delta}{L} \pm 20(1 + 4/p)r\delta^{(j)}, \tag{62}$$

where the first step follows from the definition of $A_{2,1}$ and $A_{2,2}$, and the second step follows from substituting $A_{2,1}$ and $A_{2,2}$ with Eq. (60) and Eq. (61).

We have

$$e^{\delta_i^{(j+1)}} - 1$$
$$= A_1 + A_2$$
$$= \delta_i^{(j)} \pm 10r\delta^{(j)} + A_2$$
$$= (1 - \frac{2}{pL})\delta_i^{(j)} \pm \frac{1}{L}|1 - \frac{2}{p}|\delta^{(j)} \pm \frac{3\delta}{L} \pm 40(1 + 4/p)r\delta^{(j)}, \tag{63}$$

where the first step follows from Eq. (55), the second step follows from Eq. (57), the third step follows from Eq. (62).

For the first two terms in Eq. (63), for $i \in [m]$,

$$|(1 - \frac{2}{pL})\delta_i^{(j)} \pm \frac{1}{L} \cdot |1 - 2/p| \cdot \delta^{(j)}| \le (1 - \frac{2}{pL} + \frac{1}{L} \cdot |1 - 2/p|)\delta^{(j)}, \tag{64}$$

where the step follows from triangle inequality.

For the last two terms in Eq. (63), we have

$$|\pm 40(1 + 4/p)r\delta^{(j)} \pm \frac{3\delta}{L}| \le 40(1 + 4/p)r\delta^{(j)} + \frac{3\delta}{L}, \tag{65}$$

where the step follows from triangle inequality.

We have

$$\delta^{(j+1)} \le e^{\delta_i^{(j+1)}} - 1$$
$$\le \text{LHS of Eq. (64)} + \text{LHS of Eq. (65)}$$
$$\le (1 - \frac{2}{pL} + \frac{1}{L} \cdot |1 - 2/p|)\delta^{(j)} + 40(1 + 4/p)r\delta^{(j)} + \frac{3\delta}{L}$$
$$= (1 - \frac{1}{L}(2/p - |1 - 2/p|) + 40(1 + 4/p)r)\delta^{(j)} + \frac{3\delta}{L}, \tag{66}$$

where the first step follows from $x \le e^x - 1$, the second step follows from triangle inequality and Eq. (63), the third step follows from Eq. (64) and Eq. (65), the fourth step follows from merging the terms related to $\delta^{(j)}$.

We can show that

$$r \cdot L \cdot (1 + 4/p) = r \cdot \max\{4, 8/p\}(1 + 4/p)$$
$$\le r \cdot (4 + 8/p)(1 + 4/p)$$
$$\le \frac{p^2(4 - p)}{2^{20}} \cdot (4 + 8/p)(1 + 4/p)$$
$$\le \frac{1}{2^{10}}(1 - p/4), \tag{67}$$

where the first step follows from choice of $L$ (see Lemma statement), the second step follows from $\max\{a, b\} \le a + b$ for $a, b \ge 0$, the third step follows from choice of $r$ (see Lemma statement), and the last step follows from Fact C.14.

We have

$$40(1 + 4/p)r \le \frac{1}{2L} \cdot (1 - p/4)$$
$$\le \frac{1}{2L} \cdot (2/p - |1 - 2/p|), \tag{68}$$

where the first step follows from Eq. (67), and the second step follows from Fact C.13.

And hence

$$\delta^{(j+1)} \le (1 - \underbrace{\frac{1}{2L}(2/p - |1 - 2/p|)}_{:=\alpha})\delta^{(j)} + \underbrace{\frac{3\delta}{L}}_{:=\beta} , \qquad (69)$$

where the step follows from Eq. (68) and Eq. (66).

Recursively applying the above equation, we can show

$$
\begin{aligned}
\delta^{(j)} &\le (1 - \alpha)\delta^{(j-1)} + \beta \\
&\le (1 - \alpha)^2 \delta^{(j-2)} + (1 - \alpha)\beta + \beta \\
&\le \cdots \\
&\le (1 - \alpha)^j \cdot \delta^{(0)} + \sum_{i=0}^{j-1}(1 - \alpha)^i \beta \\
&= \sum_{i=0}^{j-1}(1 - \alpha)^i \beta \\
&= \frac{1 - (1 - \alpha)^j}{\alpha}\beta \\
&\le \frac{\beta}{\alpha},
\end{aligned}
$$

where the first step follows from using Eq. (68) for $j$, the second step follows using Eq. (68) for $j - 1$, the fifth step follows from $\delta^{(0)}$ (in that 0-th iteration, we can trivially think $v(0) = w(0)$), the sixth step follows geometric sum, and the last step follows from simple algebra.

Using $\alpha = \frac{1}{2L}(2/p - |1 - 2/p|)$ and $\beta = 3\delta/L$ to substitute $\alpha$ and $\beta$ in the bound above, we further have

$$
\begin{aligned}
\delta^{(j)} &\le \frac{1}{\frac{1}{2L}(2/p - |1 - 2/p|)} \cdot \frac{3\delta}{L} \\
&\le \frac{8\delta}{2/p - |1 - 2/p|} \\
&\le \frac{1}{4}\epsilon,
\end{aligned}
$$

where the first step follows from Eq. (69), the second step follows from basic algebra, and the third step follows from $\delta = \frac{4}{256} \cdot \epsilon \cdot (1 - p/4) \le \frac{1}{32} \cdot \epsilon \cdot (2/p - |1 - 2/p|)$ (Fact C.13).

Recalling that

$$k = \lceil 80(2/p + 2)p \log(\frac{pn}{32\epsilon}) \rceil,$$

we have

$$
\begin{aligned}
\|W_p^{-1}(w_p - w(k))\|_\infty &\le \|W_p^{-1}(w_p - v(k))\|_\infty + \|W_p^{-1}(v(k) - w(k))\|_\infty \\
&\le 4(1 + 2/p)^2 \sqrt{n} \cdot \|w - w_p\|_{W_p^{-1}} + 2\delta^{(k)} \\
&\le 4(1 + 2/p)^2 \sqrt{n} \cdot 2\sqrt{n} \cdot (1 - \frac{1}{16(2/p + 2)})^{\frac{k}{2}} \cdot \frac{p}{160} + 2\delta^{(k)} \\
&\le \epsilon/2 + 2\delta^{(k)} \\
&\le \epsilon,
\end{aligned}
$$

where the first step follows from triangle inequality, the second step follows from Lemma E.13, the third step follows from Lemma E.12, the fourth step follows from choice of $k$, and the last step follows from $\delta^{(k)} \le \epsilon/4$.

$\square$

### E.11  COMPUTE LEVERAGE SCORE

We provide an algorithm and theorem statement below most closely resembling the one from (Spielman & Srivastava, 2008).

---

**Algorithm 4** Compute leverage score

---

1: **procedure** COMPUTELEVERAGESCORES($A \in \mathbb{R}^{m \times n}, \epsilon \in (0, 1)$)
2:    Let $q^{(j)}$ be $k$ random $\pm 1/\sqrt{k}$ vectors of length $m$ with $k = O(\log(m)/\epsilon^2)$
3:    Compute $l^{(j)} \leftarrow (A^\top A)^{-1} q^{(j)}$ and $p^{(j)} \leftarrow Al^{(i)}$
4:    **return** $\sum_{j=1}^{k} (p_i^{(j)})^2$
5: **end procedure**

---

**Lemma E.17** (Lemma 59 in page 56 in (Lee & Sidford, 2019)). *If the following conditions hold:*

- *For $\epsilon \in (0, 1)$ with probability at least $1 - \frac{1}{m^{O(1)}}$.*

*The algorithm* COMPUTELEVERAGESCORES *returns* $\sigma^{(\mathrm{apx})}$ *such that for all* $i \in [m], (1 - \epsilon)\sigma(A)_i \leq \sigma_i^{(\mathrm{apx})} \leq (1 + \epsilon)\sigma(A)_i$, *by solving only* $O(\epsilon^{-2} \log m)$ *linear systems.*

### E.12  INITIAL WEIGHT

---

**Algorithm 5** Compute initial weight

---

1: **procedure** COMPUTEINITIALWEIGHT($A \in \mathbb{R}^{m \times n}, p_{\mathrm{target}} \in (0, 4), \epsilon \in (0, 1)$)
2:    $p \leftarrow 2$
3:    **while** $p \neq p_{\mathrm{target}}$ **do**
4:        Let $r$ be defined as in COMPUTEAPXWEIGHT or COMPUTEEXACTWEIGHT
5:        $h \leftarrow \frac{\min\{2,p\}}{\sqrt{n \log \frac{m\epsilon^2}{n}}} \cdot r$
6:        $p^{(\mathrm{new})} \leftarrow \mathrm{median}(p - h, p_{\mathrm{target}}, p + h)$
7:        Either $w \leftarrow$ COMPUTEAPXWEIGHT($p^{(\mathrm{new})}, w\frac{p^{(\mathrm{new})}}{p}, \frac{r}{4}$)
8:        Or $w \leftarrow$ COMPUTEEXACTWEIGHT($p^{(\mathrm{new})}, w\frac{p^{(\mathrm{new})}}{p}, \frac{r}{4}$)
9:        $p \leftarrow p^{(\mathrm{new})}$
10:    **end while**
11:    **return** COMPUTEAPXWEIGHT($p_{\mathrm{target}}, w, \epsilon$)
12: **end procedure**

---

**Lemma E.18** (Lemma 60 in page 57 in (Lee & Sidford, 2019)). *If the following conditions hold:*

- *Let $m \geq n$.*

- *Let $q > 0$.*

- *Let $\widetilde{w}_q \in \mathbb{R}_{>0}^m$ denote the vector with $\widetilde{w}_{q,i} = w_p(A)_i^{q/p}$ for all $i \in [m]$.*

- *Let $|p - q| \leq \frac{\min\{2,p\}}{\sqrt{n} \log(me^2/n)}$.*

*Then we have*

$$\| \log(\frac{w_q(A)}{\widetilde{w}_q})\|_\infty \leq \max\{1/2, 1/p\}\sqrt{n} \log(\frac{me^2}{n}).$$

*Proof.* For notational convenience, let $w := w_p(A)$, $W := \mathrm{diag}(w)$ and $\Lambda := \Lambda(W^{1/2-1/p}A)$.

We have

$$\frac{\mathrm{d}w}{\mathrm{d}p} = \frac{\mathrm{d}w_p(A)}{\mathrm{d}p}$$

$$= 2\Lambda(\frac{(1/2-1/p)w^{-1/2-1/p}\frac{\mathrm{d}w}{\mathrm{d}p} + w^{1/2-1/p}\frac{1}{p^2}\log w}{w^{1/2-1/p}})$$

$$= \Lambda((1-2/p)W^{-1}\frac{\mathrm{d}w}{\mathrm{d}p} + \frac{2}{p^2}\log w), \tag{70}$$

where the first step follows from $w = w_p(A)$, the second step follows from taking the derivative with respect to $p$ on both sides and Part 21 of Lemma D.16, and the last step follows from basic algebra.

We define $J := J_w(\mathbf{1}_m)$.

Since Eq. (70) is an equation about $\frac{\mathrm{d}w}{\mathrm{d}p}$, we can solve for $\frac{\mathrm{d}w}{\mathrm{d}p}$ and obtain the following:

$$\frac{\mathrm{d}w}{\mathrm{d}p} = 2W(W - (1-2/p)\Lambda)^{-1}\Lambda\frac{\log w}{p^2}$$

$$= J\frac{\log w}{p^2}, \tag{71}$$

where the first step follows from Eq. (70), the second step follows from Lemma E.5.

And for all $h \in \mathbb{R}^m$,

$$\|(W^{-1}J - p)h\|_\infty \le p \cdot \max\{p/2, 1\} \cdot \|h\|_W, \tag{72}$$

where the step follows from Lemma E.6.

We have

$$\|W^{-1}\frac{\mathrm{d}w_p(A)}{\mathrm{d}p} - \frac{\log w_p}{p}\|_\infty = \|W^{-1}J\frac{\log w}{p^2} - \frac{\log w_p}{p}\|_\infty$$

$$\le p \cdot \max\{p/2, 1\} \cdot \|p^{-2}\log w\|_W$$

$$\le \max\{1/2, 1/p\} \cdot \|\log w\|_W, \tag{73}$$

where the first step follows from Eq. (71), the second step follows from Eq. (72) and $h = p^{-2}\log w$, and the last step follows from basic algebra.

We define

$$a_1 := \sum_{w_i \in (0, \frac{1}{e}]} w_i \log^2 w_i,$$

$$a_2 := \sum_{w_i \in (\frac{1}{e}, 1]} w_i \log^2 w_i.$$

Finally,

$$\|\log w\|_W^2 = \sum_{i=1}^m w_i \log^2 w_i$$

$$= a_1 + a_2,$$

where the first step follows from the definition of $\|\cdot\|_W$, and the second step follows from splitting the sum.

For the first term,

$$a_1 = \sum_{w_i \in (0, 1/e]} w_i \log^2 w_i$$

$$= m \cdot \frac{1}{m} \sum_{w_i \in (0, 1/e]} w_i \log^2 w_i$$

$$= m \cdot \frac{1}{m} \sum_{w_i \in (0, 1/e]} f(w_i)$$

$$\leq m \cdot f(\frac{1}{m} \sum_{w_i \in (0, 1/e]} w_i)$$

$$\leq m \cdot f(\frac{n}{m})$$

$$\leq m \cdot \frac{n}{m} \log^2(n/m)$$

$$= m \cdot \frac{n}{m} \log^2(m/n)$$

$$= n \log^2 \frac{m}{n}, \tag{74}$$

where the first step follows from definition of $a_1$, the second step follows from basic algebra, the third step follows from $f(x) := x \log^2 x$, the fourth step follows from that $f$ is concave, the fifth step follows from $\sum_{w_i \in (0, 1/e]} w_i \leq n$, the sixth step follows from the definition of $f$, the seventh step follows from $\log(1/x) = -\log(x)$, and the last step follows from basic algebra.

For the second term,

$$a_2 \leq \sum_{w_i \in (\frac{1}{e}, 1]} w_i$$

$$\leq n, \tag{75}$$

where the first step follows from $\log^2 w_i \leq 1$ for $w_i \in (\frac{1}{e}, 1]$, the second step follows from $\sum_{w_i \in [n]} w_i \leq n$.

Thus, we have

$$\| \log w \|_W^2 \leq n \log^2 \frac{m}{n} + n$$

$$\leq n \log^2 \frac{em}{n},$$

where the first step follows from Eq. (74) and Eq. (75), and the second step follows from $(x+y)^2 \geq x^2 + y^2$ for $x, y \in \mathbb{R}$

We can compute

$$\frac{\mathrm{d}}{\mathrm{d}q} \log(\widetilde{w}_q) = \frac{\mathrm{d}}{\mathrm{d}q}(q/p) \log(w_p(A))$$

$$= p^{-1} \log(w_p(A))$$

$$= q^{-1}(q/p) \log(w_p(A))$$

$$= q^{-1} \log(\widetilde{w}_q).$$

Thus, for all $q$, $w_q := w_q(A)$, and $W_q := \mathrm{diag}(w_q)$, we have

$$\| \frac{\mathrm{d}}{\mathrm{d}q} \log(w_q/\widetilde{w}_q) \|_\infty = \| \frac{\mathrm{d} \log(w_q)}{\mathrm{d}q} - \frac{\mathrm{d} \log(\widetilde{w}_q)}{\mathrm{d}q} \|_\infty$$

$$= \| W_q^{-1} \frac{\mathrm{d}}{\mathrm{d}q} w_q - q^{-1} \log(\widetilde{w}_q) \|_\infty$$

$$= \| W_q^{-1} \frac{\mathrm{d}}{\mathrm{d}q} w_q - q^{-1} \log(w_q) + q^{-1} \log(w_q) - q^{-1} \log(\widetilde{w}_q) \|_\infty$$

$$\leq \| W_q^{-1} \frac{\mathrm{d}}{\mathrm{d}q} w_q - q^{-1} \log(w_q) \|_\infty + \| q^{-1} \log(w_q) - q^{-1} \log(\widetilde{w}_q) \|_\infty$$

$$\leq \max\{1/2, 1/q\} \sqrt{n} \log(\frac{me}{n}) + q^{-1} \| \log(w_q/\widetilde{w}_q) \|_\infty$$

$$\leq \max\{1/2, 1/q\} \sqrt{n} \log(\frac{em}{n}) + q^{-1}$$

$$\leq \max\{1/2, 1/q\} \sqrt{n} \log(\frac{me^2}{n}),$$

where the first step follows from basic algebra, the second step follows from computing the derivative, the third step follows from adding a term and removing the same term, the fourth step follows from triangle inequality, the fifth step follows from Eq. (73), the sixth step follows from $\|\log(w_q/\widetilde{w}_q)\|_\infty \leq 1$( the reason is letting $\delta$ be the largest number for which $q$ satisfying $|p-q| \leq \delta$, it will imply that equation), and the last step follows from $q^{-1} \leq \max\{1/2, 1/q\}\sqrt{n}$ and merging the logarithm.

Therefore, it must be the case that

$$\delta \leq (\max\{1/2, 1/q\}\sqrt{n}\log(\frac{me^2}{n}))^{-1}$$

and the result follows. $\qquad\square$

### E.13 THEOREM OF EXACT WEIGHT COMPUTATION

**Theorem E.19** (Exact Weight Computation, Theorem 45 in page 40 in (Lee & Sidford, 2019)). *If the following conditions hold:*

- *Let $A \in \mathbb{R}^{m \times n}$ be a non-degenerate matrix.*

- *Let $\epsilon \in (0, 1)$.*

- *Let $p \in (0, \infty)$.*

- *Let $w(0) \in \mathbb{R}^m_{>0}$ with $\|w(0)^{-1}(w_p(A) - w(0))\|_\infty \leq \frac{p}{20(p+2)}$.*

*Then, we have*

- **Part 1.** *The algorithm* COMPUTEEXACTWEIGHT$(A, p, w(0), \epsilon)$ *(Algorithm 2) can be implemented to return $w$*

  - *so that $w$ satisfies $\|w_p(A)^{-1}(w_p(A) - w)\|_\infty \leq \epsilon$,*
  - *the algorithm uses $O(mn^{w-1}(p + 1/p)\log(n(1 + 1/p)\epsilon^{-1}))$ work,*
  - *$O((p + 1/p)\log(m)\log(n(1 + 1/p)\epsilon^{-1}))$ depth.*

- **Part 2.** *Without $w(0)$, the algorithm* COMPUTEINITIALWEIGHT$(A, p, \epsilon)$ *(Algorithm 5) can be implemented to achieve the same guarantee*

  - *with $O(mn^{w-(1/2)}(p + p^{-1})^2 \log\frac{m}{n}\log(n\epsilon^{-1}(p + p^{-1})))$ work,*
  - *$O((p + p^{-1})^2\log(\frac{m}{n})\log(m)\log(n\epsilon^{-1}(p + p^{-1})))$depth.*

*Proof.* From Lemma E.18, we know each step of $p$ lies within the requirement of Theorem E.14.

Furthermore, Lemma E.18 shows that it takes

$$O(\sqrt{n}(p + 1/p)\log(\frac{m}{n}))$$

steps in the COMPUTEINITIALWEIGHT.

Each call of COMPUTEEXACTWEIGHT involves

$$O((p + 1/p)\log(n\epsilon^{-1}(1 + 1/p))$$

iterations and each iteration takes $O(mn^{w-1})$ work and $O(\log m)$ depth to compute leverage score.
$\qquad\square$

### E.14 THEOREM OF APPROXIMATE WEIGHT COMPUTATION

**Theorem E.20** (Approximate Weight Computation, Theorem 39 in page 58 in (Lee & Sidford, 2019)). *If the following conditions hold:*

- *Let $A \in \mathbb{R}^{m \times n}$ be non-degenerate.*

- Let $\mathcal{T}_w$ and $\mathcal{T}_d$ denote the work and depth needed to compute $(A^\top D A)^{-1} z$ for an arbitrary positive diagonal matrix $D$ and vector $z$.

- Let $\epsilon \in (0, 1)$.

- Let $p \in (0, 4)$.

- Define $r := 2^{-20} p^2 (4 - p)$.

- Let $w(0) \in \mathbb{R}^m_{>0}$ with $\|w(0)^{-1}(w_p(A) - w(0))\|_\infty \le r$.

*Then we have*

- **Part 1.** *The algorithm* COMPUTEAPXWEIGHT$(x, w(0), \epsilon)$ *can be implemented to return* $w$ *such that*

  - *with high probability in* $\|w_p(A)^{-1}(w_p(A) - w)\|_\infty \le \epsilon$ *in* $O(p^{-1}(4 - p)^{-2} \epsilon^{-2} \log^2(n/(p\epsilon)))$ *steps,*
  - *each of which can be implemented in* $O(\mathrm{nnz}(A) + \mathcal{T}_w)$ *work and* $O(\mathcal{T}_d)$ *depth.*

- **Part 2.** *Without* $w(0)$, *the algorithm* COMPUTEINITIALWEIGHT$(A, p, \epsilon)$ *(Algorithm 5) can be implemented to have the same guarantee*

  - *with* $O(\sqrt{n}(4 - p)^{-3} p^{-3}) \log(\frac{m}{n}) \log^2(n/(p\epsilon))$ *steps of the same cost.*

*Proof.* From Lemma E.18, we know each step of $p$ lies within the requirement of Theorem E.16.

Furthermore, Lemma E.18 shows that it takes

$$O(\sqrt{n}((4 - p)^{-1} + p^{-2}) \log \frac{m}{n})$$

steps in the COMPUTEINITIALWEIGHT. Each call of COMPUTEAPXWEIGHT involves

$$O(p^{-1} \log(n/(p\epsilon)))$$

iterations and each iteration involves computing leverage score up to accuracy

$$\frac{\epsilon}{32(2/p - |1 - 2/p|)} = \Theta((4 - p) \cdot \epsilon).$$

Finally, Lemma E.17 shows this involves solving

$$O((4 - p)^{-2} \epsilon^{-2} \log m)$$

linear systems. $\qquad\square$

### E.15 Weight Function Theorem

**Theorem E.21** (Theorem 29 in page 25 in (Lee & Sidford, 2019))**.** *If the following conditions hold:*

- *Define* $A_x := (\Phi''(x))^{-1/2} A$.

- *Let* $p \in (0, 1)$.

- *Let* $c_0 \ge 0$.

- *Define the weight function* $g : \Omega^\circ \to \mathbb{R}^m_{>0}$ *for all* $x \in \mathbb{R}^m_{>0}$ *as* $g(x) := w_p(A_x) + c_0$.

*Then we have,*

- **Part 1.** $c_1(g) \le n + c_0 m, c_s(g) \le 2m^{1-p}$, *and* $c_k(g) \le \frac{2}{1-p}$.

- **Part 2.** *For* $p = 1 - \frac{1}{\log(4m)}$ *and* $c_0 = \frac{n}{2m}$, *we have*

- $c_1(g) \leq \frac{3}{2}n, c_s(g) \leq 4$
- *and $c_k(g) \leq 2\log(4m)$.*

*Proof.* To bound the size, $c_1(g)$, recall that $w_p(A_x) = \sigma(W^{1/2-1/p}A_x)$ and therefore Lemma D.17 implies for $i \in [m]$,

$$\sum_{i=1}^{m} w_p(A_x)_i = n.$$

We define $\alpha := 2(1-p)$.

To bound the sensitivity $c_s(g)$, then for $i \in [m]$,

$$e_i^\top G(x)^{-1} A_x (A_x^\top G(x)^{-1} A_x)^{-1} A_x^\top G(x)^{-1} e_i = g(x)_i^{-1} \sigma(G(x)^{-1/2} A_x)_i$$
$$\leq 2m^{\frac{1+\alpha}{\alpha}}$$
$$\leq 2m^{1-p},$$

where the first step follows from the definition of leverage score, and the second step follows from Lemma E.8 (with choosing $r = \infty$), the third step follows from $\frac{\alpha}{1+\alpha} = \frac{2-2p}{2-2p+p} \leq 1 - p$.

Using the following two equations,

$$\frac{dg(x)_j}{dx_i} = \frac{dw_{p,j}(A_x)}{dx_i}$$
$$= \frac{dw_j}{dx_i}$$

and

$$\frac{dw_j}{d\Phi''(x)_i^{-1/2}} = \frac{dw_j}{dx_i} \cdot \frac{dx_i}{d\Phi''(x)_i^{-1/2}}$$
$$= \frac{dw_j}{dx_i} \cdot \left(\frac{\Phi''(x)_i^{-1/2}}{dx_i}\right)^{-1}$$
$$= \frac{dw_j}{dx_i} \cdot (-0.5 \cdot \Phi''(x)_i^{-1.5}\Phi'''(x)_i)^{-1}.$$

Then, we can show

$$J_g(x)(\Phi''(x))^{1.5}(-2)(\Phi'''(x))^{-1} = J_w((\Phi''(x))^{-1/2}).$$

Multiplying $G(x)^{-1} \cdot () \cdot z$ on the both sides of above equation, we get

$$G(x)^{-1}J_g(x)(\Phi''(x))^{1.5}(-2)(\Phi'''(x))^{-1}z = G(x)^{-1}J_w((\Phi''(x))^{-1/2})z,$$

which is equivalent to for an arbitrary $h \in \mathbb{R}^m$

$$G(x)^{-1}J_g(x)(\Phi''(x))^{-1/2}h = G(x)^{-1}J_w((\Phi''(x))^{-1/2})z, \tag{76}$$

where $z = -0.5(\Phi''(x))^{-2}\Phi'''(x)h$.

We have that

$$\|G(x)^{-1}J_w((\Phi''(x))^{-1/2})z\|_{g(x)} \leq p\|(\Phi''(x))^{1/2}z\|_{g(x)}, \tag{77}$$

where the step follows from Part 1 of Lemma E.6.

We can show

$$\|G(x)^{-1}J_w((\Phi''(x))^{-1/2})z\|_\infty$$
$$\leq p\|(\Phi''(x))^{1/2}z\|_\infty + \|G(x)^{-1}J_w((\Phi''(x))^{-1/2})z - p(\Phi''(x))^{1/2}z\|_\infty$$

$$\leq p\|(\Phi''(x))^{1/2}z\|_\infty + p\cdot \max\{p/2,1\}\|(\Phi''(x))^{1/2}z\|_{g(x)}$$

$$\leq p\|(\Phi''(x))^{1/2}z\|_\infty + p\|(\Phi''(x))^{1/2}z\|_{g(x)}, \tag{78}$$

where the first step follows from triangle inequality, the second step follows from Part 2 of Lemma E.6, and the last step follows from $p \in (0,1)$.

We have

$$\|G(x)^{-1}J_g(x)(\Phi''(x))^{-1/2}h\|_{g(x)+\infty} = \|G(x)J_w((\Phi''(x))^{-1/2})z\|_{g(x)+\infty}$$

Then, we can show

$$\|G(x)^{-1}J_w((\Phi''(x))^{-1/2})z\|_{g(x)+\infty}$$
$$= \|G(x)^{-1}J_w((\Phi''(x))^{-1/2})z\|_\infty + C_{\mathrm{norm}}\|G(x)^{-1}J_w((\Phi''(x))^{-1/2})z\|_{g(x)}$$
$$\leq p\|(\Phi''(x))^{1/2}z\|_\infty + p(1+C_{\mathrm{norm}})\cdot\|(\Phi''(x))^{1/2}z\|_{g(x)}, \tag{79}$$

where the first step follows from $\|\cdot\|_{g(x)+\infty} = \|\cdot\|_\infty + C_{\mathrm{norm}}\cdot\|\cdot\|_{g(x)}$ (see Definition D.12), and the second step follows from Eq. (76).

Note that

$$|(\Phi''(x))^{1/2}z|_i = 0.5|\Phi''(x)^{-3/2}\Phi'''(x)h|_i$$
$$\leq |h|_i, \tag{80}$$

where the first step follows from choice of $z$, the second step follows from property of $\Phi$ (see Definition D.2).

Therefore,

$$\|G(x)^{-1}J_w((\Phi''(x))^{-1/2})z\|_{g(x)+\infty} \leq p\|(\Phi''(x))^{1/2}z\|_\infty + p(1+C_{\mathrm{norm}})\cdot\|(\Phi''(x))^{1/2}z\|_{g(x)}$$
$$\leq p\|h\|_\infty + p(1+C_{\mathrm{norm}})\cdot\|h\|_{g(x)}$$
$$\leq p(1+\frac{1}{C_{\mathrm{norm}}})\|h\|_\infty + p(1+C_{\mathrm{norm}})\cdot\|h\|_{g(x)}$$
$$= p(1+\frac{1}{C_{\mathrm{norm}}})(\|h\|_\infty + C_{\mathrm{norm}}\cdot\|h\|_{g(x)})$$
$$= p(1+\frac{1}{C_{\mathrm{norm}}})\cdot\|h\|_{g(x)+\infty},$$

where the first step follows from Eq. (79), the second step follows from Eq. (80) and Fact C.4, the third step follows from $C_{\mathrm{norm}} > 0$, the fourth step follows from basic algebra, and the last step follows from the definition of $\|\cdot\|_{g(x)+\infty}$ (see Definition D.12).

Thus, following Eq. (76), we further have

$$\|G(x)^{-1}J_g(x)(\Phi''(x))^{-1/2}h\|_{g(x)+\infty} \leq p(1+\frac{1}{C_{\mathrm{norm}}})\cdot\|h\|_{g(x)+\infty}.$$

The bound of $c_k(g) = \frac{2}{1-p}$ follows from

$$p(1+\frac{1}{C_{\mathrm{norm}}})$$
$$\leq p + \frac{1}{C_{\mathrm{norm}}}$$
$$= p + \frac{1}{24\sqrt{c_s(g)c_k(g)}}$$
$$\leq p + \frac{1}{24c_k(g)}$$
$$= 1 - \frac{2}{c_k(g)} + \frac{1}{24c_k(g)}$$

$$\leq 1 - \frac{1}{c_k(g)},$$

where the first step follows from $p \in (0, 1)$, the second step follows from $C_{\text{norm}} = 24\sqrt{c_s(g)}c_k(g)$, the third step follows from $c_s(g) \geq 1$, the fourth step follows from $p = 1 - \frac{2}{c_k(g)}$, and the last step follows from simple algebra. $\qquad\square$

## LLM USAGE DISCLOSURE

LLMs were used only to polish language, such as grammar and wording. These models did not contribute to idea creation or writing, and the authors take full responsibility for this paper's content.

