# OpenReview forum: "Differentially Private Lewis Weight Computation"
_ICLR.cc/2026/Conference — Submitted to ICLR 2026_

### Official Review · Reviewer_C3zb · 2025-10-29

**Soundness:** 3
**Presentation:** 2
**Contribution:** 3
**Rating:** 4
**Confidence:** 2

**Summary:**

In this work, the authors present a privacy-preserving algorithm for Lewis weight computation that provides formal differential privacy guarantees. The theoretical analysis establishes both convergence and privacy properties, offering an effective balance between utility and privacy in Lewis weight estimation.

The major contributions claimed in the paper:

C1: It establishes a formal DP guarantee for Lewis weight computation using a new optimization-based analysis.

C2: It proves the convergence and privacy of the proposed DP-Lewis Weight algorithm under truncated Gaussian perturbations.

C3: It provides a unified perturbation framework for weighted leverage scores.

**Strengths:**

S1: The paper establishes rigorous sensitivity bounds and privacy analysis grounded in Rényi DP and Gaussian noise calibration.

S2: The paper formally proves convergence under DP noise and provide explicit composition accounting.

S3: The paper is well-organized and logically structured, presenting definitions, lemmas, and proofs in a coherent sequence.

**Weaknesses:**

W1: There are no experiments (even small synthetic ones) to illustrate calibration, convergence speed, or practical accuracy–privacy trade-offs.

W2: The paper looks very notation-heavy, which may reduce readability for readers who are not already familiar with Lewis weights or matrix perturbation theory. It would improve accessibility to include brief intuitive explanations or high-level summaries before diving into the technical details. For example, in Section 4.1, a short roadmap paragraph explaining how Lemmas 4.1–4.3 build toward Theorem 4.4 would help readers follow the logical flow.

W3: While the paper is mathematically well-motivated, the practical motivation could be made clearer. It is better to explicitly explain why each row of the input matrix should be considered a sensitive unit (e.g., in the sections of introduction or preliminary, give an example that a row representing an individual participant, record, or interaction).

W4: Again, the paper’s main results, Theorem 4.4 (Global sensitivity bound), Theorem 4.6 (DP guarantee), and Theorem 4.7, are all presented with detailed derivations but minimal intuitive interpretation.

**Questions:**

Q1: Could the authors clarify whether the privacy analysis is conducted on the log-domain variable log(σ_i) or directly on the weight variable σ_i? How is the Lipschitz constant and sensitivity bound defined consistently under this choice?

Q2: Is there more specific example showing that rows of input matrix are sensitive information?

Q3: Could the authors explain the practical meaning or implications of Theorem 4.4, Theorem 4.6, and Theorem 4.7?

---

### Official Review · Reviewer_CtEg · 2025-10-30

**Soundness:** 2
**Presentation:** 2
**Contribution:** 1
**Rating:** 2
**Confidence:** 4

**Summary:**

The paper presents an algorithm that estimates the (row) leverage scores of a matrix in a differentially private manner. The new notion of neighborhood has been adapted for this problem, and the corresponding sensitivity has been characterized. The privacy guarantee and convergence error bound of the proposed algorithm have been presented.

**Strengths:**

The problem the paper is trying to address is interesting and not easy to address. However there are several technical concerns which outweigh the positives. Please see the questions textbox below.

**Weaknesses:**

Please see the questions textbox below

**Questions:**

Major comments:
1. In Algorithm 1, $m$ leverage scores are released in each iteration. However, the privacy guarantee in Lemma 4.5 does not account for the release of this $m$-dimensional vector, but just that of a single leverage score $\tilde{\sigma}_i$. Hence, the privacy guarantee seems incorrect.
2. In contributions and conclusions, it has been stated that truncated Gaussian noise is added to the estimated leverage scores at each iteration. Moreover, step 6 in Algorithm 1 also requires bounded drift in the leverage score estimates. However, the privacy guarantees in Lemma 4.5 and Theorem 4.6 consider (non-truncated) Gaussian noise. This raises concerns about the validity and relevance of the results.
3. In line 368, $\log(\sigma_i)$ is added with Gaussian noise. It will result in a perturbation of $\sigma_i$ as given in (2), i.e., $\tilde{\sigma}_i = \sigma_i e^{z_i}$. However, this is inconsistent with the perturbations considered in the statements of Lemma 4.5 and Theorem 4.6 ($\tilde{\sigma}_i = \sigma_i (1+z_i)$).
4. No proof/validation has been provided for the claim of optimal privacy-utility trade-off. Also, it is crucial to include empirical results that (i) validate the theoretical results and (ii) demonstrate the efficacy of the algorithm.
5. The vector being perturbed is the hat matrix diagonal and by definitions its entries have to be between 0 and 1, however the normal perturbation can result in the entries falling outside this range and that seems to have been overlooked

Minor comments:
1. Please clarify why the optimal Rényi order has not been considered for the conversion in Theorem 4.6 (see Remark B.4. in https://arxiv.org/abs/2110.11688). Also, it is unclear how the number of iterations ($T$ in step 3 of Algorithm 1) is obtained from the convergence error ($\epsilon$ in line 436).
2. There are several incomplete sentences: lines 228, 340, 393.
3. Several entities are undefined or appear before they appear ($\kappa$, $\sigma_{\max}$, $\sigma_{\min}$, $f(w,A)$, $\mathcal{T}_w$,  $\mathcal{T}_d$). Also, the use of similar notations for different entities ($\sigma$, $\epsilon$, $\delta$) leads to confusion and ambiguity.
4. It is recommended to avoid different notations for the same operation (lines 210-211).
5. Wrong reference to a lemma: line 429.
6. Typo in line 235 ('= \neq').
7. Also, there are several grammatical and typographic errors, especially missing punctuation (for example, after 'thus'), 'is' instead of 'are' in lines 297 and 304, etc.

---

### Official Review · Reviewer_EmsC · 2025-10-30

**Soundness:** 2
**Presentation:** 1
**Contribution:** 2
**Rating:** 2
**Confidence:** 2

**Summary:**

This paper studies the problem of estimating the Lewis weights of a non-degenerate data matrix $A \in \mathbb{R}^{m \times n}$ where each row corresponds to a user. The Lewis weights of $A$ are a vector of $m$ weights, one per row / user, indicating the importance of that row in a way that generalizes leverage scores.

This paper considers two data matrices $A$ and $A'$ to be neighbors if they differ on at most one row, and the $\ell_2$ difference in that row is at most $\epsilon_0$. They introduce a mechanism for approximating the $\ell_p$ Lewis weights for $p \in (0, 4)$ and argue in Theorem 4.7 that is satisfies differential privacy and has a utility guarantee.

**Strengths:**

Approximating the Lewis weights of a data matrix privately is an interesting problem that has many applications.

**Weaknesses:**

I have two main concerns with the paper: first, I think the organization could be greatly improved to make the key results easier to understand and to move more of the main ideas into the main body from the appendix. Second, it appears to me that there is a fundamental conflict between releasing accurate Lewis weights and privacy. More detailed comments are below:

**Privacy Conflict:**
In the problem setup each row of the data matrix $A$ represents a single user, and two matrices $A$ and $A'$ are neighboring if they agree on all users except one, and the $\ell_2$ norm of the difference for that user is at most $\epsilon_0$.
Intuitively the privacy guaranteed by this neighboring definition is that an adversary who sees the output of the mechanism will not be able to confidently distinguish between the real row for a user and an alternative row that is within distance $\epsilon_0$.
If $\epsilon_0$ is too small, this does not correspond to a meaningful privacy guarantee (since it could be that identifying a row within distance $\epsilon_0$ of the user's true row is enough to violate their privacy).
So, the results of this paper are useful in the situation where we can achieve high utility in the case when $\epsilon_0$ is large, ideally it should be large enough that replacing row $i$ of $A$ by the data in row $j$ of $A$ should result in a neighboring matrix $A'$ (since this is substituting the data for user $j$ in place of the data for user $i$).

But it seems to me that if $\epsilon_0$ is large enough to allow this then it should be impossible to get good utility: let $A$ be a data matrix such that users $i$ and $j$ have significantly different Lewis scores. Now let $A'$ be the dataset obtained by swapping rows $i$ and $j$, which also swaps the Lewis scores of rows $i$ and $j$. $A$ and $A'$ are just two neighboring hops away, but there are two Lewis scores that they significantly disagree on (namely the $i$th and $j$th Lewis scores). If the mechanism is private with strong parameters, it cannot change its distribution significantly between $A$ and $A'$, but it is required to do so in order to get high utility.

It is possible that the results from the paper do show that for large enough $\epsilon_0$ to get a meaningful privacy guarantee the mechanism has high utility, but I couldn't easily understand the relationship between these parameters, the privacy guarantee, and the utility guarantee.

**Clarity:** The paper provides little to no explanation of the results presented in the main body . For example:
- I get the impression that Section 4.1 is dealing with some sensitivity analysis, but it is not clear to me how these Lemmas are related to the mechanism described in Algorithm 1 and its privacy analysis.
- In Theorem 4.6 it is not stated what $\epsilon_{DP}$ and $\delta_{DP}$ are (though there is an upper bound on $\epsilon_{DP}$ in the proof).
- It would be helpful to include a few words about why the utility guarantee in Theorem 4.7 corresponds to accurately computing the Lewis weights. In that statement, is $w$ the vector of true Lewis weights for the matrix $A$?

**Questions:**

1. If $\epsilon_0$ is large enough to allow significant changes to the row contributed by a user, do we get a meaningful utility guarantee for the mechanism?
2. Several Lemmas and Theorems require either that $\sigma_i \in [\gamma, 1]$ or $w_i \in [\gamma, 1]$. Are these conditions required for the privacy analysis to go through?

---

### Official Review · Reviewer_ZcZ7 · 2025-11-04

**Soundness:** 3
**Presentation:** 3
**Contribution:** 3
**Rating:** 4
**Confidence:** 2

**Summary:**

This paper studies differential private computation of $\ell_p$ Lewis weights for a data matrix $A$.  Building on Lee and Sidford (2019), the author proposes an iterative algorithm that differentially privately estimates the leverage score $\sigma_i$ by adding appropriate Gaussian noise.  The utility guarantee follows from analyzing the perturbation/sensitivity of the leverage score under small changes in the matrix $A$.

**Strengths:**

The paper provides a careful analysis of perturbation bounds by controlling the Lipschitz constants.

**Weaknesses:**

- The definition of a neighboring dataset is non-standard: $A$ and $A'$ are adjacent if they differ by one role but also have a bounded two norm difference, equivalently $\|A-A'\|_F\le \epsilon_0$.  The paper offers limited motivation for this choice. The authors might consider adding a lower bound or impossibility result to justify why such a restriction is necessary.
- The techniques are largely standard. Algorithm 1 essentially adapts the Lee and Sidford procedure with added Gaussian noise.
- The paper would benefit from a stronger motivation for private Lewis weights specifically.  Otherwise, many related computational problems (singular values, matrix norms, PCA, ...) could also be candidates for privatization.

**Questions:**

What is the motivation before the neighboring definition?
What is the importance of Lewis weights over other related matrix problems?

---

### Meta-Review · Area_Chair_6K9n · 2026-01-07

**Summary:**

The paper proposes a differentially private algorithm for estimating Lewis weights using Gaussian noise injection, analyzing perturbation bounds and Lipschitz constants. However, the submission suffers from significant technical inconsistencies regarding the privacy analysis, a lack of empirical validation, and unclear motivation for the specific problem setting.

**Reviewer Concerns:**

All concerns remain outstanding; specifically, the validity of the privacy guarantee (vector vs. scalar release) and the inconsistency between the algorithm and proofs regarding noise truncation are critical unaddressed flaws. Additionally, the non-standard neighboring definition and the complete absence of experimental results to verify utility trade-offs remain significant issues.

**Reviewer Scores:**

Reviewers EmsC and CtEg would retain their rejection scores due to the fundamental technical errors and clarity issues. Reviewers ZcZ7 and C3zb would likely lower their scores from borderline to reject upon recognizing the severity of the correctness issues raised by CtEg regarding the privacy proofs.

Reasons for Rejection:Technical Incorrectness: There are critical discrepancies between the algorithm and the proofs, specifically regarding truncated vs. non-truncated noise and the failure to account for the privacy cost of releasing a vector vs. a single scalar.

Problem Formulation: The definition of neighboring datasets is non-standard and insufficiently motivated, creating a fundamental conflict where a meaningful privacy parameter ($\beta$) may preclude useful utility.

Lack of Validation: The paper completely lacks empirical experiments to validate the theoretical claims, convergence speed, or practical accuracy-privacy trade-offs.

Clarity and Motivation: The presentation is heavily notation-dense with insufficient intuition, and the paper fails to adequately motivate why private Lewis weights are necessary compared to other related matrix problems.

---

### Decision · Program_Chairs · 2026-01-26

Reject